# Taming Communication and Sample Complexities in Decentralized Policy Evaluation for Cooperative Multi-Agent Reinforcement Learning

**Xin Zhang**[*]
Dept. of Statistics
Iowa State University
Ames, IA 50010
xinzhang@iastate.edu

**Zhuqing Liu**[*]
Dept. of ECE
The Ohio State University
Columbus, OH, 43210
liu.9384@osu.edu

**Jia Liu**
Dept. of ECE
The Ohio State University
Columbus, OH, 43210
liu@ece.osu.edu

**Zhengyuan Zhu**
Dept. of Statistics
Iowa State University
Ames, IA 50010
zhuz@iastate.edu

**Songtao Lu**
IBM Research AI
IBM Thomas J. Watson Research Center
Yorktown Heights, NY 10598
songtao@ibm.com

## Abstract

Cooperative multi-agent reinforcement learning (MARL) has received increasing attention in recent years and has found many scientific and engineering applications. However, a key challenge arising from many cooperative MARL algorithm designs (e.g., the actor-critic framework) is the policy evaluation problem, which can only be conducted in a *decentralized* fashion. In this paper, we focus on decentralized MARL policy evaluation with nonlinear function approximation, which is often seen in deep MARL. We first show that the empirical decentralized MARL policy evaluation problem can be reformulated as a decentralized nonconvex-strongly-concave minimax saddle point problem. We then develop a decentralized gradient-based descent ascent algorithm called GT-GDA that enjoys a convergence rate of $\mathcal{O}(1/T)$. To further reduce the sample complexity, we propose two decentralized stochastic optimization algorithms called GT-SRVR and GT-SRVRI, which enhance GT-GDA by variance reduction techniques. We show that all algorithms all enjoy an $\mathcal{O}(1/T)$ convergence rate to a stationary point of the reformulated minimax problem. Moreover, the fast convergence rates of GT-SRVR and GT-SRVRI imply $\mathcal{O}(\epsilon^{-2})$ communication complexity and $\mathcal{O}(m\sqrt{n}\epsilon^{-2})$ sample complexity, where $m$ is the number of agents and $n$ is the length of trajectories. To our knowledge, this paper is the first work that achieves $\mathcal{O}(\epsilon^{-2})$ in both sample and communication complexities in decentralized policy evaluation for cooperative MARL. Our extensive experiments also corroborate the theoretical results of our proposed decentralized policy evaluation algorithms.

## 1 Introduction

In recent years, multi-agent reinforcement learning (MARL) has found important applications in many scientific and engineering fields, such as robotic network [45, 22, 41], sensor network [6, 40, 43], and power network [5, 11, 15, 16], just to name a few. In MARL, *multiple* agents observe the current joint state over a network, perform their own actions based on the current state, and transition to

---

[*]Xin Zhang and Zhuqing Liu contributed equally to this work.

35th Conference on Neural Information Processing Systems (NeurIPS 2021).

the next joint state. Each agent can only observe its local reward, which is a function of the joint state and actions. One important paradigm in MARL is the *cooperative* MARL, where the agents share a common goal to find an optimal global policy to achieve the maximum global accumulative reward [63, 62, 13, 37]. As in classical reinforcement learning (RL) problems, a key component in cooperative MARL algorithms is the policy evaluation (PE) problem, whose goal is to evaluate the expected long-term accumulative reward for a given global policy. In cooperative MARL, agents share a common value function based on the joint states. As a result, the parameterization of the value function is the same across all agents. Examples of cooperative MARL include but are not limited to traffic light control [56], autonomous driving [21], financial trading [26]), etc. In cooperative MARL, the PE problem emerges as a key step for the agents to find an optimal global policy in MARL tasks [3, 23]. For example, in the actor-critic algorithmic framework for MARL, the actors conduct the policy improvement step, while the critics perform the policy evaluation step and estimates the value function. The overall actor-critic algorithm tries to find an optimal policy by iterating between the policy evaluation and improvement steps. Thus, developing efficient policy evaluation algorithms is critical to the success of RL algorithms based on the actor-critic framework.

However, developing efficient PE algorithms for MARL is highly non-trivial. On one hand, the global accumulative reward is not directly observable in an MARL system. As a result, the PE problem of MARL can only be solved in a *decentralized* fahsion. On the other hand, modern MARL tasks have been increasingly complex and often not directly computable. As a result, MARL often use highly nonlinear parametric models (e.g., deep neural network (DNN)) for policy approximation. In this paper, we focus on PE based on nonlinear function approximations due to the following reasons: 1) linear approximation schemes are based on their pre-defined basis space, which may not be able to approximate the non-linear value function with high accuracy; 2) non-linear neural network approximation can handle the cases where the states space that is mixed with continuous and (infinite) discrete state values; and 3) nonlinear neural network approximation usually have a better generalization performance than linear approximation [61], [20], [60]. However, it has been shown that the convergence performance of RL algorithms with nonlinear function approximations is not guaranteed [49].

In light of the growing importance of MARL, in this paper, we focus on addressing the above challenges. The key contributions of this paper are summarized as follows:

- To our knowledge, this work is the *first* to investigate the decentralized PE (DPE) problem for MARL with *nonlinear* function approximations. Via the Fenchel's duality and local reward decomposition, we first reformulate the DPE problem of MARL as a decentralized non-convex-strongly-concave minimax saddle point problem. To solve the minimax problem in a decentralized fashion, we propose a gradient-tracking-based gradient descent-ascent (GT-GDA) algorithm. We show that GT-GDA enjoys a convergence rate of $\mathcal{O}(1/T)$, which leads to an $\mathcal{O}(mn\epsilon^{-2})$ sample complexity and an $\mathcal{O}(\epsilon^{-2})$ communication complexity, where $m$ is the number of agents, $n$ is the data size, and $\epsilon$ is the convergence accuracy.

- To further reduce the sample complexity, we develop two variance-reduced algorithms, namely gradient-tracking stochastic recursive variance reduction (GT-SRVR) algorithm and its variant with incremental batch size (GT-SRVRI). We show that both algorithms achieve the same communication complexity $\mathcal{O}(\epsilon^{-2})$ as GT-GDA, but requiring a lower sample complexity $\mathcal{O}(m\sqrt{n}\epsilon^{-2})$.

- It is worth noting that, in our theoretical analysis, we relax the commonly-used compactness conditions of the feasible set with some mild assumptions on objectives. Thus, the solutions found by our algorithms are exactly the stationary points for the original policy evaluation problem. This result may be of independent interest for general RL problems.

The rest of the paper is organized as follows. In Section 2, we first provide the preliminaries of the DPE problem of MARL and discuss related works. In Section 3, we first introduce the GT-GDA algorithm, and then propose two stochastic variance reduced algorithms, namely GT-SRVR and GT-SRVRI. We present their theoretical properties in Section 4. Section 5 provides numerical results to verify our theoretical findings, and Section 6 concludes this paper.

## 2    Problem formulation and related work

In this section, we first introduce the DPE problem formulation in in Section 2.1. Then in Section 2.2, we review the recent developments of PE algorithms and compare them with our work. In Sec-

tion 2.3, we highlight the challenges in designing efficient DPE algorithms with nonlinear function approximation and the significance of our work.

## 2.1 Problem formulation of decentralized policy evaluation for MARL

Consider a multi-agent network system $\mathcal{G} = (\mathcal{N}, \mathcal{L})$, where $\mathcal{N}$ and $\mathcal{L}$ denote the sets of agents and edges, respectively, with $|\mathcal{N}| = m$. In the system, the agents cooperatively perform a learning task. The agents can communicate with each other through edges in $\mathcal{L}$. An MARL problem is formulated based on the multi-agent Markov decision process (MDP) framework, which is characterized by a quintuple $(\mathcal{S}, \mathcal{A}, \mathcal{P}^{\mathbf{a}}_{\mathbf{ss}'}, \{\mathcal{R}_i(\mathbf{s}, \mathbf{a})\}_{i=1}^m, \zeta)$, where $\mathcal{S}$ and $\mathcal{A}$ are the state and action spaces, respectively; $\mathbf{s} \in \mathcal{S}$ and $\mathbf{a} \in \mathcal{A}$ are joint state and action; $\mathcal{P}^{\mathbf{a}}_{\mathbf{ss}'}$ is the transition probability from state $\mathbf{s}$ to state $\mathbf{s}'$ after taking action $\mathbf{a}$; $\mathcal{R}_i(\mathbf{s}, \mathbf{a})$ is the local reward received by agent $i$ after taking action $\mathbf{a}$ in state $\mathbf{s}$; and $\zeta \in (0, 1)$ is a discount factor. Both the joint state $\mathbf{s}$ and action $\mathbf{a}$ are available to all agents, while the local reward $\mathcal{R}_i$ is private to agent $i$. In a multi-agent system, the global reward function is defined as the average of the local rewards $\frac{1}{m} \sum_{i=1}^m \mathcal{R}_i(\mathbf{s}, \mathbf{a})$. Moreover, a joint policy $\pi$ specifies sequential decision rules for all agents. Policy $\pi(\mathbf{a}|\mathbf{s})$ is the conditional probability of taking joint action $\mathbf{a}$ given state $\mathbf{s}$. The goal of PE is to estimate the value function of a given policy $\pi$, which is defined as the long-term discounted accumulative reward: $\mathcal{V}^\pi(\mathbf{s}_0) = \mathbb{E}\left[\frac{1}{m} \sum_{t=0}^\infty \zeta^t \sum_{i=1}^m \mathcal{R}_i(\mathbf{s}_t, \mathbf{a}_t)|\mathbf{s}_0, \pi\right]$, where the expectation is taken over all possible state-action trajectories and initial states.

To determine $\mathcal{V}^\pi(\cdot)$, one of the most effective methods is the temporal-difference (TD) learning algorithm, which focuses on solving the Bellman equation for $\mathcal{V}^\pi(\cdot)$: $\mathcal{V}(\mathbf{s}) = \mathcal{T}^\pi \mathcal{V}(\mathbf{s}) \triangleq \frac{1}{m} \sum_{i=1}^m \mathcal{R}_i^\pi(\mathbf{s}) + \zeta \sum_{\mathbf{s}' \in \mathcal{S}} \mathcal{P}^\pi_{\mathbf{ss}'} \mathcal{V}(\mathbf{s}')$, where $\mathcal{T}^\pi$ denotes the Bellman operator, $\mathcal{R}_i^\pi(\mathbf{s}) = \mathbb{E}_{\mathbf{a} \sim \pi(\cdot|\mathbf{s})} \mathcal{R}(\mathbf{s}, \mathbf{a})$ and $\mathcal{P}^\pi_{\mathbf{ss}'} = \mathbb{E}_{\mathbf{a} \sim \pi(\cdot|\mathbf{s})} \mathcal{P}^a_{\mathbf{ss}'}$. However, $\mathcal{P}^\pi_{\mathbf{ss}'}$ is unknown in MARL and the size of the state space $\mathcal{S}$ could be infinite. To address this challenge, a widely adopted approach is to approximate $\mathcal{V}^\pi(\cdot)$ by a function $\mathcal{V}_{\boldsymbol{\theta}}(\cdot)$ parameterized by $\boldsymbol{\theta} \in \mathbb{R}^p$. According to the formulation in [38, 50, 10, 9, 30], the Bellman equation can be solved by minimizing the following mean-squared projected bellman error (MSPBE): $\text{MSPBE}(\boldsymbol{\theta}) \triangleq \frac{1}{2}\left\|\mathbb{E}_{\mathbf{s} \sim d^\pi}\left[(\mathcal{T}^\pi \mathcal{V}_{\boldsymbol{\theta}}(\mathbf{s}) - \mathcal{V}_{\boldsymbol{\theta}}) \nabla_{\boldsymbol{\theta}} \mathcal{V}_{\boldsymbol{\theta}}(\mathbf{s})^\top\right]\right\|^2_{\mathbf{K}_{\boldsymbol{\theta}}^{-1}}$, where $\mathbf{K}_{\boldsymbol{\theta}} = \mathbb{E}_{\mathbf{s} \sim d^\pi}[\nabla_{\boldsymbol{\theta}} \mathcal{V}_{\boldsymbol{\theta}}(\mathbf{s}) \nabla_{\boldsymbol{\theta}} \mathcal{V}_{\boldsymbol{\theta}}(\mathbf{s})^\top] \in \mathbb{R}^{p \times p}$ and $d^\pi$ is the stationary distribution of the MDP under policy $\pi$. From the Fenchel's duality $\|\mathbf{x}\|^2_{\mathbf{A}^{-1}} = \max_{\mathbf{y} \in \mathbb{R}^p} 2\langle \mathbf{x}, \mathbf{y}\rangle - \mathbf{y}^\top \mathbf{A} \mathbf{y}$, we can reformulate the MSPBE minimization problem as the following primal-dual minimax problem:

$$\min_{\boldsymbol{\theta} \in \mathbb{R}^p} \max_{\boldsymbol{\omega} \in \mathbb{R}^p} \mathcal{L}(\boldsymbol{\theta}, \boldsymbol{\omega}) \triangleq \mathbb{E}\left[\langle \boldsymbol{\delta} \cdot \nabla_{\boldsymbol{\theta}} \mathcal{V}_{\boldsymbol{\theta}}(\mathbf{s}), \boldsymbol{\omega}\rangle - \frac{1}{2} \boldsymbol{\omega}^\top [\nabla_{\boldsymbol{\theta}} \mathcal{V}_{\boldsymbol{\theta}}(\mathbf{s}) \nabla_{\boldsymbol{\theta}} \mathcal{V}_{\boldsymbol{\theta}}(\mathbf{s})^\top] \boldsymbol{\omega}\right], \tag{1}$$

where the expectation is taken over $\mathbf{s} \sim d^\pi(\cdot)$, $\mathbf{a} \sim \pi(\cdot|\mathbf{s})$, $\mathbf{s}' \sim \mathcal{P}^{\mathbf{a}}_{\mathbf{s} \cdot}$, and $\boldsymbol{\delta} = \frac{1}{m} \sum_{i=1}^m \mathcal{R}_i(\mathbf{s}, \mathbf{a}) + \zeta \mathcal{V}_{\boldsymbol{\theta}}(\mathbf{s}') - \mathcal{V}_{\boldsymbol{\theta}}(\mathbf{s})$. In practice, we only have access to a finite dataset with $n$-step trajectories $\mathcal{D} = \left\{(\mathbf{s}_t, \mathbf{a}_t, \{\mathcal{R}_i(\mathbf{s}_t, \mathbf{a}_t)\}_{i=1}^n, \mathbf{s}_{t+1})\right\}_{t=0}^n$. By replacing the unknown expectation with the finite sample average, we have the following empirical minimax problem:

$$\min_{\boldsymbol{\theta} \in \mathbb{R}^p} \max_{\boldsymbol{\omega} \in \mathbb{R}^p} F(\boldsymbol{\theta}, \boldsymbol{\omega}) = \frac{1}{n} \sum_{t=1}^n \langle \boldsymbol{\delta}_t \cdot \nabla_{\boldsymbol{\theta}} \mathcal{V}_{\boldsymbol{\theta}}(\mathbf{s}_t), \boldsymbol{\omega}\rangle - \frac{1}{2} \boldsymbol{\omega}^\top \widehat{\mathbf{K}}_{\boldsymbol{\theta}} \boldsymbol{\omega}, \tag{2}$$

where $\boldsymbol{\delta}_t \triangleq \frac{1}{m} \sum_{i=1}^m \mathcal{R}_i(\mathbf{s}_t, \mathbf{a}_t) + \zeta \mathcal{V}_{\boldsymbol{\theta}}(\mathbf{s}_{t+1}) - \mathcal{V}_{\boldsymbol{\theta}}(\mathbf{s}_t)$ and $\widehat{\mathbf{K}}_{\boldsymbol{\theta}} \triangleq \frac{1}{n} \sum_{t=1}^n \nabla_{\boldsymbol{\theta}} \mathcal{V}_{\boldsymbol{\theta}}(\mathbf{s}_t) \nabla_{\boldsymbol{\theta}} \mathcal{V}_{\boldsymbol{\theta}}(\mathbf{s}_t)^\top$. In this paper, we assume that both $\mathbf{K}_{\boldsymbol{\theta}}$ and its empirical estimate $\widehat{\mathbf{K}}_{\boldsymbol{\theta}}$ are positive definite for all $\boldsymbol{\theta}$. Define $J(\boldsymbol{\theta}) \triangleq F(\boldsymbol{\theta}, \boldsymbol{\omega}^*) = \max_{\boldsymbol{\omega} \in \mathbb{R}^p} F(\boldsymbol{\theta}, \boldsymbol{\omega})$, where $\boldsymbol{\omega}^* = \arg\max_{\boldsymbol{\omega} \in \mathbb{R}^p} F(\boldsymbol{\theta}, \boldsymbol{\omega})$. $J(\boldsymbol{\theta})$ can be viewed as the finite empirical version of MSPBE. Here, we aim to minimize $J(\boldsymbol{\theta})$ by finding a stationary point of $F(\boldsymbol{\theta}, \boldsymbol{\omega})$. Recall that in MARL, the local reward is only observable for each individual agent. Thus, it is hard to obtain the global reward $\frac{1}{m} \sum_{i=1}^m \mathcal{R}_i(\mathbf{s}_t, \mathbf{a}_t)$ and $\boldsymbol{\delta}_t$ in a multi-agent network. To address this challenge, we define $\boldsymbol{\delta}_{i,t} = \mathcal{R}_i(\mathbf{s}_t, \mathbf{a}_t) + \zeta \mathcal{V}_{\boldsymbol{\theta}}(\mathbf{s}_{t+1}) - \mathcal{V}_{\boldsymbol{\theta}}(\mathbf{s}_t)$ and decompose the minimax problem in (2) as follows: $\min_{\boldsymbol{\theta} \in \mathbb{R}^p} \max_{\boldsymbol{\omega} \in \mathbb{R}^p} F(\boldsymbol{\theta}, \boldsymbol{\omega}) = \frac{1}{m} \sum_{i=1}^m F_i(\boldsymbol{\theta}, \boldsymbol{\omega}) = \frac{1}{mn} \sum_{i=1}^m \sum_{t=1}^n f_{it}(\boldsymbol{\theta}, \boldsymbol{\omega})$, where $f_{it}(\boldsymbol{\theta}, \boldsymbol{\omega}) \triangleq \langle \boldsymbol{\delta}_{i,t} \cdot \nabla_{\boldsymbol{\theta}} \mathcal{V}_{\boldsymbol{\theta}}(\mathbf{s}_t), \boldsymbol{\omega}\rangle - \frac{1}{2} \boldsymbol{\omega}^\top [\nabla_{\boldsymbol{\theta}} \mathcal{V}_{\boldsymbol{\theta}}(\mathbf{s}_t) \nabla_{\boldsymbol{\theta}} \mathcal{V}_{\boldsymbol{\theta}}(\mathbf{s}_t)^\top] \boldsymbol{\omega}$. We call this step as local reward decomposition. In cooperative MARL, a key challenge is that the PE problem in (2) has to be solved in a *decentralized* fashion, which is due to the fact that i) the locally observed rewards are private and cannot be shared with the other agents/central server; ii) it is difficult to set up a central sever in many MARL applications while decentralized setting is more flexible (e.g.,wireless network [59], UAV network [7]); and iii) the central server is vulnerable to cyber-attacks and would be a significant communication bottleneck [57], [27]. To solve Problem (2)

in a decentralized fashion, we can rewrite it in the following equivalent form:

$$\min_{\{\boldsymbol{\theta}_i\}_{i=1}^m} \max_{\{\boldsymbol{\omega}_i\}_{i=1}^m} \frac{1}{m} \sum_{i=1}^m F_i(\boldsymbol{\theta}_i, \boldsymbol{\omega}_i) = \frac{1}{mn} \sum_{i=1}^m \sum_{t=1}^n f_{it}(\boldsymbol{\theta}_i, \boldsymbol{\omega}_i),$$

$$\text{subject to} \quad \boldsymbol{\theta}_i = \boldsymbol{\theta}_j, \boldsymbol{\omega}_i = \boldsymbol{\omega}_j, \ \ \forall (i,j) \in \mathcal{L}, \tag{3}$$

where $\boldsymbol{\theta}_i$ and $\boldsymbol{\omega}_i$ are the local copies of the original primal-dual parameters at agent $i$. In (3), the equality constraint ensures that the local copies of all nodes are equal to each other, so the formulation is also referred to as the "consensus form."

Clearly, Problems (2) and (3) are equivalent. For a fixed $\boldsymbol{\theta}$, each local function $F_i(\boldsymbol{\theta}_i, \cdot)$ is a strongly concave function of $\boldsymbol{\omega}$. For a fixed $\boldsymbol{\omega}$, $F_i(\cdot, \boldsymbol{\omega})$ is a non-convex function of $\boldsymbol{\theta}$. Thus, Problem (3) is a decentralized non-convex-strongly-concave minimax consensus optimization problem. In this paper, we adopt two complexity metrics that are widely used in the decentralized optimization literature (e.g., [47]) to measure the efficiency of an algorithm:

**Definition 1** (Sample Complexity). *The sample complexity is defined as the total number of incremental first-order oracle (IFO) calls required across all nodes until algorithm converges, where one IFO call evaluates a pair of $(f_{it}(\boldsymbol{\theta}, \boldsymbol{\omega}), \nabla f_{it}(\boldsymbol{\theta}, \boldsymbol{\omega}))$ at node $i$.*

**Definition 2** (Communication Complexity). *The communication complexity is defined as the total rounds of communications required until algorithm converges, where each node can send and receive a $p$-dimensional vector with its neighboring nodes in one communication round.*

## 2.2   Related work on policy evaluation

**1) The tabular approach:** The study of MARL under the MDP formulation traces its roots to the seminal work by [29]. Motivated by this formalization, several methods have been developed to solve and analyze MARL problems, including [24, 54, 17, 1], etc. However, most of these early works approximate the value function in a tabular form, which only works for cases where the state and action spaces are relatively small. For complex MARL tasks where the state space is large or even infinite, the tabular approach becomes intractable.

**2) Policy evaluation with linear function approximation:** To address limitation in tabular approaches for MARL, the work in [25] proposed to estimate the value function with a *linear approximation* (i.e., $\mathcal{V}(\mathbf{s}) \approx \phi(\mathbf{s})^\top \boldsymbol{\theta}, \boldsymbol{\theta} \in \mathbb{R}^p$, where $\phi : \mathcal{S} \to \mathbb{R}^p$ is a feature mapping) and developed a distributed gradient temporal-difference (DGTD) algorithm. However, this work only considered asymptotic convergence analysis and required diminishing step-sizes to ensure convergence. In [12], the authors proposed a distributed homotopy primal-dual algorithm (DHPD) for the PE problem in MARL. They also cast MSPBE minimization as a stochastic primal-dual optimization problem, where the objective is convex in primal and strongly-concave in dual. By using an adaptive restarting scheme, DHPD achieves an $\mathcal{O}(1/T)$ convergence rate in finding stationary points. The work in [13] developed a distributed consensus-based TD(0) algorithm, which integrates the network consensus step and local TD(0) updates. They provided a finite-time analysis and showed a convergence rate of $\mathcal{O}(1/T)$. To further improve convergence, the work in [51] proposed a primal-dual distributed incremental aggregated gradient (PD-DistIAG) method to integrate gradient-tracking and incremental aggregated gradient methods to achieve linear convergence. However, a major limitation of the linear approximation approach is that it is not applicable for nonlinear MARL models (e.g., DNN).

**3) Policy evaluation with nonlinear function approximation (single-agent):** In the literature, PE with nonlinear approximation is by far only limited to *single-agent* RL. For policy evaluation, linear and nonlinear approximation approaches differ fundamentally in the following aspects. Under linear approximation for policy evaluation, the problem boils down to finding a solution for a linear equation system, which is in essence similar to solving a relatively easy strongly-convex optimization problem [14], [52], [48], [46]. In stark contrast, under non-linear approximation for policy evaluation, the problem possesses a non-convex-strongly-concave structure, and it is far more challenging to find a saddle point solution. To our knowledge, the work in [4] was the first to study the PE problem with nonlinear approximations and developed a nonlinear TD algorithm. However, the proposed algorithms adopted two-timescale step-sizes[2], resulting in a slow convergence performance. Recently, the work

---

[2]A primal-dual algorithm is a two-timescale if $\gamma_t/\eta_t \to 0$ or $\gamma_t/\eta_t \to \infty$ as $t \to \infty$, where $\gamma_t$ and $\eta_t$ denote the primal and dual step-sizes at time $t$, respectively.

Table 1: Comparisons among existing policy evaluation algorithms, where $m$ is the number of agents; $n$ is the size of dataset; $\epsilon^2$ is the convergence error. Our proposed algorithms are marked in bold.

| Algorithm | Reference | Decentralized Multi-Agent | Nonlinear Approxi. | Convex Sets[1] | Sample Complex. | Commun. Complex. |
|---|---|---|---|---|---|---|
| STSG | [42] | ✗ | ✓ | ✓ | $\mathcal{O}(\epsilon^{-4})$ | - |
| ASTSG | | ✗ | ✓ | ✓ | $\mathcal{O}(\epsilon^{-3})$ | - |
| DHPD | [12] | ✓ | ✗ | ✓ | $\mathcal{O}(mn\epsilon^{-2})$ | $\mathcal{O}(\epsilon^{-2})$ |
| PD-DistIAG | [51] | ✓ | ✗ | ✗ | $\mathcal{O}(m\log\epsilon^{-2})$ | $\mathcal{O}(\log\epsilon^{-2})$ |
| APP-SAG | [44] | ✓ | ✗ | ✗ | $\mathcal{O}(m\log\epsilon^{-2})$ | $\mathcal{O}(\log\epsilon^{-2})$ |
| DTDT | [53] | ✓ | ✗ | ✗ | $\mathcal{O}(m\log\epsilon^{-2})$ | $\mathcal{O}(\log\epsilon^{-2})$ |
| **GT-GDA** | Theorem 1 | ✓ | ✓ | ✗ | $\mathcal{O}(mn\epsilon^{-2})$ | $\mathcal{O}(\epsilon^{-2})$ |
| **GT-SRVR** | Theorem 3 | ✓ | ✓ | ✗ | $\mathcal{O}(m\sqrt{n}\epsilon^{-2})$ | $\mathcal{O}(\epsilon^{-2})$ |
| **GT-SRVRI** | Theorem 4 | | | | | |

[1] The feasible parameter spaces are required to be closed convex sets.

in [50] showed that PE with nonlinear approximation in RL is equivalent to a non-convex-strongly-concave minimax optimization problem. To find a stationary point for such minimax problem, they proposed a non-convex primal-dual gradient with variance reduced (nPD-VR) algorithm. However, their algorithm requires an $\mathcal{O}(1/m)$ step-size, where $m$ is the size of the dataset. This is problematic in cases with a large transition dataset. More recently, the authors of [42] proposed two single-time scale first-order stochastic algorithms for the nonconvex-strongly-concave minimax optimization. These two algorithms utilized stochastic gradient with momentum and variance-reduced momentum [8], and achieved $\mathcal{O}(1/\sqrt{T})$ and $\mathcal{O}(1/T^{2/3})$ convergence rates, respectively. However, diminishing step-sizes are required in these two algorithms, which do not work well in practice.

**4) Relations with decentralized nonconvex-strongly-concave minimax optimization:** As mentioned earlier, the PE problem of cooperative MARL can be reformulated as a non-convex-strongly-concave optimization (NCSC) problem (see details in Problem (2). Thus, our work is also closely related to the area of decentralized NCSC minimax optimization. To efficiently solve the decentralized minimax problem in Problem (2), our proposed algorithms are primal-dual-based algorithms, where we update the two variables simultaneously rather than alternatively. Thus, our proposed algorithms are much simpler and significantly different from existing related works that require to solve maximization subproblem for dual variable in each iteration [36]. Further, we propose a "hybrid" scheme that non-trivially integrates variance reduction and gradient tracking techniques for both primal and dual variables. Compared with algorithms in [31],[32], with simple stochastic gradient updates and variable mixing, our proposed scheme enjoys much improved theoretical and numerical performances. Also, we note that the theoretical analysis of the more sophisticated hybrid scheme in the PE problem of cooperative MARL is more involved compared to existing works and necessitates new proof techniques.

## 2.3 Significance of our work and challenges of DPE with nonlinear function approximation

To our knowledge, our work is the *first* to solve the DPE problem with *nonlinear* function approximation for MARL. However, such nonlinear approximation imposes several significant challenges on algorithm development and analysis. As discussed in Section 2.1, we propose to reformulate the DPE problem as a *decentralized* nonconvex-strongly-concave minimax optimization problem. In the literature, although a few works [32, 31, 34] have studied similar decentralized minimax problem, their variable updates rely on either stochastic or full gradients, which are not sample/communication-efficient for MARL. To address these limitations, we first propose two decentralized variance-reduced algorithms for solving the nonconvex-strongly-concave DPE minimax problem, for which establishing the convergence rates is highly challenging. Second, we adopt the gradient tracking (GT) technique to reduce network consensus error. Under nonlinear function approximation, our algorithms need to track the gradients for both primal and dual variables. However, such "double gradient tracking" introduces new constraints on algorithm design. Third, many of the existing nonconvex-strongly-concave minimax optimization methods (e.g. [42, 50]) require prior knowledge of the compact domain of model variables, which cannot be assumed in DPE for MARL. Such unboundedness in DPE creates new challenges and necessitates new proof techniques in our algorithm analysis. To conclude this section, we summarize all related work in Table 1.

**Algorithm 1** GT-GDA Algorithm at Agent $i$.

1: Set prime-dual parameter pair $(\boldsymbol{\theta}_{i,0}, \boldsymbol{\omega}_{i,0}) = (\boldsymbol{\theta}^0, \boldsymbol{\omega}^0)$.
2: Calculate local gradients as $\mathbf{p}_{i,0} = \nabla_{\boldsymbol{\theta}} F_i(\boldsymbol{\theta}_{i,0}, \boldsymbol{\omega}_{i,0})$, and $\mathbf{d}_{i,0} = \nabla_{\boldsymbol{\omega}} F_i(\boldsymbol{\theta}_{i,0}, \boldsymbol{\omega}_{i,0})$;
3: **for** $t = 1, \cdots, T$ **do**
4:   Update local parameters $(\boldsymbol{\theta}_{i,t+1}, \boldsymbol{\omega}_{i,t+1})$ as in (6);
5:   Calculate local gradients $(\mathbf{v}_{i,t+1}, \mathbf{u}_{i,t+1})$ as in (4);
6:   Track global gradients $(\mathbf{p}_{i,t+1}, \mathbf{d}_{i,t+1})$ as in (5);
7: **end for**

## 3 Gradient-tracking gradient descent ascent algorithm.

In this section, we first present a gradient-tracking gradient descent ascent (GT-GDA) method for solving the DPE problem in (3) for MARL, and then provide the its theoretical results.

**1) The Algorithm:** For the consensus problem in (3), a popular approach is to let agents aggregate their neighbor information through a consensus weight matrix $\mathbf{M} \in \mathbb{R}^{m \times m}$ [39, 51]. Let $[\mathbf{M}]_{ij}$ denote the element in the $i$-th row and the $j$-th column in $\mathbf{M}$. $\mathbf{M}$ satisfies the following properties: (a) *Doubly stochastic:* $\sum_{i=1}^{m}[\mathbf{M}]_{ij} = \sum_{j=1}^{m}[\mathbf{M}]_{ij} = 1$; (b) *Symmetric:* $[\mathbf{M}]_{ij} = [\mathbf{M}]_{ji}, \forall i, j \in \mathcal{N}$; and (c) *Network-Defined Sparsity:* $[\mathbf{M}]_{ij} > 0$ if $(i,j) \in \mathcal{L}$; otherwise $[\mathbf{M}]_{ij} = 0, \forall i, j \in \mathcal{N}$. The above properties imply that the eigenvalues of $\mathbf{M}$ are real and can be sorted as $-1 < \lambda_m(\mathbf{M}) \leq \cdots \leq \lambda_2(\mathbf{M}) < \lambda_1(\mathbf{M}) = 1$. We define the second-largest eigenvalue in magnitude of $\mathbf{M}$ as $\lambda \triangleq \max\{|\lambda_2(\mathbf{M})|, |\lambda_m(\mathbf{M})|\}$. Our GT-GDA algorithm for each agent $i$ is illustrated in Algorithm 1. Specifically, at the $t$-th iteration, agent $i$ first calculates the local full gradients as follows:

$$\mathbf{v}_{i,t} = \nabla_{\boldsymbol{\theta}} F_i(\boldsymbol{\theta}_{i,t}, \boldsymbol{\omega}_{i,t}), \quad \mathbf{u}_{i,t} = \nabla_{\boldsymbol{\omega}} F_i(\boldsymbol{\theta}_{i,t}, \boldsymbol{\omega}_{i,t}). \tag{4}$$

Note that $\mathbf{v}_{i,t}$ and $\mathbf{u}_{i,t}$ only contain the gradient information of the local objective function $F_i(\boldsymbol{\theta}, \boldsymbol{\omega})$. Thus, merely updating with $\mathbf{v}_{i,t}$ and $\mathbf{u}_{i,t}$ cannot guarantee the convergence of the global objective function $F(\boldsymbol{\theta}, \boldsymbol{\omega})$. To address this challenge, we introduce two auxiliary variables, $\mathbf{p}_{i,t}$ and $\mathbf{d}_{i,t}$. The agent updates the two variables by performing the following local weighted aggregation:

$$\mathbf{p}_{i,t} = \sum_{j \in \mathcal{N}_i} [\mathbf{M}]_{ij} \mathbf{p}_{j,t-1} + \mathbf{v}_{i,t} - \mathbf{v}_{i,t-1}, \quad \mathbf{d}_{i,t} = \sum_{j \in \mathcal{N}_i} [\mathbf{M}]_{ij} \mathbf{d}_{j,t-1} + \mathbf{u}_{i,t} - \mathbf{u}_{i,t-1}, \tag{5}$$

where $\mathcal{N}_i \triangleq \{j \in \mathcal{N}, : (i,j) \in \mathcal{L}\}$ denotes the set of agent $i$'s neighbors. Technically, $\mathbf{p}_{i,t}$ and $\mathbf{d}_{i,t}$ track the directions of global gradients. With some derivations, it can be shown that $\sum_{i \in \mathcal{N}} \mathbf{p}_{i,t} = \sum_{i \in \mathcal{N}} \nabla_{\boldsymbol{\theta}} F_i(\boldsymbol{\theta}_{i,t}, \boldsymbol{\omega}_{i,t})$ and $\sum_{i \in \mathcal{N}} \mathbf{d}_{i,t} = \sum_{i \in \mathcal{N}} \nabla_{\boldsymbol{\omega}} F_i(\boldsymbol{\theta}_{i,t}, \boldsymbol{\omega}_{i,t})$. Lastly, each agent updates local parameters following the conventional decentralized gradient descent and ascent steps:

$$\boldsymbol{\theta}_{i,t+1} = \sum_{j \in \mathcal{N}_i} [\mathbf{M}]_{ij} \boldsymbol{\theta}_{j,t} - \gamma \mathbf{p}_{i,t}, \quad \boldsymbol{\omega}_{i,t+1} = \sum_{j \in \mathcal{N}_i} [\mathbf{M}]_{ij} \boldsymbol{\omega}_{j,t} + \eta \mathbf{d}_{i,t}, \tag{6}$$

where the constants $\gamma$ and $\eta$ are the step-sizes for individual primal and dual variables, respectively.

**2) Theoretical Results of GT-GDA:** In this section, we will establish the convergence behaviors of the proposed GT-GDA algorithm. Toward this end, we first state several assumptions as follows:

**Assumption 1.** *The function $F(\boldsymbol{\theta}, \boldsymbol{\omega}) = \frac{1}{m} \sum_{i=1}^{m} F_i(\boldsymbol{\theta}, \boldsymbol{\omega})$ and $J(\boldsymbol{\theta}) = \max_{\boldsymbol{\omega} \in \mathbb{R}^p} F(\boldsymbol{\theta}, \boldsymbol{\omega})$ satisfy:*

*(a) (Boundness from Below): There exists a finite lower bound $J^* = \inf_{\boldsymbol{\theta}} J(\boldsymbol{\theta}) > -\infty$;*

*(b) (Lipschitz Smoothness): Local objective function $F_i(\boldsymbol{\theta}, \cdot)$ is $L_F$-Lipschitz smooth, i.e., there exists a positive constant $L_F$ such that the gradient $\nabla F_i(\boldsymbol{\theta}, \boldsymbol{\omega}) = [\nabla_{\boldsymbol{\theta}} F_i(\boldsymbol{\theta}, \boldsymbol{\omega})^\top, \nabla_{\boldsymbol{\omega}} F_i(\boldsymbol{\theta}, \boldsymbol{\omega})^\top]^\top$ satisfies $\|\nabla F_i(\boldsymbol{\theta}, \boldsymbol{\omega}) - \nabla F_i(\boldsymbol{\theta}', \boldsymbol{\omega}')\|^2 \leq L_F^2 \|\boldsymbol{\theta} - \boldsymbol{\theta}'\|^2 + L_F^2 \|\boldsymbol{\omega} - \boldsymbol{\omega}'\|^2, \forall \boldsymbol{\theta}, \boldsymbol{\theta}', \boldsymbol{\omega}, \boldsymbol{\omega}' \in \mathbb{R}^p, i \in [m]$;*

*(c) (Strong Concavity in Dual): Local objective function $F_i(\boldsymbol{\theta}, \cdot)$ is $\mu$-strongly concave for fixed $\boldsymbol{\theta} \in \mathbb{R}^p$, i.e., there exists a positive constant $\mu$ such that $\|\nabla_{\boldsymbol{\omega}} F_i(\boldsymbol{\theta}, \boldsymbol{\omega}) - \nabla_{\boldsymbol{\omega}} F_i(\boldsymbol{\theta}, \boldsymbol{\omega}')\| \geq \mu \|\boldsymbol{\omega} - \boldsymbol{\omega}'\|, \forall \boldsymbol{\theta}, \boldsymbol{\omega}, \boldsymbol{\omega}' \in \mathbb{R}^p, i \in [m]$*

*(d) (Bounded Dual Maximizer): For any primal variable $\boldsymbol{\theta} \in \mathbb{R}^p$, its associated dual maximizer $\boldsymbol{\omega}^*(\boldsymbol{\theta}) \triangleq \arg\max_{\boldsymbol{\omega} \in \mathbb{R}^p} F(\boldsymbol{\theta}, \boldsymbol{\omega})$ is bounded, i.e., $\|\boldsymbol{\omega}^*(\boldsymbol{\theta})\| < \infty$;*

*(e) (Bounded Gradient at Maximum): The partial derivative at maximum point $\nabla_{\boldsymbol{\theta}} F(\boldsymbol{\theta}, \boldsymbol{\omega}^*(\boldsymbol{\theta}))$ is bounded, i.e., $\|\nabla_{\boldsymbol{\theta}} F(\boldsymbol{\theta}, \boldsymbol{\omega}^*(\boldsymbol{\theta}))\| < \infty, \forall \boldsymbol{\theta} \in \mathbb{R}^p$.*

In these assumptions, (a) and (b) are standard in literature. It can be verified that (c) holds when $\widehat{\mathbf{K}}_{\boldsymbol{\theta}}$ is positive definite; (d)-(e) guarantee that $\nabla J(\boldsymbol{\theta}) = \nabla_{\boldsymbol{\theta}} F(\boldsymbol{\theta}, \boldsymbol{\omega}^*(\boldsymbol{\theta}))$ (see Lemma 10 in the supplementary material). Note that most of the existing works [28, 42] adopt the compactness assumption to ensure such gradient equivalence. The compactness assumption restricts the feasible parameter space as a closed convex set. Although we also make these boundedness assumptions, the convergence performance of our algorithm is *independent* of the upper bound of $\|\boldsymbol{\omega}^*(\boldsymbol{\theta})\|$ and $\|\nabla_{\boldsymbol{\theta}} F(\boldsymbol{\theta}, \boldsymbol{\omega}^*(\boldsymbol{\theta}))\|$. To quantify the convergence rate, we propose to use the following *new* metric, which is the key to the success of establishing all convergence results in this paper:

$$\mathfrak{M}_t \triangleq \|\nabla J(\bar{\boldsymbol{\theta}}_t)\|^2 + 2\|\boldsymbol{\omega}_t^* - \bar{\boldsymbol{\omega}}_t\|^2 + \frac{1}{m}\sum_{i=1}^m \left(\|\boldsymbol{\theta}_{i,t} - \bar{\boldsymbol{\theta}}_t\|^2 + \|\boldsymbol{\omega}_{i,t} - \bar{\boldsymbol{\omega}}_t\|^2\right), \tag{7}$$

where $\boldsymbol{\omega}_t^*$ denotes $\boldsymbol{\omega}^*(\bar{\boldsymbol{\theta}}_t) = \arg\max_{\boldsymbol{\omega} \in \mathbb{R}^p} F(\bar{\boldsymbol{\theta}}_t, \boldsymbol{\omega})$. The first term in (7) measures the convergence of primal variable $\boldsymbol{\theta}$: $\|\nabla J(\bar{\boldsymbol{\theta}}_t)\|^2 = 0$ indicates that $\bar{\boldsymbol{\theta}}_t$ is a first-order stationary point for $J(\cdot)$. The second term in (7) measures $\bar{\boldsymbol{\omega}}_t$'s convergence to the unique maximizer $\boldsymbol{\omega}_t^*$ for $F(\bar{\boldsymbol{\theta}}_t, \cdot)$. The last term in (7) is the average consensus error of local copies. Thus, as $\mathfrak{M}_t \to 0$, we have that the algorithm reaches a consensus first-order stationary point of the original MSPBE problem. In comparison, the single-agent PE problem [42] does not have the last four consensus error terms over multi-agents, and so it is dramatically different from our DPE problem. Also, for the linear approximation PE problem in [51], the first term is replaced with $\|\bar{\theta}_t - \theta^*\|^2$ as it can be viewed as a strongly convex optimization, while the other terms are the same. Based on the metric in (7), we have the following:

**Theorem 1** (Convergence of GT-GDA). *Under Assumption 1, if the step-sizes satisfy that $\kappa \triangleq \gamma/\eta \leq \mu^2/13L_F^2$ and $\eta \leq \min\{k_1, k_2, k_3, k_4\}$, then GT-GDA has the following convergence result:*

$$\frac{1}{T+1}\sum_{t=0}^T \mathbb{E}[\mathfrak{M}_t] \leq \frac{2\mathbb{E}[\mathfrak{P}_0 - J^*]}{\min\{1, L_F^2\}(T+1)\gamma},$$

*where $\mathfrak{P}_t$ is a potential function defined as:*

$$\mathfrak{P}_t \triangleq J(\bar{\boldsymbol{\theta}}_t) + \frac{8\gamma L_F^2}{\mu\eta}\|\bar{\boldsymbol{\omega}}_t - \boldsymbol{\omega}_t^*\|^2 + \frac{1}{m}\sum_{i=1}^m \|\boldsymbol{\theta}_{i,t} - \bar{\boldsymbol{\theta}}_t\|^2 + \|\boldsymbol{\omega}_{i,t} - \bar{\boldsymbol{\omega}}_t\|^2 + \gamma\|\mathbf{p}_{i,t} - \bar{\mathbf{p}}_t\|^2 + \eta\|\mathbf{d}_{i,t} - \bar{\mathbf{d}}_t\|^2,$$

*and the constants in the step-size $\eta$ are as follows:*

$$k_1 = \frac{13L_F^2}{2\mu^2}\left(L_F + \frac{L_F^2}{\mu} + (1-\lambda)\right), \quad k_2 = \frac{13L_F^2}{\mu^2\left(1/2 + 1/(1-\lambda)^2\right)}, \quad k_3 = \frac{(1-\lambda)}{6\mu(1+1/\kappa)},$$

$$k_4 = \frac{26(1-\lambda)L_F^2}{\left(\mu^2 + 144L_F^4 + 4L_F^2\mu^2 + \frac{48\mu^2 L_F^2(1+1/\kappa)}{1-\lambda}\right)}.$$

**Remark 1.** In Theorem 1, the step-sizes and convergence rate depend on by the network topology. For a sparse network, $\lambda$ is close to (but not exactly) one (recall that $\lambda = \max\{|\lambda_2|, |\lambda_m|\} < 1$). Therefore, $k_2$ and $k_4$ are close to zero in this case. Also, the ratio of the step-sizes $\kappa \triangleq \gamma/\eta$ is required to be a non-zero constant. Either a too small or a too large value of $\kappa$ might affect the primal or dual convergence of the algorithm. This restriction is due to the consensus error in the decentralized training. In practice, one can first determine $\kappa$ and then select $\eta$ and $\gamma$.

From Theorem 1, we immediately have the following complexity results for GT-GDA:

**Corollary 2.** *Under the same conditions in Theorem 1, to achieve an $\epsilon^2$-stationary solution, i.e., $\frac{1}{T+1}\sum_{t=0}^T \mathbb{E}[\mathfrak{M}_t] \leq \epsilon^2$, the total communication rounds are on the order of $\mathcal{O}(\epsilon^{-2})$ and the total samples evaluated across the network system is on the order of $\mathcal{O}(mn\epsilon^{-2})$.*

## 4 Gradient-tracking stochastic variance reduction algorithms

In the GT-GDA algorithm, agents need to evaluate local full gradients in each iteration, which may result in a high sample complexity when the trajectory length $n$ is large in MARL. This limitation motivates us to leverage the stochastic recursive variance-reduced approach (e.g., [55]) to achieve low sample complexity in MARL decentralized policy evaluation.

**Algorithm 2** GT-SRVR/GT-SRVRI Algorithm at Agent $i$.

---

- If GT-SRVR: $|\mathcal{R}_{i,t}| = n$, $|\mathcal{S}_{i,t}| = q$; If GT-SRVRI: $|\mathcal{R}_{i,t}| = \min\{(t/q+1)^{\alpha}q, c_{\epsilon}\epsilon^{-2}, n\}$, $|\mathcal{S}_{i,t}| = q$.
1: Set prime-dual parameter pair $(\boldsymbol{\theta}_{i,0}, \boldsymbol{\omega}_{i,0}) = (\boldsymbol{\theta}^0, \boldsymbol{\omega}^0)$.
2: Draw $\mathcal{R}_{i,0}$ samples without replacement and calculate local stochastic gradient estimators as
$\mathbf{p}_{i,0} = \mathbf{v}_{i,0} = \frac{1}{|\mathcal{R}_{i,0}|}\sum_{j \in \mathcal{R}_{i,0}} \nabla_{\boldsymbol{\theta}} f_{ij}(\boldsymbol{\theta}_{i,0}, \boldsymbol{\omega}_{i,0})$, and $\mathbf{d}_{i,0} = \mathbf{u}_{i,0} = \frac{1}{|\mathcal{R}_{i,0}|}\sum_{j \in \mathcal{R}_{i,0}} \nabla_{\boldsymbol{\omega}} f_{ij}(\boldsymbol{\theta}_{i,0}, \boldsymbol{\omega}_{i,0})$;
3: **for** $t = 1, \cdots, T$ **do**
4:    Update local parameters $(\boldsymbol{\theta}_{i,t+1}, \boldsymbol{\omega}_{i,t+1})$ as in (6);
5:    Calculate local gradient estimators $(\mathbf{v}_{i,t+1}, \mathbf{u}_{i,t+1})$ as in (8);
6:    Track global gradients $(\mathbf{p}_{i,t+1}, \mathbf{d}_{i,t+1})$ as in (5);
7: **end for**

---

**1) The Algorithms:** We first propose an algorithm called gradient-tracking stochastic recursive variance reduction (GT-SRVR) algorithm. Different from GT-GDA, in iteration $t$ and at agent $i$, GT-SRVR estimates the local gradient with the following estimators:

$$\mathbf{v}_{i,t} = \begin{cases} \nabla_{\boldsymbol{\theta}} F_i(\boldsymbol{\theta}_{i,t}, \boldsymbol{\omega}_{i,t}), & \text{if } \mod(t,q) = 0, \\ \mathbf{v}_{i,t-1} + \frac{1}{|\mathcal{S}_{i,t}|}\sum_{j \in \mathcal{S}_{i,t}}\left(\nabla_{\boldsymbol{\theta}} f_{ij}(\boldsymbol{\theta}_{i,t}, \boldsymbol{\omega}_{i,t}) - \nabla_{\boldsymbol{\theta}} f_{ij}(\boldsymbol{\theta}_{i,t-1}, \boldsymbol{\omega}_{i,t-1})\right), & \text{otherwise,} \end{cases} \tag{8a}$$

$$\mathbf{u}_{i,t} = \begin{cases} \nabla_{\boldsymbol{\omega}} F_i(\boldsymbol{\theta}_{i,t}, \boldsymbol{\omega}_{i,t}), & \text{if } \mod(t,q) = 0, \\ \mathbf{u}_{i,t-1} + \frac{1}{|\mathcal{S}_{i,t}|}\sum_{j \in \mathcal{S}_{i,t}}\left(\nabla_{\boldsymbol{\omega}} f_{ij}(\boldsymbol{\theta}_{i,t}, \boldsymbol{\omega}_{i,t}) - \nabla_{\boldsymbol{\omega}} f_{ij}(\boldsymbol{\theta}_{i,t-1}, \boldsymbol{\omega}_{i,t-1})\right), & \text{otherwise,} \end{cases} \tag{8b}$$

where $\mathcal{S}_{i,t}$ is a local subsample at the $t$th iteration for agent $i$. In (8), the algorithm evaluates full gradients $\nabla F_i(\boldsymbol{\theta}_{i,t}, \boldsymbol{\omega}_{i,t})$ only every $q$ steps. For other iterations with $\mod(t,q) \neq 0$, the algorithm estimates the local gradients with a mini-batch of gradients $\frac{1}{|\mathcal{S}_{i,t}|}\sum_{j \in \mathcal{S}_{i,t}}\nabla_{\boldsymbol{\omega}} f_{ij}(\boldsymbol{\theta}_{i,t}, \boldsymbol{\omega}_{i,t})$ and a recursive correction term $\mathbf{u}_{i,t-1} - \frac{1}{|\mathcal{S}_{i,t}|}\sum_{j \in \mathcal{S}_{i,t}}\nabla_{\boldsymbol{\omega}} f_{ij}(\boldsymbol{\theta}_{i,t-1}, \boldsymbol{\omega}_{i,t-1})$. It will be shown later that, thanks to the periodic full gradient (when $\mod(t,q) = 0$) and recursive correction term, GT-SRVR is able to achieve the *same* convergence rate and communication complexity as GT-GDA. Moreover, because of the $\mathcal{S}_{i,t}$ subsampling, GT-SRVR has a *lower* sample complexity than GT-GDA. The full description of GT-SRVR is shown in Algorithm 2.

Note that in GT-SRVR, full gradients are still required for every $q$ steps, which may still incur a high computational cost. Also, in the initialization phase (before the main loop), agents need to evaluate full gradients, which could be time-consuming. To address these limitations, we propose an enhanced version of GT-SRVR called GT-SRVR with Incremental batch size (GT-SRVRI). Specifically, we modify the gradient estimators in (8a) and (8b) for the $t$th iteration with $\mod(t,q) = 0$ as follows :

$$\mathbf{v}_{i,t} = \frac{1}{|\mathcal{R}_{i,t}|}\sum_{j \in \mathcal{R}_{i,t}}\nabla_{\boldsymbol{\theta}} f_{ij}(\boldsymbol{\theta}_{i,t}, \boldsymbol{\omega}_{i,t}), \quad \mathbf{u}_{i,t} = \frac{1}{|\mathcal{R}_{i,t}|}\sum_{j \in \mathcal{R}_{i,t}}\nabla_{\boldsymbol{\omega}} f_{ij}(\boldsymbol{\theta}_{i,t}, \boldsymbol{\omega}_{i,t}), \tag{9}$$

where $\mathcal{R}_{i,t}$ is a subsample set (sampling without replacement), whose size is chosen as $|\mathcal{R}_{i,t}| = \min\{(t/q+1)^{\alpha}q, c_{\epsilon}\epsilon^{-2}, n\}$. Here, $\alpha > 0$ is a constant, $\epsilon$ is a desired convergence error, and $c_{\epsilon} > 0$ is a constant that depends on $\epsilon$. Our design of $|\mathcal{R}_{i,t}|$ is motivated by the fact that the periodic full gradient evaluation only plays an important role in the later stage of the convergence process for achieving high accuracy. Later, we will see that under some mild assumptions and parameter settings, GT-SRVRI has similar convergence performance as GT-SRVR. The full description of GT-SRVR/GT-SRVRI is shown in Algorithm 2.

**Remark 2.** In GT-SRVRI, we increase the batch-size as the number of iterations increases. We note that the work in [18] also proposed a batch-size adaptation scheme based on the historical gradient information. Although similar idea can be also adopted in our algorithms, it requires the exact value of the stochastic gradient variance $\sigma^2$ for batch-size selection as well as extra memory cost to store the history-gradient information, which is less practical compared to our approach.

**2) Theoretical Results of GT-SRVR/GT-SRVRI:** Now, we establish the convergence performance of GT-SRVR/GT-SRVRI. First, we replace Assumption 1(b) with the following individual Lipschitz smoothness assumption:

**Assumption 2** (Lipschitz smoothness). *The function $f_{ij}(\boldsymbol{\theta}, \cdot)$ is $L_f$-Lipschitz smooth, i.e., there exists a constant $L_f > 0$, such that the gradient $\nabla f_{ij}(\boldsymbol{\theta}, \boldsymbol{\omega}) = [\nabla_{\boldsymbol{\theta}} f_{ij}(\boldsymbol{\theta}, \boldsymbol{\omega})^{\top}, \nabla_{\boldsymbol{\omega}} f_{ij}(\boldsymbol{\theta}, \boldsymbol{\omega})^{\top})^{\top}$ satisfies $\|\nabla f_{ij}(\boldsymbol{\theta}, \boldsymbol{\omega}) - \nabla f_{ij}(\boldsymbol{\theta}', \boldsymbol{\omega}')\|^2 \leq L_f^2\|\boldsymbol{\theta} - \boldsymbol{\theta}'\|^2 + L_f^2\|\boldsymbol{\omega} - \boldsymbol{\omega}'\|^2, \ \forall \boldsymbol{\theta}, \boldsymbol{\theta}', \boldsymbol{\omega}, \boldsymbol{\omega}' \in \mathbb{R}^p, i \in [m], j \in [n]$.*

We note that Assumption 2 is a common assumption for stochastic variance reduced methods [55, 47, 42]. Further, we make the following assumption only for GT-SRVRI algorithm:

**Assumption 3** (Bounded Variance). *There exists a constant $\sigma^2 > 0$, such that $\mathbb{E}\|\nabla f_{ij}(\boldsymbol{\theta}, \boldsymbol{\omega}) - \nabla F_i(\boldsymbol{\theta}, \boldsymbol{\omega})\|^2 \leq \sigma^2, \forall \boldsymbol{\theta}, \boldsymbol{\omega}, \in \mathbb{R}^p, i \in [m], j \in [n]$.*

With the metric in (7), the convergence of GT-SRVR/GT-SRVRI can be characterized as follows:

**Theorem 3** (Convergence of GT-SRVR). *Under Assumption 1 (a)&(c)-(e) and Assumption 2, if the step-sizes satisfy $\kappa \triangleq \gamma/\eta \leq \mu^2/13L_f^2$ and $\eta \leq \min\{k_5, k_6, k_7, k_8\}$, then we have the following convergence result for GT-SRVR:*

$$\frac{1}{T+1}\sum_{t=0}^{T}\mathbb{E}[\mathfrak{M}_t] \leq \frac{2\mathbb{E}[\mathfrak{p}_0 - J^*]}{\min\{1, L_f^2\}(T+1)\gamma},$$

*where $\mathfrak{p}_t$ is the potential function defined as:*

$$\mathfrak{p}_t \triangleq J(\bar{\boldsymbol{\theta}}_t) + \frac{8\gamma L_f^2}{\mu\eta}\|\bar{\boldsymbol{\omega}}_t - \boldsymbol{\omega}_t^*\|^2 + \frac{1}{m}\sum_{i=1}^{m}\left(\|\boldsymbol{\theta}_{i,t} - \bar{\boldsymbol{\theta}}_t\|^2 + \|\boldsymbol{\omega}_{i,t} - \bar{\boldsymbol{\omega}}_t\|^2 + \gamma\|\mathbf{p}_{i,t} - \bar{\mathbf{p}}_t\|^2 + \eta\|\mathbf{d}_{i,t} - \bar{\mathbf{d}}_t\|^2\right),$$

*and the constants in the step-size $\eta$ are:*

$$C_0 = \frac{1}{1-\lambda}\left(1 + \frac{1}{\kappa}\right) + \frac{1}{2} + \frac{18L_f^2}{\mu^2}, \quad k_5 = \frac{13L_f^2}{\mu^2\left(1/2 + 1/(1-\lambda)^2\right)}, \quad k_6 = \frac{1}{8\mu C_0},$$

$$k_7 = \frac{26(1-\lambda)L_f^2}{(\mu^2 + 144L_f^4 + 4L_f^2\mu^2 + 64C_0L_f^2\mu^2)}, \quad k_8 = \frac{13L_f^2}{2\mu^2}\left(L_f + \frac{L_f^2}{\mu} + (1-\lambda)\right).$$

**Theorem 4** (Convergence of GT-SRVRI). *Under Assumption 1 (a)&(c)-(e), Assumption 2, and the same parameter settings and potential function as stated in Theorem 3, we have the following convergence result for GT-SRVRI:*

$$\frac{1}{T+1}\sum_{t=0}^{T}\mathbb{E}[\mathfrak{M}_t] \leq \frac{2\mathbb{E}[\mathfrak{p}_0 - \mathfrak{p}_{T+1}]}{\min\{1, L_f^2\}(T+1)\gamma} + \frac{1}{\min\{1, L_f^2\}}\times$$

$$\left(\frac{12}{\lambda}\left(1 + \frac{1}{\kappa}\right) + 4 + \frac{144L_f^2}{\mu^2}\right)\left(\frac{\sigma^2\epsilon^2}{c_\epsilon} + \frac{\sigma^2 C(n, q, \alpha)}{T+1}\right), \quad (10)$$

*where the constant $C(n, q, \alpha)$ is defined as:*

$$C(n, q, \alpha) \triangleq \begin{cases} \frac{1}{1-\alpha}\left(\frac{n}{q}\right)^{(\frac{1}{\alpha}-1)} - \frac{\alpha}{1-\alpha}, & \text{if } \alpha > 0 \text{ and } \alpha \neq 1 \\ \log\left(\frac{n}{q}\right) + 1, & \text{if } \alpha = 1. \end{cases}$$

**Remark 3.** In Theorems 3 and 4, it can be seen that the step-sizes and convergence performance depend on the network topology and the ratio of the step-sizes $\kappa$. Additionally, the convergence performance of GT-SRVRI is affected by the constant $C(n, q, \alpha)$, which depends on the inexact gradient estimation in $t$th iteration with $\text{mod}(t, q) = 0$.

Following from Theorems 3 and 4, we immediately have the sample and communication complexity results for GT-SRVR/GT-SRVRI:

**Corollary 5.** *Under the conditions in Theorems 3 and 4, and with $q = \sqrt{n}$, to achieve an $\epsilon^2$-stationary solution (i.e., $\frac{1}{T+1}\sum_{t=0}^{T}\mathbb{E}[\mathfrak{M}_t] \leq \epsilon^2$) with $\sqrt{n}\epsilon^2 \leq 1$, we have:*

- *for GT-SRVR, the total communication rounds are $\mathcal{O}(\epsilon^{-2})$ and the total samples evaluated across the network are $\mathcal{O}(m\sqrt{n}\epsilon^{-2})$;*

- *GT-SRVRI with $\alpha \geq 1$, the total communication rounds are bounded by $\mathcal{O}(\log(\sqrt{n})\epsilon^{-2})$ and the total samples evaluated across the network are bounded by $\mathcal{O}(m\log(\sqrt{n})\sqrt{n}\epsilon^{-2})$.*

**Remark 4.** From Corollary 5, we can see that GT-SRVR has the same communication complexity as GT-GDA, but the sample complexity is lower than GT-GDA. For GT-SRVRI with $\alpha \geq 1$, the upper bounds of both the complexities have an additional factor $\log(\sqrt{n})$ factor compared with GT-SRVR. Although the theoretical complexity bounds for GT-SRVRI is weaker than GT-SRVR (due to abandoning full gradients completely), we show through experiments in Section 5 that GT-SRVRI empirically outperforms than GT-SRVR in practice.

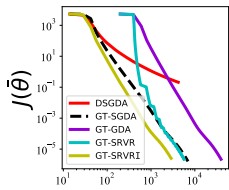 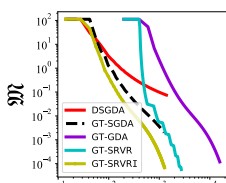 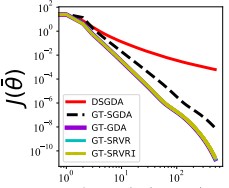 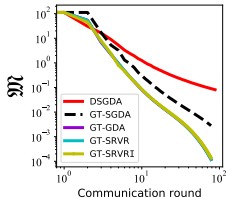

(a) MSPBE value vs. sample complexity.  (b) Convergence metric $\mathfrak{M}$ vs. sample complexity.  (c) [MSPBE value vs. comm. complexity.  (d) Convergence metric $\mathfrak{M}$ vs. comm. complexity.

Figure 1: MARL decentralized policy evaluation performance comparison.

# 5  Experimental Results

In this section, we demonstrate the performance of our proposed GT-GDA and GT-SRVR/GT-SRVRI algorithms for cooperative MARL decentralized policy evaluation. We adopt the environment of Cooperative Navigation task in [33], which consists of $m$ agents inhabiting a two-dimensional world with continuous space and discrete time. In the system, agents cooperate with each other to reach their own landmarks. Due to space limitation, the detailed experimental settings and different network topologies are relegated to the supplementary material.

Since there is no existing work in the literature on solving decentralized policy evaluation with *nonlinear* function approximation for MARL, we compare our algorithms with two stochastic algorithms as simple baselines in our experiments: 1) Decentralized Stochastic Gradient Descent Ascent (DSGDA) and 2) Gradient-Tracking-Based Stochastic Gradient Descent Ascent (GT-SGDA) (see the supplementary material for their detailed definitions). We initialize the parameters from the normal distribution for all the algorithms. The learning rates is fixed as $\gamma = 10^{-1}, \eta = 10^{-1}$.

From Figure 1(a) and 1(b), it can be seen that GT-SRVRI converges much faster than other algorithms (GT-GDA, GT-SRVR, GT-SGD and DSGD) in terms of the total number of gradient evaluations. We can also observe that both GT-SRVR and GT-SRVRI have better sample efficiency in attaining high accuracy (error smaller than $10^{-9}$) than the other three algorithms thanks to the variance-reduced techniques. As is shown in Figure 1(c) and 1(d), GT-SRVR and GT-SRVRI have the same communication cost as GT-GDA, which is much lower than those of DSGDA and GT-SGDA. Our experimental results confirm our theoretical analysis that GT-SRVR/GT-SRVRI enjoy low sample and communication complexities.

# 6  Conclusion

In this paper, we studied the decentralized MARL policy evaluation problem with nonlinear function approximation. We first reformulated the problem as a decentralized non-convex-strongly-concave minimax problem and developed a gradient tracking based algorithm called GT-GDA. We showed that GT-GDA algorithm has the communication complexity of $\mathcal{O}(\epsilon^{-2})$ and sample complexity of $\mathcal{O}(mn\epsilon^{-2})$. To further reduce the sample complexity while maintaining the communication complexity, we proposed two stochastic variance-reduced methods called GT-SRVR and GT-SRVRI, both of which can achieve the same communication complexity as GT-GDA but improve the sample complexity to $\mathcal{O}(m\sqrt{n}\epsilon^{-2})$. We have also conducted extensive numerical studies to verify the performance of our proposed algorithms. We note that our work opens up several interesting direction for future research. First, It is interesting to adopt communication-efficient mechanisms to further reduce the communication cost, especially when the parameters are high-dimensional. Second, it is also interesting to study MARL problems with partially observable information. Lastly, decentralized MARL policy evaluation with non-linear approximation with Markovian online sampling remains an important open problem, which is worth further investigation.

## Acknowledgments and Disclosure of Funding

This work has been supported in part by NSF grants CAREER CNS-2110259, CNS-2112471, CNS-2102233, CCF-2110252, ECCS-2140277, NSF-CCF-1934884 and a Google Faculty Research Award.

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
