# Supplementary Material

## A    Further experiments and additional results

In the followings, we provide the detailed settings for our experiments:

**1) RL environment:** We adopt the environment of Cooperative Navigation task in [33], which consists of $m$ agents inhabiting a two-dimensional world with continuous space and discrete time. In the system, agents cooperate with each other to reach their own landmarks.

**2) Multi-agent networks:** We use a six-node multi-agent system, with the communication graph $\mathcal{G}$ being generated by the Erdös-Rènyi graph, where the edge connectivity probability is $p_c = 0.35$. The network consensus matrix is chosen as $\mathbf{W} = \mathbf{I} - \frac{2}{3\lambda_{\max}(\mathbf{L})}\mathbf{L}$, where $\mathbf{L}$ is the Laplacian matrix of $\mathcal{G}$, and $\lambda_{\max}(\mathbf{L})$ denotes the largest eigenvalue of $\mathbf{L}$. The generated topology is shown in Figure 2.

**3) Data generation and model:** We first obtain a good policy and then generate a trajectory of the state-action pairs. In all experiments, the trajectory length is $n = 200$. We set the discount factor $\gamma = 0.95$. Then, $\mathcal{V}_{\boldsymbol{\theta}}(\cdot)$ is parametrized by a 2-hidden-layer neural network with 20 hidden units, where the Sigmoid activation is used at each unit.

**4 Decentralized stochastic algorithms for comparisons:**

1) *Decentralized Stochastic Gradient Descent Ascent (DSGDA):* This algorithm is motivated by DSGD [39, 19]. Each agent updates its local parameters as $\boldsymbol{\theta}_{i,t+1} = \sum_{j\in\mathcal{N}_i}[\mathbf{M}]_{ij}\boldsymbol{\theta}_{j,t} - \gamma\frac{1}{|\mathcal{S}_{i,t}|}\sum_{j\in\mathcal{S}_{i,t}}\nabla_{\boldsymbol{\theta}}f_{ij}(\boldsymbol{\theta}_{i,t},\boldsymbol{\omega}_{i,t})$ and $\boldsymbol{\omega}_{i,t+1} = \sum_{j\in\mathcal{N}_i}[\mathbf{M}]_{ij}\boldsymbol{\omega}_{j,t} - \eta\frac{1}{|\mathcal{S}_{i,t}|}\sum_{j\in\mathcal{S}_{i,t}}\nabla_{\boldsymbol{\omega}}f_{ij}(\boldsymbol{\theta}_{i,t},\boldsymbol{\omega}_{i,t})$.

2) *Gradient-Tracking-Based Stochastic Gradient Descent Ascent (GT-SGDA)*: This algorithm is motivated by the GT-SGD algorithm [58, 35]. GT-SGDA has the same structure as that of GT-GDA, but it updates $\mathbf{v}_{i,t}$ and $\mathbf{u}_{i,t}$ using stochastic gradients as follows: $\mathbf{v}_{i,t} = \frac{1}{|\mathcal{S}_{i,t}|}\sum_{j\in\mathcal{S}_{i,t}}\nabla_{\boldsymbol{\theta}}f_{ij}(\boldsymbol{\theta}_{i,t},\boldsymbol{\omega}_{i,t})$ and $\mathbf{u}_{i,t} = \frac{1}{|\mathcal{S}_{i,t}|}\sum_{j\in\mathcal{S}_{i,t}}\nabla_{\boldsymbol{\omega}}f_{ij}(\boldsymbol{\theta}_{i,t},\boldsymbol{\omega}_{i,t})$.

### A.1    Algorithms comparison

In this subsection, we provide an additional experiment on the algorithms' comparison. We run all algorithms for solving optimization problem over cooperative MARL decentralized policy evaluation. From Figure 3(a) and 3(b), it can be seen that GT-SRVRI converges faster than other algorithms (GT-GDA, GT-SRVR, GT-SGD and DSGD) in terms of the total number of gradient evaluations. In this experiment, we initialized the parameters from the normal distribution for all the algorithms and fixed learning rates as $\gamma = 10^{-2}, \eta = 10^{-2}$. As is shown in Figure 3(c) and 3(d), GT-SRVR and GT-SRVRI have the

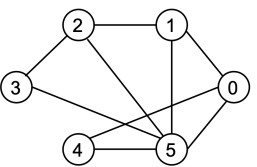

Figure 2: Network topology

same communication cost as GT-GDA, which is much lower than those of DSGDA and GT-SGDA. The experimental results confirm our theoretical analysis again that GT-SRVR/GT-SRVRI enjoy low sample and communication complexities.

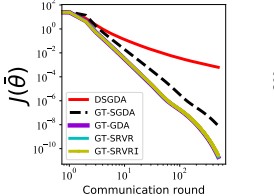 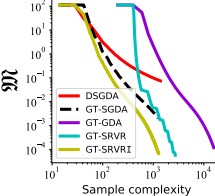 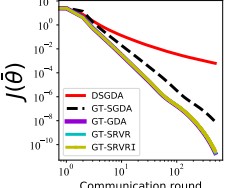 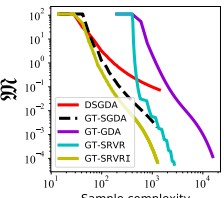

(a) MSPBE value vs. sample complexity.  (b) Convergence metric $\mathfrak{M}$ vs. sample complexity.  (c) MSPBE value vs. comm. complexity.  (d) Convergence metric $\mathfrak{M}$ vs. comm. complexity.

Figure 3: MARL decentralized policy evaluation performance comparison.

## A.2   Learning rate setting

We use a 6-node multi-agent system with a generated topology as shown in Figure 2. In this experiment, we choose the trajectory length $n = 200$, discount factor $\gamma = 0.95$ and mini-batch size $q = \lceil \sqrt{n} \rceil$. $\mathcal{V}_{\boldsymbol{\theta}}(\cdot)$ is parametrized by a 2-hidden-layer neural network with 20 hidden units, where the Sigmoid activation is used at each unit. Figs. 4-5 illustrate the objective function $J(\boldsymbol{\theta})$ and convergence metric $\mathfrak{M}$ performance of GT-GDA and GT-SRVR with different learning rates $\gamma$ and $\eta$. Since excessive learning rates will result in large fluctuations in loss function values. Thus, we fix a relatively small learning rate $\gamma = 10^{-3}$ while comparing $\eta$; and set $\eta = 10^{-3}$ while comparing $\gamma$. In this experiment, we observe that methods with a smaller learning rate have a smaller slope in the figure, which leads to a slower convergence.

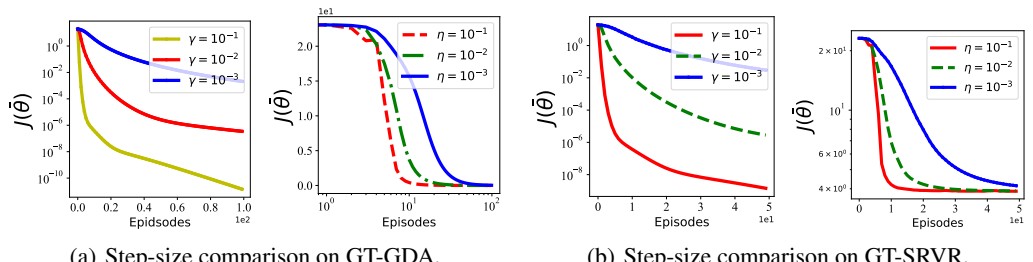

(a) Step-size comparison on GT-GDA.          (b) Step-size comparison on GT-SRVR.

Figure 4:  Performance of objective function with different step-size.

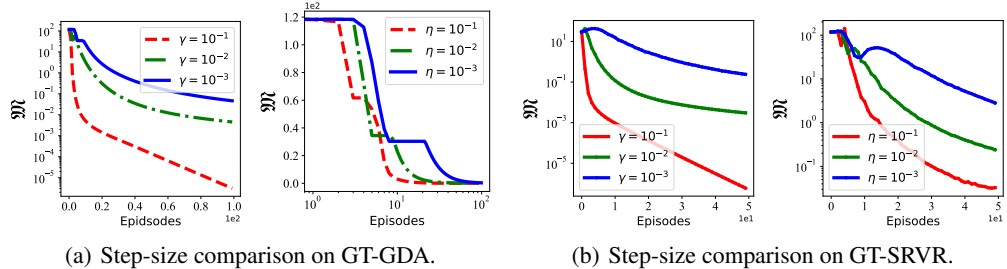

(a) Step-size comparison on GT-GDA.          (b) Step-size comparison on GT-SRVR.

Figure 5:  Performance of convergence metric $\mathfrak{M}$ with different step-size.

## A.3   Topology setting

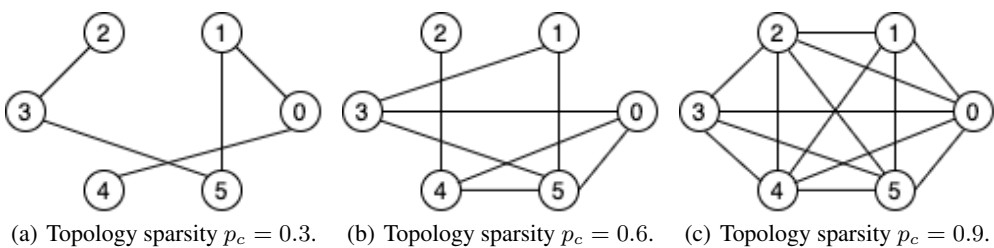

(a) Topology sparsity $p_c = 0.3$.   (b) Topology sparsity $p_c = 0.6$.   (c) Topology sparsity $p_c = 0.9$.

Figure 6:  Topology.

We use a 6-node multi-agent system and experiment on three different topologies. The generated topology with different sparsity are shown in Fig. 6. The trajectory length is $n = 200$ and we set the discount factor $\gamma = 0.95$, constant learning rate $\gamma = 0.1$, $\eta = 0.1$ and mini-batch size $q = \lceil \sqrt{n} \rceil$. We observe that the objective function $J(\theta)$ and convergence metric $\mathfrak{M}$ are insensitive to the network topology. The subplot in Fig. 7 shows that the $J(\theta)$ and $\mathfrak{M}$ slightly increase as $p_c$ decreases.

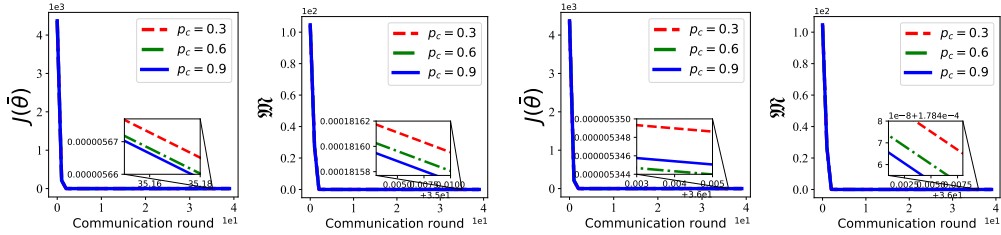

(a) Topology sparsity comparison on GT-GDA.  (b) Topology sparsity comparison on GT-SRVR.

Figure 7: Performance with different topology.

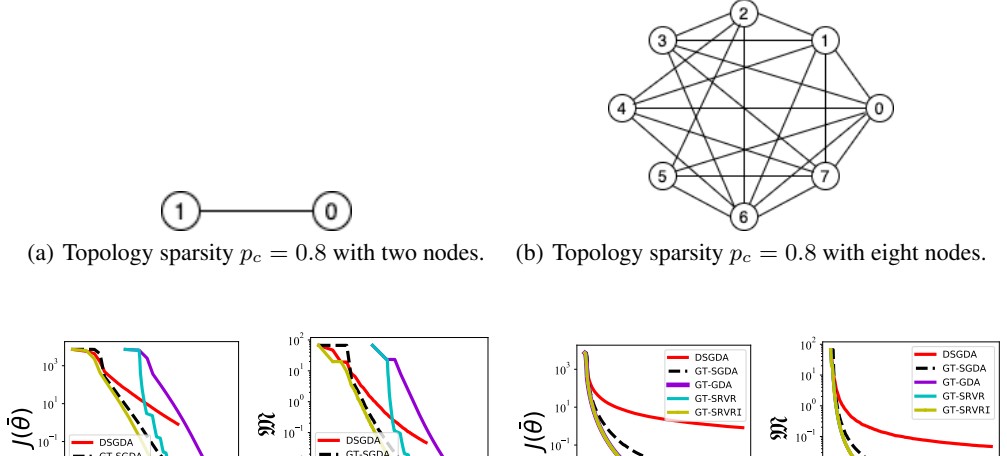

(a) Topology sparsity $p_c = 0.8$ with two nodes.  (b) Topology sparsity $p_c = 0.8$ with eight nodes.

(c) Sample complexity v.s. MSPBE & convergence (d) Communication complexity v.s. MSPBE & convergence metric.

Figure 8: Performance with two nodes.

## A.4   Node setting

We test the following experiments on different multi-agent systems. The generated topology with a 2-node system and a 8-node system are shown in Figs. A.4. We further adopt a 20-node system with topology $p_c = 0.5$. The trajectory length is $n = 200$ and we set the discount factor $\gamma = 0.95$, constant learning rate $\gamma = 0.1$, $\eta = 0.1$ and mini-batch size $q = \lceil \sqrt{n} \rceil$. We compare our proposed algorithm GT-GDA and GT-SRVR/GT-SRVRI with two baseline algorithms GT-SGDA and DSGDA mentioned in Section 5 in terms of MSPBE $J(\boldsymbol{\theta}) = \max_{\boldsymbol{\omega} \in \mathbb{R}^p} F(\boldsymbol{\theta}, \boldsymbol{\omega})$ and the convergence metric in (7). In Figs. 8,9,10 ,we observe similar results as shown in Section 5. Thus, we can conclude that our proposed algorithms GT-SRVR/GT-SRVRI enjoy low sample and communication complexities in general.

## A.5   Linear approximation v.s. Nonlinear approximation

We fixed learning rates as $\gamma = 10^{-1}, \eta = 10^{-1}$ and compared the Mean Squared Error between the ground truth value function and the estimated value function over three independent runs with nonlinear approximation. We note that the ground truth can be calculated using tabular policy evaluation and the estimated value functions is learned by our stated algorithms SRVR and SRVR$\mathcal{I}$. The mean square error (MSE) of SRVR is $0.084 \pm 0.003$, MSE of SRVR$\mathcal{I}$ is $0.092 \pm 0.005$. Furthermore, with linear approximation being applied on our stated algorithms, we have MSE result as $0.1747 \pm 0.012$

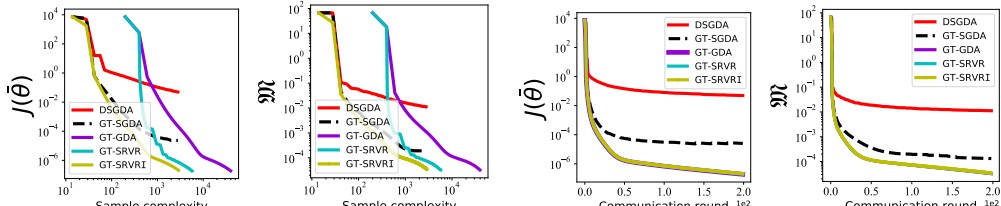

(a) Sample complexity v.s. MSPBE & convergence metric.

(b) Communication complexity v.s. MSPBE & convergence metric.

Figure 9: Performance with 8 Nodes.

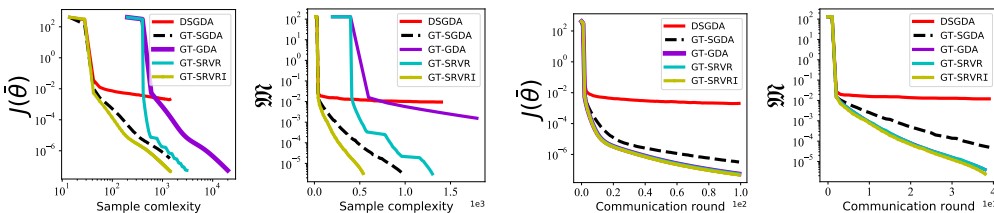

(a) Sample complexity v.s. MSPBE & convergence metric.

(b) Communication complexity v.s. MSPBE & convergence metric.

Figure 10: Performance with 20 Nodes and topology sparsity $p_c = 0.5$.

on SRVR and $0.1895 \pm 0.014$ on SRVR$\mathcal{I}$. Thus, our algorithms achieve lower MSE and perform better under nonlinear approximation than linear approximation.

## B  Proof of Theorem 1

In this section, we organize the proof of Theorem 1 into several key lemmas. Before diving in our theoretical analysis, we first define the following notations:

- $\bar{\mathbf{x}}_t = \frac{1}{m} \sum_{i=1}^{m} \mathbf{x}_{i,t}$ and $\mathbf{x}_t = [\mathbf{x}_{1,t}^\top, \cdots, \mathbf{x}_{m,t}^\top]^\top$ for any vector $\mathbf{x}$;
- $\nabla_{\boldsymbol{\theta}} F_t = [\nabla_{\boldsymbol{\theta}} F(\boldsymbol{\theta}_{1,t}, \boldsymbol{\omega}_{1,t})^\top, \cdots, \nabla_{\boldsymbol{\theta}} F(\boldsymbol{\theta}_{m,t}, \boldsymbol{\omega}_{m,t})^\top]^\top$;
- $\nabla_{\boldsymbol{\omega}} F_t = [\nabla_{\boldsymbol{\omega}} F(\boldsymbol{\theta}_{1,t}, \boldsymbol{\omega}_{1,t})^\top, \cdots, \nabla_{\boldsymbol{\omega}} F(\boldsymbol{\theta}_{m,t}, \boldsymbol{\omega}_{m,t})^\top]^\top$;
- $\mathcal{E}(\mathbf{x}_t) = \frac{1}{m} \sum_{i=1}^{m} \|\mathbf{x}_{i,t} - \bar{\mathbf{x}}_t\|^2$ for any vector $\mathbf{x}$.

Our first step is to show the following descent property of GT-GDA algorithm on the function $J(\cdot)$:

**Lemma 1** (Descent Inequality on $J(\boldsymbol{\theta})$). *Under Assumption 1, the following descent inequality holds under GT-GDA:*

$$J(\bar{\boldsymbol{\theta}}_{t+1}) - J(\bar{\boldsymbol{\theta}}_t) \leq -\frac{\gamma}{2}\|\nabla J(\bar{\boldsymbol{\theta}}_t)\|^2 - (\frac{\gamma}{2} - \frac{L_J\gamma^2}{2})\|\bar{\mathbf{p}}_t\|^2$$
$$+ \gamma L_F^2 \|\boldsymbol{\omega}_t^* - \bar{\boldsymbol{\omega}}_t\|^2 + \gamma\|\nabla_{\boldsymbol{\theta}} F(\bar{\boldsymbol{\theta}}_t, \bar{\boldsymbol{\omega}}_t) - \bar{\mathbf{p}}_t\|^2, \quad (11)$$

*where $J(\boldsymbol{\theta}_t) = \max_{\boldsymbol{\omega}} F(\boldsymbol{\theta}_t, \boldsymbol{\omega})$ and $\boldsymbol{\omega}_t^* = \arg\max_{\boldsymbol{\omega}} F(\boldsymbol{\theta}_t, \boldsymbol{\omega})$.*

*Proof.* According to the algorithm update, we have:

$$J(\bar{\boldsymbol{\theta}}_{t+1}) - J(\bar{\boldsymbol{\theta}}_t) \overset{(a)}{\leq} \langle \nabla J(\bar{\boldsymbol{\theta}}_t), \bar{\boldsymbol{\theta}}_{t+1} - \bar{\boldsymbol{\theta}}_t \rangle + \frac{L_J}{2}\|\bar{\boldsymbol{\theta}}_{t+1} - \bar{\boldsymbol{\theta}}_t\|^2$$

$$\overset{(b)}{=} -\gamma\langle \nabla J(\bar{\boldsymbol{\theta}}_t), \bar{\mathbf{p}}_t \rangle + \frac{L_J\gamma^2}{2}\|\bar{\mathbf{p}}_t\|^2$$

$$= -\frac{\gamma}{2}\|\nabla J(\bar{\boldsymbol{\theta}}_t)\|^2 - (\frac{\gamma}{2} - \frac{L_J\gamma^2}{2})\|\bar{\mathbf{p}}_t\|^2 + \frac{\gamma}{2}\|\nabla J(\bar{\boldsymbol{\theta}}_t) - \bar{\mathbf{p}}_t\|^2$$

$$= -\frac{\gamma}{2}\|\nabla J(\bar{\boldsymbol{\theta}}_t)\|^2 - (\frac{\gamma}{2} - \frac{L_J\gamma^2}{2})\|\bar{\mathbf{p}}_t\|^2 + \frac{\gamma}{2}\|\nabla J(\bar{\boldsymbol{\theta}}_t) - \nabla_{\boldsymbol{\theta}}F(\bar{\boldsymbol{\theta}}_t, \bar{\boldsymbol{\omega}}_t) + \nabla_{\boldsymbol{\theta}}F(\bar{\boldsymbol{\theta}}_t, \bar{\boldsymbol{\omega}}_t) - \bar{\mathbf{p}}_t\|^2$$

$$\overset{(c)}{\leq} -\frac{\gamma}{2}\|\nabla J(\bar{\boldsymbol{\theta}}_t)\|^2 - (\frac{\gamma}{2} - \frac{L_J\gamma^2}{2})\|\bar{\mathbf{p}}_t\|^2 + \gamma\|\nabla J(\bar{\boldsymbol{\theta}}_t) - \nabla_{\boldsymbol{\theta}}F(\bar{\boldsymbol{\theta}}_t, \bar{\boldsymbol{\omega}}_t)\|^2 + \gamma\|\nabla_{\boldsymbol{\theta}}F(\bar{\boldsymbol{\theta}}_t, \bar{\boldsymbol{\omega}}_t) - \bar{\mathbf{p}}_t\|^2$$

$$\overset{(d)}{\leq} -\frac{\gamma}{2}\|\nabla J(\bar{\boldsymbol{\theta}}_t)\|^2 - (\frac{\gamma}{2} - \frac{L_J\gamma^2}{2})\|\bar{\mathbf{p}}_t\|^2 + \gamma L_F^2\|\boldsymbol{\omega}_t^* - \bar{\boldsymbol{\omega}}_t\|^2 + \gamma\|\nabla_{\boldsymbol{\theta}}F(\bar{\boldsymbol{\theta}}_t, \bar{\boldsymbol{\omega}}_t) - \bar{\mathbf{p}}_t\|^2, \qquad (12)$$

where (a) is because of Lipschitz continuous gradients of $J$, (b) follows from the update rule (5), (c) follows from $\|\mathbf{x} + \mathbf{y}\|^2 \leq 2\|\mathbf{x}\|^2 + 2\|\mathbf{y}\|^2 \ \forall \mathbf{x}, \mathbf{y}$, and (d) follows from Lemma 10 and Assumption 1 (b). This completes the proof of the lemma. $\qquad\square$

Note that in the RHS of (11), there is an error term $\|\boldsymbol{\omega}_t^* - \bar{\boldsymbol{\omega}}_t\|^2$. The following lemma states the contraction property of this term.

**Lemma 2** (Error Bound on $\boldsymbol{\omega}^*(\boldsymbol{\theta})$)**.** *Under Assumption 1 and with $\eta \leq 1/2L_F$, the following inequality holds for GT-GDA:*

$$\|\bar{\boldsymbol{\omega}}_{t+1} - \boldsymbol{\omega}_{t+1}^*\|^2 \leq (1 - \frac{\mu\eta}{4})\|\boldsymbol{\omega}_t^* - \bar{\boldsymbol{\omega}}_t\|^2 + \frac{9\eta}{2\mu}\|\nabla_{\boldsymbol{\omega}}F(\bar{\boldsymbol{\theta}}_t, \bar{\boldsymbol{\omega}}_t) - \bar{\mathbf{d}}_t\|$$

$$- (1 + \frac{\mu\eta}{4})\frac{\eta^2}{4}\|\bar{\mathbf{d}}_t\|^2 + \frac{5L_{\boldsymbol{\omega}}^2\gamma^2}{\mu\eta}\|\bar{\mathbf{p}}_t\|^2, \qquad (13)$$

*where $\boldsymbol{\omega}_t^* = \boldsymbol{\omega}^*(\bar{\boldsymbol{\theta}}_t) = \arg\max_{\boldsymbol{\omega}} F(\bar{\boldsymbol{\theta}}_t, \boldsymbol{\omega})$.*

*Proof.* Define $\boldsymbol{\omega}_t^* = \boldsymbol{\omega}^*(\bar{\boldsymbol{\theta}}_t) = \arg\max_{\boldsymbol{\omega}} F(\bar{\boldsymbol{\theta}}_t, \boldsymbol{\omega})$. We have:

$$\|\bar{\boldsymbol{\omega}}_{t+1} - \boldsymbol{\omega}_t^*\|^2 = \|\bar{\boldsymbol{\omega}}_t + \eta\bar{\mathbf{d}}_t - \boldsymbol{\omega}_t^*\|^2 = \|\bar{\boldsymbol{\omega}}_t - \boldsymbol{\omega}_t^*\|^2 + \eta^2\|\bar{\mathbf{d}}_t\|^2 + 2\eta\langle\bar{\boldsymbol{\omega}}_t - \boldsymbol{\omega}_t^*, \bar{\mathbf{d}}_t\rangle. \qquad (14)$$

To remove the third term in (14), we have:

$$F(\bar{\boldsymbol{\theta}}_t, \boldsymbol{\omega}) - F(\bar{\boldsymbol{\theta}}_t, \bar{\boldsymbol{\omega}}_t) + \frac{\mu}{2}\|\boldsymbol{\omega} - \bar{\boldsymbol{\omega}}_t\|^2 \leq \langle\nabla_{\boldsymbol{\omega}}F(\bar{\boldsymbol{\theta}}_t, \bar{\boldsymbol{\omega}}_t), \boldsymbol{\omega} - \bar{\boldsymbol{\omega}}_t\rangle$$

$$= \langle\bar{\mathbf{d}}_t, \boldsymbol{\omega} - \bar{\boldsymbol{\omega}}_{t+1}\rangle + \langle\nabla_{\boldsymbol{\omega}}F(\bar{\boldsymbol{\theta}}_t, \bar{\boldsymbol{\omega}}_t) - \bar{\mathbf{d}}_t, \boldsymbol{\omega} - \bar{\boldsymbol{\omega}}_{t+1}\rangle + \langle\nabla_{\boldsymbol{\omega}}F(\bar{\boldsymbol{\theta}}_t, \bar{\boldsymbol{\omega}}_t), \bar{\boldsymbol{\omega}}_{t+1} - \bar{\boldsymbol{\omega}}_t\rangle. \qquad (15)$$

With $\eta \leq 1/2L_F$ and by Assumption 1 (b), it follows that

$$-\frac{1}{4\eta}\|\bar{\boldsymbol{\omega}}_{t+1} - \bar{\boldsymbol{\omega}}_t\|^2 \leq -\frac{L_F}{2}\|\bar{\boldsymbol{\omega}}_{t+1} - \bar{\boldsymbol{\omega}}_t\|^2$$

$$\leq F(\bar{\boldsymbol{\theta}}_t, \boldsymbol{\omega}_{t+1}) - F(\bar{\boldsymbol{\theta}}_t, \bar{\boldsymbol{\omega}}_t) - \langle\nabla_{\boldsymbol{\omega}}F(\bar{\boldsymbol{\theta}}_t, \bar{\boldsymbol{\omega}}_t), \bar{\boldsymbol{\omega}}_{t+1} - \bar{\boldsymbol{\omega}}_t\rangle. \qquad (16)$$

Combining (15) and (16), with the update $\bar{\boldsymbol{\omega}}_{t+1} - \bar{\boldsymbol{\omega}}_t = \eta\bar{\mathbf{d}}_t$, we have:

$$F(\bar{\boldsymbol{\theta}}_t, \boldsymbol{\omega}) - F(\bar{\boldsymbol{\theta}}_t, \bar{\boldsymbol{\omega}}_{t+1}) + \frac{\mu}{2}\|\boldsymbol{\omega} - \bar{\boldsymbol{\omega}}_t\|^2$$

$$\leq \langle\bar{\mathbf{d}}_t, \boldsymbol{\omega} - \bar{\boldsymbol{\omega}}_{t+1}\rangle + \langle\nabla_{\boldsymbol{\omega}}F(\bar{\boldsymbol{\theta}}_t, \bar{\boldsymbol{\omega}}_t) - \bar{\mathbf{d}}_t, \boldsymbol{\omega} - \bar{\boldsymbol{\omega}}_{t+1}\rangle + \frac{1}{4\eta}\|\bar{\boldsymbol{\omega}}_{t+1} - \bar{\boldsymbol{\omega}}_t\|^2$$

$$= \langle\bar{\mathbf{d}}_t, \boldsymbol{\omega} - \bar{\boldsymbol{\omega}}_t\rangle + \langle\bar{\mathbf{d}}_t, \bar{\boldsymbol{\omega}}_t - \bar{\boldsymbol{\omega}}_{t+1}\rangle + \langle\nabla_{\boldsymbol{\omega}}F(\bar{\boldsymbol{\theta}}_t, \bar{\boldsymbol{\omega}}_t) - \bar{\mathbf{d}}_t, \boldsymbol{\omega} - \bar{\boldsymbol{\omega}}_{t+1}\rangle + \frac{\eta}{4}\|\bar{\mathbf{d}}_t\|^2$$

$$= \langle\bar{\mathbf{d}}_t, \boldsymbol{\omega} - \bar{\boldsymbol{\omega}}_t\rangle - \eta\|\bar{\mathbf{d}}_t\|^2 + \langle\nabla_{\boldsymbol{\omega}}F(\bar{\boldsymbol{\theta}}_t, \bar{\boldsymbol{\omega}}_t) - \bar{\mathbf{d}}_t, \boldsymbol{\omega} - \bar{\boldsymbol{\omega}}_{t+1}\rangle + \frac{\eta}{4}\|\bar{\mathbf{d}}_t\|^2$$

$$= \langle\bar{\mathbf{d}}_t, \boldsymbol{\omega} - \bar{\boldsymbol{\omega}}_t\rangle + \langle\nabla_{\boldsymbol{\omega}}F(\bar{\boldsymbol{\theta}}_t, \bar{\boldsymbol{\omega}}_t) - \bar{\mathbf{d}}_t, \boldsymbol{\omega} - \bar{\boldsymbol{\omega}}_{t+1}\rangle - \frac{3\eta}{4}\|\bar{\mathbf{d}}_t\|^2. \qquad (17)$$

Let $\boldsymbol{\omega} = \boldsymbol{\omega}_t^*$, we have

$$F(\bar{\boldsymbol{\theta}}_t, \boldsymbol{\omega}_t^*) - F(\bar{\boldsymbol{\theta}}_t, \bar{\boldsymbol{\omega}}_{t+1}) + \frac{\mu}{2}\|\boldsymbol{\omega}_t^* - \bar{\boldsymbol{\omega}}_t\|^2$$

$$\leq \langle\bar{\mathbf{d}}_t, \boldsymbol{\omega}_t^* - \bar{\boldsymbol{\omega}}_t\rangle + \langle\nabla_{\boldsymbol{\omega}}F(\bar{\boldsymbol{\theta}}_t, \bar{\boldsymbol{\omega}}_t) - \bar{\mathbf{d}}_t, \boldsymbol{\omega}_t^* - \bar{\boldsymbol{\omega}}_{t+1}\rangle - \frac{3\eta}{4}\|\bar{\mathbf{d}}_t\|^2$$

$$\overset{(a)}{\leq} \langle\bar{\mathbf{d}}_t, \boldsymbol{\omega}_t^* - \bar{\boldsymbol{\omega}}_t\rangle + \frac{2}{\mu}\|\nabla_{\boldsymbol{\omega}}F(\bar{\boldsymbol{\theta}}_t, \bar{\boldsymbol{\omega}}_t) - \bar{\mathbf{d}}_t\| + \frac{\mu}{8}\|\boldsymbol{\omega}_t^* - \bar{\boldsymbol{\omega}}_{t+1}\| - \frac{3\eta}{4}\|\bar{\mathbf{d}}_t\|^2$$

$$\overset{(b)}{\le} \langle \bar{\mathbf{d}}_t, \boldsymbol{\omega}_t^* - \bar{\boldsymbol{\omega}}_t \rangle + \frac{2}{\mu} \| \nabla_{\boldsymbol{\omega}} F(\bar{\boldsymbol{\theta}}_t, \bar{\boldsymbol{\omega}}_t) - \bar{\mathbf{d}}_t \| + \frac{\mu}{4} \| \boldsymbol{\omega}_t^* - \bar{\boldsymbol{\omega}}_t \|^2 + \frac{\mu}{4} \| \bar{\boldsymbol{\omega}}_t - \bar{\boldsymbol{\omega}}_{t+1} \|^2 - \frac{3\eta}{4} \| \bar{\mathbf{d}}_t \|^2$$

$$\overset{(c)}{=} \langle \bar{\mathbf{d}}_t, \boldsymbol{\omega}_t^* - \bar{\boldsymbol{\omega}}_t \rangle + \frac{2}{\mu} \| \nabla_{\boldsymbol{\omega}} F(\bar{\boldsymbol{\theta}}_t, \bar{\boldsymbol{\omega}}_t) - \bar{\mathbf{d}}_t \| + \frac{\mu}{4} \| \boldsymbol{\omega}_t^* - \bar{\boldsymbol{\omega}}_t \|^2 - (\frac{3\eta}{4} - \frac{\mu\eta^2}{4}) \| \bar{\mathbf{d}}_t \|^2, \tag{18}$$

where (a) follows from $-\langle \mathbf{x}, \mathbf{y} \rangle \le \frac{1}{2c} \| \mathbf{x} \|^2 + \frac{c}{2} \| \mathbf{y} \|^2$ and $c = \frac{\mu}{4}$, (b) is due to $\| \mathbf{x} + \mathbf{y} \|^2 \le 2 \| \mathbf{x} \|^2 + 2 \| \mathbf{y} \|^2$, and (c) is from $\bar{\boldsymbol{\omega}}_{t+1} - \bar{\boldsymbol{\omega}}_t = \eta \bar{\mathbf{d}}_t$.

Since $F(\bar{\boldsymbol{\theta}}_t, \boldsymbol{\omega}_t^*) \ge F(\bar{\boldsymbol{\theta}}_t, \bar{\boldsymbol{\omega}}_{t+1})$, we have

$$\frac{\mu\eta}{2} \| \boldsymbol{\omega}_t^* - \bar{\boldsymbol{\omega}}_t \|^2 \le 2\eta \langle \bar{\mathbf{d}}_t, \boldsymbol{\omega}_t^* - \bar{\boldsymbol{\omega}}_t \rangle + \frac{4\eta}{\mu} \| \nabla_{\boldsymbol{\omega}} F(\bar{\boldsymbol{\theta}}_t, \bar{\boldsymbol{\omega}}_t) - \bar{\mathbf{d}}_t \| - (\frac{3\eta^2}{2} - \frac{\mu\eta^3}{2}) \| \bar{\mathbf{d}}_t \|^2. \tag{19}$$

Combining (14) and (19) and setting $\eta \le 1/2\mu \le 1/2L_F$, we have

$$\| \bar{\boldsymbol{\omega}}_{t+1} - \boldsymbol{\omega}_t^* \|^2 \le (1 - \frac{\mu\eta}{2}) \| \boldsymbol{\omega}_t^* - \bar{\boldsymbol{\omega}}_t \|^2 + \frac{4\eta}{\mu} \| \nabla_{\boldsymbol{\omega}} F(\bar{\boldsymbol{\theta}}_t, \bar{\boldsymbol{\omega}}_t) - \bar{\mathbf{d}}_t \| - \frac{\eta^2}{4} \| \bar{\mathbf{d}}_t \|^2. \tag{20}$$

Then, it holds that

$$\| \bar{\boldsymbol{\omega}}_{t+1} - \boldsymbol{\omega}_{t+1}^* \|^2 = \| \bar{\boldsymbol{\omega}}_{t+1} - \boldsymbol{\omega}_t^* + \boldsymbol{\omega}_t^* - \boldsymbol{\omega}_{t+1}^* \|^2$$

$$\overset{(a)}{\le} (1 + \frac{\mu\eta}{4}) \| \bar{\boldsymbol{\omega}}_{t+1} - \boldsymbol{\omega}_t^* \|^2 + (1 + \frac{4}{\mu\eta}) \| \boldsymbol{\omega}_t^* - \boldsymbol{\omega}_{t+1}^* \|^2$$

$$\overset{(b)}{\le} (1 + \frac{\mu\eta}{4}) \| \bar{\boldsymbol{\omega}}_{t+1} - \boldsymbol{\omega}_t^* \|^2 + (1 + \frac{4}{\mu\eta}) L_{\boldsymbol{\omega}}^2 \| \bar{\boldsymbol{\theta}}_t - \bar{\boldsymbol{\theta}}_{t+1} \|^2$$

$$\overset{(c)}{\le} (1 + \frac{\mu\eta}{4})(1 - \frac{\mu\eta}{2}) \| \boldsymbol{\omega}_t^* - \bar{\boldsymbol{\omega}}_t \|^2 + (1 + \frac{\mu\eta}{4}) \frac{4\eta}{\mu} \| \nabla_{\boldsymbol{\omega}} F(\bar{\boldsymbol{\theta}}_t, \bar{\boldsymbol{\omega}}_t) - \bar{\mathbf{d}}_t \|$$

$$\quad - (1 + \frac{\mu\eta}{4}) \frac{\eta^2}{4} \| \bar{\mathbf{d}}_t \|^2 + (1 + \frac{4}{\mu\eta}) L_{\boldsymbol{\omega}}^2 \| \bar{\boldsymbol{\theta}}_t - \bar{\boldsymbol{\theta}}_{t+1} \|^2$$

$$\overset{(d)}{\le} (1 - \frac{\mu\eta}{4}) \| \boldsymbol{\omega}_t^* - \bar{\boldsymbol{\omega}}_t \|^2 + \frac{9\eta}{2\mu} \| \nabla_{\boldsymbol{\omega}} F(\bar{\boldsymbol{\theta}}_t, \bar{\boldsymbol{\omega}}_t) - \bar{\mathbf{d}}_t \| - (1 + \frac{\mu\eta}{4}) \frac{\eta^2}{4} \| \bar{\mathbf{d}}_t \|^2 + \frac{5 L_{\boldsymbol{\omega}}^2 \gamma^2}{\mu\eta} \| \bar{\mathbf{p}}_t \|^2, \tag{21}$$

where (a) follows from $\| \mathbf{x} + \mathbf{y} \|^2 \le (1 + 1/c) \| \mathbf{x} \|^2 + (1 + c) \| \mathbf{y} \|^2$ and $c = \mu\eta/4$, (b) follows from Lemma 9, (c) follows from plugging (20), and (d) due to the facts that:

$$(1 + \frac{\mu\eta}{4})(1 - \frac{\mu\eta}{2}) = 1 + \frac{\mu\eta}{4} - \frac{\mu\eta}{2} - \frac{\mu^2\eta^2}{8} \le 1 - \frac{\mu\eta}{4},$$

$$(1 + \frac{\mu\eta}{4}) \frac{4\eta}{\mu} \le (1 + \frac{\mu}{4} \cdot \frac{1}{2\mu}) \frac{4\eta}{\mu} = \frac{9\eta}{2\mu},$$

$$1 + \frac{4}{\mu\eta} \le \frac{1}{\mu\eta} + \frac{4}{\mu\eta} = \frac{5}{\mu\eta} \quad \text{, and } \bar{\boldsymbol{\theta}}_t - \bar{\boldsymbol{\theta}}_{t+1} = \gamma \bar{\mathbf{p}}_t. \tag{22}$$

Plugging (22) into (21) yields:

$$\| \bar{\boldsymbol{\omega}}_{t+1} - \boldsymbol{\omega}_{t+1}^* \|^2 - \| \boldsymbol{\omega}_t^* - \bar{\boldsymbol{\omega}}_t \|^2$$

$$\le - \frac{\mu\eta}{4} \| \boldsymbol{\omega}_t^* - \bar{\boldsymbol{\omega}}_t \|^2 + \frac{9\eta}{2\mu} \| \nabla_{\boldsymbol{\omega}} F(\bar{\boldsymbol{\theta}}_t, \bar{\boldsymbol{\omega}}_t) - \bar{\mathbf{d}}_t \| - (1 + \frac{\mu\eta}{4}) \frac{\eta^2}{4} \| \bar{\mathbf{d}}_t \|^2 + \frac{5 L_{\boldsymbol{\omega}}^2 \gamma^2}{\mu\eta} \| \bar{\mathbf{p}}_t \|^2. \tag{23}$$

This completes the proof of the lemma. $\qquad \square$

Next, by combining the results from Lemmas 1-2, we have the following descent result:

**Lemma 3.** *Under Assumption 1 and with $\eta \le 1/2L_F$, the following inequality holds for GT-GDA:*

$$J(\bar{\boldsymbol{\theta}}_{T+1}) - J(\bar{\boldsymbol{\theta}}_0) + \frac{8\gamma L_F^2}{\mu\eta} \left[ \| \bar{\boldsymbol{\omega}}_{T+1} - \boldsymbol{\omega}_{T+1}^* \|^2 - \| \boldsymbol{\omega}_0^* - \bar{\boldsymbol{\omega}}_0 \|^2 \right]$$

$$\le - \frac{\gamma}{2} \sum_{t=0}^T \| \nabla J(\bar{\boldsymbol{\theta}}_t) \|^2 - \gamma L_F^2 \sum_{t=0}^T \| \boldsymbol{\omega}_t^* - \bar{\boldsymbol{\omega}}_t \|^2 - (\frac{\gamma}{2} - \frac{L_J \gamma^2}{2} - \frac{40\gamma^3 L_{\boldsymbol{\omega}}^2 L_F^2}{\mu^2\eta^2}) \sum_{t=0}^T \| \bar{\mathbf{p}}_t \|^2 - \frac{2\gamma\eta L_F^2}{\mu} \sum_{t=0}^T \| \bar{\mathbf{d}}_t \|^2$$

$$+ (\frac{72\gamma L_F^4}{\mu^2 m} + \frac{2\gamma L_F^2}{m}) \sum_{t=0}^T [\mathcal{E}(\boldsymbol{\theta}_t) + \mathcal{E}(\boldsymbol{\omega}_t)] + \frac{2\gamma}{m} \sum_{t=0}^T \| \nabla_{\boldsymbol{\omega}} F_t - \mathbf{v}_t \|^2 + \frac{72 L_F^2 \gamma}{m\mu^2} \sum_{t=0}^T \| \nabla_{\boldsymbol{\theta}} F_t - \mathbf{u}_t \|^2.$$

*Proof.* From Lemmas 1-2, we have

$$J(\bar{\boldsymbol{\theta}}_{t+1}) - J(\bar{\boldsymbol{\theta}}_t) + \frac{8\gamma L_F^2}{\mu\eta}\left[\|\bar{\boldsymbol{\omega}}_{t+1} - \boldsymbol{\omega}_{t+1}^*\|^2 - \|\boldsymbol{\omega}_t^* - \bar{\boldsymbol{\omega}}_t\|^2\right]$$

$$\leq -\frac{\gamma}{2}\|\nabla J(\bar{\boldsymbol{\theta}}_t)\|^2 - \gamma L_F^2\|\boldsymbol{\omega}_t^* - \bar{\boldsymbol{\omega}}_t\|^2 - (\frac{\gamma}{2} - \frac{L_J\gamma^2}{2} - \frac{40\gamma^3 L_{\boldsymbol{\omega}}^2 L_F^2}{\mu^2\eta^2})\|\bar{\mathbf{p}}_t\|^2 - \frac{2\gamma\eta L_F^2}{\mu}\|\bar{\mathbf{d}}_t\|^2$$

$$+ \gamma\|\nabla_{\boldsymbol{\theta}}F(\bar{\boldsymbol{\theta}}_t, \bar{\boldsymbol{\omega}}_t) - \bar{\mathbf{p}}_t\|^2 + \frac{36L_F^2\gamma}{\mu^2}\|\nabla_{\boldsymbol{\omega}}F(\bar{\boldsymbol{\theta}}_t, \bar{\boldsymbol{\omega}}_t) - \bar{\mathbf{d}}_t\|^2. \tag{24}$$

Note that

$$\|\nabla_{\boldsymbol{\theta}}F(\bar{\boldsymbol{\theta}}_t, \bar{\boldsymbol{\omega}}_t) - \bar{\mathbf{p}}_t\|^2$$

$$=\|\nabla_{\boldsymbol{\theta}}F(\bar{\boldsymbol{\theta}}_t, \bar{\boldsymbol{\omega}}_t) - \frac{1}{m}\sum_{i=1}^m \nabla_{\boldsymbol{\theta}}F_i(\boldsymbol{\theta}_{i,t}, \boldsymbol{\omega}_{i,t}) + \frac{1}{m}\sum_{i=1}^m \nabla_{\boldsymbol{\theta}}F_i(\boldsymbol{\theta}_{i,t}, \boldsymbol{\omega}_{i,t}) - \bar{\mathbf{p}}_t\|^2$$

$$\leq 2\|\nabla_{\boldsymbol{\theta}}F(\bar{\boldsymbol{\theta}}_t, \bar{\boldsymbol{\omega}}_t) - \frac{1}{m}\sum_{i=1}^m \nabla_{\boldsymbol{\theta}}F_i(\boldsymbol{\theta}_{i,t}, \boldsymbol{\omega}_{i,t})\|^2 + 2\|\frac{1}{m}\sum_{i=1}^m \nabla_{\boldsymbol{\theta}}F_i(\boldsymbol{\theta}_{i,t}, \boldsymbol{\omega}_{i,t}) - \bar{\mathbf{p}}_t\|^2$$

$$\leq \frac{2}{m}\sum_{i=1}^m \|\nabla_{\boldsymbol{\theta}}F(\bar{\boldsymbol{\theta}}_t, \bar{\boldsymbol{\omega}}_t) - \nabla_{\boldsymbol{\theta}}F_i(\boldsymbol{\theta}_{i,t}, \boldsymbol{\omega}_{i,t})\|^2 + 2\|\frac{1}{m}\sum_{i=1}^m \nabla_{\boldsymbol{\theta}}F_i(\boldsymbol{\theta}_{i,t}, \boldsymbol{\omega}_{i,t}) - \bar{\mathbf{p}}_t\|^2$$

$$\leq \frac{2L_f^2}{m}\sum_{i=1}^m [\|\bar{\boldsymbol{\theta}}_t - \boldsymbol{\theta}_{i,t}\|^2 + \|\bar{\boldsymbol{\omega}}_t - \boldsymbol{\omega}_{i,t}\|^2] + 2\|\frac{1}{m}\sum_{i=1}^m \nabla_{\boldsymbol{\theta}}F_i(\boldsymbol{\theta}_{i,t}, \boldsymbol{\omega}_{i,t}) - \bar{\mathbf{p}}_t\|^2. \tag{25}$$

Similarly, we have:

$$\|\nabla_{\boldsymbol{\omega}}F(\bar{\boldsymbol{\theta}}_t, \bar{\boldsymbol{\omega}}_t) - \bar{\mathbf{d}}_t\|^2 \leq \frac{2L_F^2}{m}\sum_{i=1}^m [\|\bar{\boldsymbol{\theta}}_t - \boldsymbol{\theta}_{i,t}\|^2 + \|\bar{\boldsymbol{\omega}}_t - \boldsymbol{\omega}_{i,t}\|^2]$$

$$+ 2\|\frac{1}{m}\sum_{i=1}^m \nabla_{\boldsymbol{\omega}}F_i(\boldsymbol{\theta}_{i,t}, \boldsymbol{\omega}_{i,t}) - \bar{\mathbf{d}}_t\|^2. \tag{26}$$

Thus, we have

$$J(\bar{\boldsymbol{\theta}}_{t+1}) - J(\bar{\boldsymbol{\theta}}_t) + \frac{8\gamma L_F^2}{\mu\eta}\left[\|\bar{\boldsymbol{\omega}}_{t+1} - \boldsymbol{\omega}_{t+1}^*\|^2 - \|\boldsymbol{\omega}_t^* - \bar{\boldsymbol{\omega}}_t\|^2\right]$$

$$\leq -\frac{\gamma}{2}\|\nabla J(\bar{\boldsymbol{\theta}}_t)\|^2 - \gamma L_F^2\|\boldsymbol{\omega}_t^* - \bar{\boldsymbol{\omega}}_t\|^2 - (\frac{\gamma}{2} - \frac{L_J\gamma^2}{2} - \frac{40\gamma^3 L_{\boldsymbol{\omega}}^2 L_F^2}{\mu^2\eta^2})\|\bar{\mathbf{p}}_t\|^2 - \frac{2\gamma\eta L_F^2}{\mu}\|\bar{\mathbf{d}}_t\|^2$$

$$+ \gamma\|\nabla_{\boldsymbol{\theta}}F(\bar{\boldsymbol{\theta}}_t, \bar{\boldsymbol{\omega}}_t) - \bar{\mathbf{p}}_t\|^2 + \frac{36L_F^2\gamma}{\mu^2}\|\nabla_{\boldsymbol{\omega}}F(\bar{\boldsymbol{\theta}}_t, \bar{\boldsymbol{\omega}}_t) - \bar{\mathbf{d}}_t\|^2$$

$$\leq -\frac{\gamma}{2}\|\nabla J(\bar{\boldsymbol{\theta}}_t)\|^2 - \gamma L_F^2\|\boldsymbol{\omega}_t^* - \bar{\boldsymbol{\omega}}_t\|^2 - (\frac{\gamma}{2} - \frac{L_J\gamma^2}{2} - \frac{40\gamma^3 L_{\boldsymbol{\omega}}^2 L_F^2}{\mu^2\eta^2})\|\bar{\mathbf{p}}_t\|^2 - \frac{2\gamma\eta L_F^2}{\mu}\|\bar{\mathbf{d}}_t\|^2$$

$$+ (\frac{72\gamma L_F^4}{\mu^2 m} + \frac{2\gamma L_F^2}{m})\sum_{i=1}^m [\|\bar{\boldsymbol{\theta}}_t - \boldsymbol{\theta}_{i,t}\|^2 + \|\bar{\boldsymbol{\omega}}_t - \boldsymbol{\omega}_{i,t}\|^2]$$

$$+ 2\gamma\|\frac{1}{m}\sum_{i=1}^m \nabla_{\boldsymbol{\omega}}F_i(\boldsymbol{\theta}_{i,t}, \boldsymbol{\omega}_{i,t}) - \bar{\mathbf{v}}_t\|^2 + \frac{72L_F^2\gamma}{\mu^2}\|\frac{1}{m}\sum_{i=1}^m \nabla_{\boldsymbol{\theta}}F_i(\boldsymbol{\theta}_{i,t}, \boldsymbol{\omega}_{i,t}) - \bar{\mathbf{u}}_t\|^2$$

$$\overset{(a)}{\leq} -\frac{\gamma}{2}\|\nabla J(\bar{\boldsymbol{\theta}}_t)\|^2 - \gamma L_F^2\|\boldsymbol{\omega}_t^* - \bar{\boldsymbol{\omega}}_t\|^2 - (\frac{\gamma}{2} - \frac{L_J\gamma^2}{2} - \frac{40\gamma^3 L_{\boldsymbol{\omega}}^2 L_F^2}{\mu^2\eta^2})\|\bar{\mathbf{p}}_t\|^2 - \frac{2\gamma\eta L_F^2}{\mu}\|\bar{\mathbf{d}}_t\|^2$$

$$+ (\frac{72\gamma L_F^4}{\mu^2 m} + \frac{2\gamma L_F^2}{m})[\mathcal{E}(\boldsymbol{\theta}_t) + \mathcal{E}(\boldsymbol{\omega}_t)] + \frac{2\gamma}{m}\|\nabla_{\boldsymbol{\omega}}F_t - \mathbf{v}_t\|^2 + \frac{72L_F^2\gamma}{m\mu^2}\|\nabla_{\boldsymbol{\theta}}F_t - \mathbf{u}_t\|^2, \tag{27}$$

where (a) due to $\|\frac{1}{m}\sum_{i=1}^m \mathbf{x}_{i,t} - \bar{x}_t\|^2 \leq \frac{1}{m}\sum_{i=1}^m \|\mathbf{x}_{i,t} - \bar{x}_t\|^2$.

Telescoping the above inequality, we have the stated result. $\qquad\square$

Next, we prove the contraction of iterations in the following lemma, which is useful in analyzing the decentralized gradient tracking algorithms.

**Lemma 4** (Iterates Contraction). *The following contraction properties of the iterates hold:*

$$\|\boldsymbol{\theta}_t - \mathbf{1} \otimes \bar{\boldsymbol{\theta}}_t\|^2 \leq (1 + c_1)\lambda^2 \|\boldsymbol{\theta}_{t-1} - \mathbf{1} \otimes \bar{\boldsymbol{\theta}}_{t-1}\|^2 + (1 + \frac{1}{c_1})\gamma^2 \|\mathbf{p}_{t-1} - \mathbf{1} \otimes \bar{\mathbf{p}}_{t-1}\|^2, \quad (28)$$

$$\|\boldsymbol{\omega}_t - \mathbf{1} \otimes \bar{\boldsymbol{\omega}}_t\|^2 \leq (1 + c_2)\lambda^2 \|\boldsymbol{\omega}_{t-1} - \mathbf{1} \otimes \bar{\boldsymbol{\omega}}_{t-1}\|^2 + (1 + \frac{1}{c_2})\eta^2 \|\mathbf{d}_{t-1} - \mathbf{1} \otimes \bar{\mathbf{d}}_{t-1}\|^2, \quad (29)$$

$$\|\mathbf{p}_t - \mathbf{1} \otimes \bar{\mathbf{p}}_t\|^2 \leq (1 + c_1)\lambda^2 \|\mathbf{p}_{t-1} - \mathbf{1} \otimes \bar{\mathbf{p}}_{t-1}\|^2 + (1 + \frac{1}{c_1})\|\mathbf{v}_t - \mathbf{v}_{t-1}\|^2, \quad (30)$$

$$\|\mathbf{d}_t - \mathbf{1} \otimes \bar{\mathbf{d}}_t\|^2 \leq (1 + c_2)\lambda^2 \|\mathbf{d}_{t-1} - \mathbf{1} \otimes \bar{\mathbf{d}}_{t-1}\|^2 + (1 + \frac{1}{c_2})\|\mathbf{u}_t - \mathbf{u}_{t-1}\|^2, \quad (31)$$

*where $c_1$ and $c_2$ are arbitrary positive constants. Additionally, we have*

$$\|\boldsymbol{\theta}_t - \boldsymbol{\theta}_{t-1}\|^2 \leq 8\|(\boldsymbol{\theta}_{t-1} - \mathbf{1} \otimes \bar{\boldsymbol{\theta}}_{t-1})\|^2 + 4\gamma^2 \|\mathbf{p}_{t-1} - \mathbf{1} \otimes \bar{\mathbf{p}}_{t-1}\|^2 + 4\gamma^2 m \|\bar{\mathbf{p}}_{t-1}\|^2, \quad (32)$$

$$\|\boldsymbol{\omega}_t - \boldsymbol{\omega}_{t-1}\|^2 \leq 8\|(\boldsymbol{\omega}_{t-1} - \mathbf{1} \otimes \bar{\boldsymbol{\omega}}_{t-1})\|^2 + 4\gamma^2 \|\mathbf{d}_{t-1} - \mathbf{1} \otimes \bar{\mathbf{d}}_{t-1}\|^2 + 4\gamma^2 m \|\bar{\mathbf{d}}_{t-1}\|^2. \quad (33)$$

*Proof.* First for the iterates $\boldsymbol{\theta}_t$, we have the following contraction:

$$\|\widetilde{\mathbf{M}}\boldsymbol{\theta}_t - \mathbf{1} \otimes \bar{\boldsymbol{\theta}}_t\|^2 = \|\widetilde{\mathbf{M}}(\boldsymbol{\theta}_t - \mathbf{1} \otimes \bar{\boldsymbol{\theta}}_t)\|^2 \leq \lambda^2 \|\boldsymbol{\theta}_t - \mathbf{1} \otimes \bar{\boldsymbol{\theta}}_t\|^2. \quad (34)$$

This is because $\boldsymbol{\theta}_t - \mathbf{1} \otimes \boldsymbol{\theta}_t$ is orthogonal to $\mathbf{1}$, which is the eigenvector corresponding to the largest eigenvalue of $\widetilde{\mathbf{M}}$, and $\lambda = \max\{|\lambda_2|, |\lambda_m|\}$. Recall that $\bar{\boldsymbol{\theta}}_t = \bar{\boldsymbol{\theta}}_{t-1} - \gamma \bar{\mathbf{p}}_{t-1}$, hence,

$$\begin{aligned}
\|\boldsymbol{\theta}_t - \mathbf{1} \otimes \bar{\boldsymbol{\theta}}_t\|^2 &= \|\widetilde{\mathbf{M}}\boldsymbol{\theta}_{t-1} - \gamma \mathbf{p}_{t-1} - \mathbf{1} \otimes (\bar{\boldsymbol{\theta}}_{t-1} - \gamma \bar{\mathbf{p}}_{t-1})\|^2 \\
&\leq (1 + c_1)\|\widetilde{\mathbf{M}}\boldsymbol{\theta}_{t-1} - \mathbf{1} \otimes \bar{\boldsymbol{\theta}}_{t-1}\|^2 + (1 + \frac{1}{c_1})\gamma^2 \|\mathbf{p}_{t-1} - \mathbf{1} \otimes \bar{\mathbf{p}}_{t-1}\|^2 \\
&\leq (1 + c_1)\lambda^2 \|\boldsymbol{\theta}_{t-1} - \mathbf{1} \otimes \bar{\boldsymbol{\theta}}_{t-1}\|^2 + (1 + \frac{1}{c_1})\gamma^2 \|\mathbf{p}_{t-1} - \mathbf{1} \otimes \bar{\mathbf{p}}_{t-1}\|^2. \quad (35)
\end{aligned}$$

Similarly, we have the result for $\boldsymbol{\omega}_t$,

$$\|\boldsymbol{\omega}_t - \mathbf{1} \otimes \bar{\boldsymbol{\omega}}_t\|^2 \leq (1 + c_2)\lambda^2 \|\boldsymbol{\omega}_{t-1} - \mathbf{1} \otimes \bar{\boldsymbol{\omega}}_{t-1}\|^2 + (1 + \frac{1}{c_2})\eta^2 \|\mathbf{d}_{t-1} - \mathbf{1} \otimes \bar{\mathbf{d}}_{t-1}\|^2. \quad (36)$$

For $\mathbf{p}_t$, we have

$$\begin{aligned}
\|\mathbf{p}_t - \mathbf{1} \otimes \bar{\mathbf{p}}_t\|^2 &= \|\widetilde{\mathbf{M}}\mathbf{p}_{t-1} + \mathbf{v}_t - \mathbf{v}_{t-1} - \mathbf{1} \otimes (\bar{\mathbf{p}}_{t-1} + \bar{\mathbf{v}}_t - \bar{\mathbf{v}}_{t-1})\| \\
&\leq (1 + c_1)\lambda^2 \|\mathbf{p}_{t-1} - \mathbf{1} \otimes \bar{\mathbf{p}}_{t-1}\|^2 + (1 + \frac{1}{c_1})\|\mathbf{v}_t - \mathbf{v}_{t-1} - \mathbf{1} \otimes (\bar{\mathbf{v}}_t - \bar{\mathbf{v}}_{t-1})\|^2 \\
&\leq (1 + c_1)\lambda^2 \|\mathbf{p}_{t-1} - \mathbf{1} \otimes \bar{\mathbf{p}}_{t-1}\|^2 + (1 + \frac{1}{c_1})\|(\mathbf{I} - \frac{1}{n}(\mathbf{1}\mathbf{1}^\top) \otimes \mathbf{I})(\mathbf{v}_t - \mathbf{v}_{t-1})\|^2 \\
&\overset{(a)}{\leq} (1 + c_1)\lambda^2 \|\mathbf{p}_{t-1} - \mathbf{1} \otimes \bar{\mathbf{p}}_{t-1}\|^2 + (1 + \frac{1}{c_1})\|\mathbf{v}_t - \mathbf{v}_{t-1}\|^2, \quad (37)
\end{aligned}$$

where (a) is due to $\|\mathbf{I} - \frac{1}{m}(\mathbf{1}\mathbf{1}^\top) \otimes \mathbf{I}\| \leq 1$. Similarly, we have

$$\|\mathbf{d}_t - \mathbf{1} \otimes \bar{\mathbf{d}}_t\|^2 \leq (1 + c_2)\lambda^2 \|\mathbf{d}_{t-1} - \mathbf{1} \otimes \bar{\mathbf{d}}_{t-1}\|^2 + (1 + \frac{1}{c_2})\|\mathbf{u}_t - \mathbf{u}_{t-1}\|^2. \quad (38)$$

According to the update, we have

$$\begin{aligned}
\|\boldsymbol{\theta}_t - \boldsymbol{\theta}_{t-1}\|^2 &= \|\widetilde{\mathbf{M}}\boldsymbol{\theta}_{t-1} - \gamma \mathbf{p}_{t-1} - \boldsymbol{\theta}_{t-1}\|^2 \\
&= \|(\widetilde{\mathbf{M}} - \mathbf{I})\boldsymbol{\theta}_{t-1} - \gamma \mathbf{p}_{t-1}\|^2 \leq 2\|(\widetilde{\mathbf{M}} - \mathbf{I})\boldsymbol{\theta}_{t-1}\|^2 + 2\gamma^2 \|\mathbf{p}_{t-1}\|^2 \\
&= 2\|(\widetilde{\mathbf{M}} - \mathbf{I})(\boldsymbol{\theta}_{t-1} - \mathbf{1} \otimes \bar{\boldsymbol{\theta}}_{t-1})\|^2 + 2\gamma^2 \|\mathbf{p}_{t-1}\|^2 \\
&\leq 8\|(\boldsymbol{\theta}_{t-1} - \mathbf{1} \otimes \bar{\boldsymbol{\theta}}_{t-1})\|^2 + 4\gamma^2 \|\mathbf{p}_{t-1} - \mathbf{1} \otimes \bar{\mathbf{p}}_{t-1}\|^2 + 4\gamma^2 m \|\bar{\mathbf{p}}_{t-1}\|^2
\end{aligned}$$

$$=8\mathcal{E}(\boldsymbol{\theta}_{t-1})+4\gamma^2\mathcal{E}(\mathbf{p}_{t-1})+4\gamma^2 m\|\bar{\mathbf{p}}_{t-1}\|^2, \tag{39}$$

and also

$$\|\boldsymbol{\omega}_t-\boldsymbol{\omega}_{t-1}\|^2 \leq 8\mathcal{E}(\boldsymbol{\omega}_{t-1})+4\eta^2\mathcal{E}(\mathbf{d}_{t-1})+4\eta^2 m\|\bar{\mathbf{d}}_{t-1}\|^2. \tag{40}$$

$\square$

**Lemma 5** (Differential Bound on Estimator for GT-GDA). *Under Assumption 1, the following inequalities hold:*

$$\sum_{t=1}^{T}\mathbb{E}\|\mathbf{v}_t-\mathbf{v}_{t-1}\|^2 \leq \sum_{t=1}^{T}3L_F^2\mathbb{E}\|\boldsymbol{\theta}_{t-1}-\boldsymbol{\theta}_t\|^2+3L_F^2\mathbb{E}\|\boldsymbol{\omega}_{t-1}-\boldsymbol{\omega}_t\|^2, \tag{41}$$

$$\sum_{t=1}^{T}\mathbb{E}\|\mathbf{u}_t-\mathbf{u}_{t-1}\|^2 \leq \sum_{t=1}^{T}3L_F^2\mathbb{E}\|\boldsymbol{\theta}_{t-1}-\boldsymbol{\theta}_t\|^2+3L_F^2\mathbb{E}\|\boldsymbol{\omega}_{t-1}-\boldsymbol{\omega}_t\|^2. \tag{42}$$

*Proof.* For $\|\mathbf{v}_t-\mathbf{v}_{t-1}\|^2$, we have

$$\mathbb{E}\|\mathbf{v}_t-\mathbf{v}_{t-1}\|^2 = \mathbb{E}\|\mathbf{v}_t-\nabla_{\boldsymbol{\theta}}\mathbf{F}_t+\nabla_{\boldsymbol{\theta}}\mathbf{F}_t-\nabla_{\boldsymbol{\theta}}\mathbf{F}_{t-1}+\nabla_{\boldsymbol{\theta}}\mathbf{F}_{t-1}-\mathbf{v}_{t-1}\|^2$$

$$\leq 3\mathbb{E}\|\mathbf{v}_t-\nabla_{\boldsymbol{\theta}}F_t\|^2+3\mathbb{E}\|\nabla_{\boldsymbol{\theta}}F_t-\nabla_{\boldsymbol{\theta}}F_{t-1}\|^2+3\mathbb{E}\|\nabla_{\boldsymbol{\theta}}F_{t-1}-\mathbf{v}_{t-1}\|^2$$

$$\leq 3L_F\mathbb{E}\|\boldsymbol{\theta}_{t-1}-\boldsymbol{\theta}_t\|^2+3L_F^2\mathbb{E}\|\boldsymbol{\omega}_{t-1}-\boldsymbol{\omega}_t\|^2. \tag{43}$$

Thus, we have: $\sum_{t=1}^{T}\mathbb{E}\|\mathbf{v}_t-\mathbf{v}_{t-1}\|^2 \leq \sum_{t=1}^{T}3L_F^2\mathbb{E}\|\boldsymbol{\theta}_{t-1}-\boldsymbol{\theta}_t\|^2+3L_F^2\mathbb{E}\|\boldsymbol{\omega}_{t-1}-\boldsymbol{\omega}_t\|^2$, and similarly, $\sum_{t=1}^{T}\mathbb{E}\|\mathbf{u}_t-\mathbf{u}_{t-1}\|^2 \leq \sum_{t=1}^{T}3L_F^2\mathbb{E}\|\boldsymbol{\theta}_{t-1}-\boldsymbol{\theta}_t\|^2+3L_F^2\mathbb{E}\|\boldsymbol{\omega}_{t-1}-\boldsymbol{\omega}_t\|^2$. $\square$

Now, we show the final step for proving Theorem 1. With Lemmas 3-5 and the defined potential function, we have:

$$\mathbb{E}\mathfrak{P}_{T+1}-\mathfrak{P}_0 \leq -\frac{\gamma}{2}\sum_{t=0}^{T}\mathbb{E}\|\nabla J(\bar{\boldsymbol{\theta}}_t)\|^2-\gamma L_f^2\sum_{t=0}^{T}\mathbb{E}\|\boldsymbol{\omega}_t^*-\bar{\boldsymbol{\omega}}_t\|^2$$

$$-\left(\frac{\gamma}{2}-\frac{L_J\gamma^2}{2}-\frac{40\gamma^3 L_{\boldsymbol{\omega}}^2 L_F^2}{\mu^2\eta^2}\right)\sum_{t=0}^{T}\mathbb{E}\|\bar{\mathbf{p}}_t\|^2-\frac{2\gamma\eta L_F^2}{\mu}\sum_{t=0}^{T}\mathbb{E}\|\bar{\mathbf{d}}_t\|^2$$

$$-\left(1-(1+c_1)\lambda^2-\frac{72\gamma L_F^4}{\mu^2}-2\gamma L_F^2\right)\sum_{t=0}^{T}\frac{\mathcal{E}(\boldsymbol{\theta}_t)}{m}$$

$$-\left(1-(1+c_2)\lambda^2-\frac{72\gamma L_F^4}{\mu^2}-2\gamma L_F^2\right)\sum_{t=0}^{T}\frac{\mathcal{E}(\boldsymbol{\omega}_t)}{m}$$

$$-\left(1-(1+c_1)\lambda^2-(1+\frac{1}{c_1})\gamma\right)\gamma\sum_{t=0}^{T}\frac{\mathcal{E}(\mathbf{p}_t)}{m}$$

$$-\left(1-(1+c_2)\lambda^2-(1+\frac{1}{c_2})\eta\right)\eta\sum_{t=0}^{T}\frac{\mathcal{E}(\mathbf{d}_t)}{m}$$

$$+\underbrace{\frac{2\gamma}{m}\sum_{t=0}^{T}\mathbb{E}\|\nabla_{\boldsymbol{\omega}}F_t-\mathbf{v}_t\|^2+\frac{72L_F^2\gamma}{m\mu^2}\sum_{t=0}^{T}\mathbb{E}\|\nabla_{\boldsymbol{\theta}}F_t-\mathbf{u}_t\|^2}_{R_1}$$

$$+\underbrace{(1+\frac{1}{c_1})\frac{\gamma}{m}\sum_{t=1}^{T}\mathbb{E}\|\mathbf{v}_t-\mathbf{v}_{t-1}\|^2+(1+\frac{1}{c_2})\frac{\eta}{m}\sum_{t=1}^{T}\mathbb{E}\|\mathbf{u}_t-\mathbf{u}_{t-1}\|^2}_{R_2}. \tag{44}$$

First, we have $R_1=0$ because of the full gradient evaluation. $R_2$ can be bounded by

$$R_2 = (1+\frac{1}{c_1})\frac{\gamma}{m}\sum_{t=1}^{T}\mathbb{E}\|\mathbf{v}_t-\mathbf{v}_{t-1}\|^2+(1+\frac{1}{c_2})\frac{\eta}{m}\sum_{t=1}^{T}\mathbb{E}\|\mathbf{u}_t-\mathbf{u}_{t-1}\|^2$$

$$\stackrel{(a)}{\leq} \left( (1+\frac{1}{c_1})\frac{\gamma}{m} + (1+\frac{1}{c_2})\frac{\eta}{m} \right) \sum_{t=1}^{T} 3L_F^2 \left( \mathbb{E}\|\boldsymbol{\theta}_{t-1} - \boldsymbol{\theta}_t\|^2 + \mathbb{E}\|\boldsymbol{\omega}_{t-1} - \boldsymbol{\omega}_t\|^2 \right)$$

$$\stackrel{(b)}{\leq} 3L_F^2 \left( (1+\frac{1}{c_1})\frac{\gamma}{m} + (1+\frac{1}{c_2})\frac{\eta}{m} \right) \sum_{t=1}^{T} \left( 8\mathcal{E}(\boldsymbol{\theta}_{t-1}) + 4\gamma^2\mathcal{E}(\mathbf{p}_{t-1}) + 4\gamma^2 m\|\bar{\mathbf{p}}_{t-1}\|^2 \right)$$

$$+ 3L_F^2 \left( (1+\frac{1}{c_1})\frac{\gamma}{m} + (1+\frac{1}{c_2})\frac{\eta}{m} \right) \sum_{t=1}^{T} \left( 8\mathcal{E}(\boldsymbol{\omega}_{t-1}) + 4\eta^2\mathcal{E}(\mathbf{d}_{t-1}) + 4\eta^2 m\|\bar{\mathbf{d}}_{t-1}\|^2 \right), \quad (45)$$

where (a) is from Lemma 5 and (b) is from Lemma 4.

Thus, we have

$$\mathbb{E}\mathfrak{P}_{T+1} - \mathfrak{P}_0 \leq -\frac{\gamma}{2}\sum_{t=0}^{T}\mathbb{E}\|\nabla J(\bar{\boldsymbol{\theta}}_t)\|^2 - \gamma L_f^2 \sum_{t=0}^{T}\mathbb{E}\|\boldsymbol{\omega}_t^* - \bar{\boldsymbol{\omega}}_t\|^2$$

$$- \left( \frac{\gamma}{2} - \frac{L_J\gamma^2}{2} - \frac{40\gamma^3 L_{\boldsymbol{\omega}}^2 L_F^2}{\mu^2\eta^2} - 12\gamma^2 L_F^2\left((1+\frac{1}{c_1})\gamma + (1+\frac{1}{c_2})\eta\right) \right) \sum_{t=0}^{T}\mathbb{E}\|\bar{\mathbf{p}}_t\|^2$$

$$- \left( \frac{2\gamma\eta L_F^2}{\mu} - 12\eta^2 L_F^2\left((1+\frac{1}{c_1})\gamma + (1+\frac{1}{c_2})\eta\right) \right) \sum_{t=0}^{T}\mathbb{E}\|\bar{\mathbf{d}}_t\|^2$$

$$- \left( 1 - (1+c_1)\lambda^2 - \frac{72\gamma L_F^4}{\mu^2} - 2\gamma L_F^2 - 24L_F^2\left((1+\frac{1}{c_1})\gamma + (1+\frac{1}{c_2})\eta\right) \right) \sum_{t=0}^{T}\frac{\mathcal{E}(\boldsymbol{\theta}_t)}{m}$$

$$- \left( 1 - (1+c_2)\lambda^2 - \frac{72\gamma L_F^4}{\mu^2} - 2\gamma L_F^2 - 24L_F^2\left((1+\frac{1}{c_1})\gamma + (1+\frac{1}{c_2})\eta\right) \right) \sum_{t=0}^{T}\frac{\mathcal{E}(\boldsymbol{\omega}_t)}{m}$$

$$- \left( 1 - (1+c_1)\lambda^2 - (1+\frac{1}{c_1})\gamma - 12\gamma L_F^2\left((1+\frac{1}{c_1})\gamma + (1+\frac{1}{c_2})\eta\right) \right)\gamma \sum_{t=0}^{T}\frac{\mathcal{E}(\mathbf{p}_t)}{m}$$

$$- \left( 1 - (1+c_2)\lambda^2 - (1+\frac{1}{c_2})\eta - 12\eta L_F^2\left((1+\frac{1}{c_1})\gamma + (1+\frac{1}{c_2})\eta\right) \right)\eta \sum_{t=0}^{T}\frac{\mathcal{E}(\mathbf{d}_t)}{m}. \quad (46)$$

Choosing $c_1 = c_2 = 1/\lambda - 1$, we have

$$\mathbb{E}\mathfrak{P}_{T+1} - \mathfrak{P}_0 \leq -\frac{\gamma}{2}\sum_{t=0}^{T}\mathbb{E}\|\nabla J(\bar{\boldsymbol{\theta}}_t)\|^2 - \gamma L_f^2 \sum_{t=0}^{T}\mathbb{E}\|\boldsymbol{\omega}_t^* - \bar{\boldsymbol{\omega}}_t\|^2 - C_{\bar{\mathbf{p}}}\gamma \sum_{t=0}^{T}\mathbb{E}\|\bar{\mathbf{p}}_t\|^2$$

$$- C_{\bar{\mathbf{d}}}\gamma\eta \sum_{t=0}^{T}\mathbb{E}\|\bar{\mathbf{d}}_t\|^2 - C_{\boldsymbol{\theta}}\sum_{t=0}^{T}\frac{\mathcal{E}(\boldsymbol{\theta}_t)}{m} - C_{\boldsymbol{\omega}}\sum_{t=0}^{T}\frac{\mathcal{E}(\boldsymbol{\omega}_t)}{m} - C_{\mathbf{p}}\gamma \sum_{t=0}^{T}\frac{\mathcal{E}(\mathbf{p}_t)}{m} - C_{\mathbf{d}}\eta \sum_{t=0}^{T}\frac{\mathcal{E}(\mathbf{d}_t)}{m}, \quad (47)$$

where the constants are

$$C_{\bar{\mathbf{p}}} = \frac{1}{2} - \frac{L_J\gamma}{2} - \frac{40\gamma^2 L_{\boldsymbol{\omega}}^2 L_F^2}{\mu^2\eta^2} - \frac{12\gamma L_F^2(\gamma + \eta)}{1-\lambda}, \quad (48)$$

$$C_{\bar{\mathbf{d}}} = \frac{2L_F^2}{\mu} - \frac{12\eta L_F^2(1+\eta/\gamma)}{1-\lambda}, \quad (49)$$

$$C_{\boldsymbol{\theta}} = C_{\boldsymbol{\omega}} = 1 - \lambda - \frac{72\gamma L_F^4}{\mu^2} - 2\gamma L_F^2 - \frac{24L_F^2(\gamma + \eta)}{1-\lambda}, \quad (50)$$

$$C_{\mathbf{p}} = 1 - \lambda - \frac{\gamma}{1-\lambda} - \frac{12\gamma L_F^2(\gamma + \eta)}{1-\lambda}, \quad (51)$$

$$C_{\mathbf{d}} = 1 - \lambda - \frac{\eta}{1-\lambda} - \frac{12\eta L_F^2(\gamma + \eta)}{1-\lambda}. \quad (52)$$

To ensure $C_{\bar{\mathbf{p}}} \geq 0$, we have

$$C_{\bar{\mathbf{p}}} \stackrel{(a)}{\geq} \frac{1}{4} - \frac{L_J\gamma}{2} - \frac{12\gamma^2 L_F^2(1+\eta/\gamma)}{1-\lambda}$$

$$\overset{(b)}{\geq} \frac{1}{4} - \frac{(L_f + L_f^2/\mu)\gamma}{2} - \frac{(1-\lambda)\gamma}{2} \overset{(c)}{\geq} 0, \tag{53}$$

where (a) follows from $\kappa := \gamma/\eta \leq \mu^2/13L_F^2$ and Lemma 9, (b) is due to $\gamma \leq (1-\lambda)^2/24L_F^2(1 + 1/\kappa)$ and Lemma 11, and (c) follows from $\gamma \leq 1/2\big((L_F + L_F^2/\mu) + (1-\lambda)\big)$. By setting $\eta \leq (1-\lambda)/6\mu(1+1/\kappa)$, we have $C_{\bar{\mathbf{d}}} \geq 0$. By setting $\gamma \leq (1-\lambda)/(\frac{1}{2} + \frac{72L_F^4}{\mu^2} + 2L_F^2 + \frac{24L_F^2(1+1/\kappa)}{1-\lambda})$, we have $C_{\boldsymbol{\theta}} = C_{\boldsymbol{\omega}} \geq \gamma/2$. To ensure $C_{\mathbf{p}} \geq 0$,

$$C_{\mathbf{p}} = 1 - \lambda - \frac{\gamma}{1-\lambda} - \frac{12\gamma^2 L_F^2(1 + \eta/\gamma)}{1-\lambda}$$
$$\overset{(a)}{\geq} 1 - \lambda - \frac{\gamma}{1-\lambda} - \frac{(1-\lambda)\gamma}{2} \overset{(b)}{\geq} 0, \tag{54}$$

where (a) is by $\gamma \leq (1-\lambda)^2/24L_F^2(1+1/\kappa)$ and (b) is by $\gamma \leq 1/\big(1/2 + 1/(1-\lambda)^2\big)$. Similarly, with $\eta \leq 1/\big(1/2 + 1/(1-\lambda)^2\big)$, we have $C_{\mathbf{d}} \geq 0$.

To summarize, we need the following conditions to ensure $C_{\bar{\mathbf{p}}} \geq 0$, $C_{\bar{\mathbf{d}}} \geq 0$, $C_{\mathbf{p}} \geq 0$, $C_{\mathbf{d}} \geq 0$, $C_{\boldsymbol{\theta}} \geq \gamma/2$, $C_{\boldsymbol{\omega}} \geq \gamma/2$,

$$\kappa = \gamma/\eta \leq \mu^2/13L_F^2 \tag{55}$$

$$\gamma \leq \min\left\{\frac{1}{2}\Big(L_F + \frac{L_F^2}{\mu} + (1-\lambda)\Big), \frac{1-\lambda}{(\frac{1}{2} + \frac{72L_F^4}{\mu^2} + 2L_F^2 + \frac{24L_F^2(1+1/\kappa)}{1-\lambda})}, \big(1/2 + 1/(1-\lambda)^2\big)^{-1}\right\}, \tag{56}$$

$$\eta \leq \min\left\{\frac{(1-\lambda)}{6\mu(1+1/\kappa)}, \big(1/2 + 1/(1-\lambda)^2\big)^{-1}\right\}, \tag{57}$$

which can be satisfied with

$$\kappa = \gamma/\eta \leq \mu^2/13L_F^2 \tag{58}$$

$$\eta \leq \min\left\{\frac{13L_F^2}{2\mu^2}\Big(L_F + \frac{L_F^2}{\mu} + (1-\lambda)\Big), \frac{26(1-\lambda)L_F^2}{(\mu^2 + 144L_F^4 + 4L_F^2\mu^2 + \frac{48\mu^2 L_F^2(1+1/\kappa)}{1-\lambda})},\right.$$
$$\left.\frac{(1-\lambda)}{6\mu(1+1/\kappa)}, \frac{13L_F^2}{\mu^2\big(1/2 + 1/(1-\lambda)^2\big)}\right\}. \tag{59}$$

Note that $\frac{(1-\lambda)}{6\mu(1+1/\kappa)} \leq \frac{1}{2L_F}$, which satisfies the step-size condition in Lemma 2 and Lemma 3.

Thus, with such parameter settings, we have

$$\frac{\gamma}{2}\sum_{t=0}^{T}\Big(\mathbb{E}\|\nabla J(\bar{\boldsymbol{\theta}}_t)\|^2 + 2L_f^2\mathbb{E}\|\boldsymbol{\omega}_t^* - \bar{\boldsymbol{\omega}}_t\|^2 + \frac{\mathcal{E}(\boldsymbol{\theta}_t)}{m} + \frac{\mathcal{E}(\boldsymbol{\omega}_t)}{m}\Big) \leq \mathbb{E}\mathfrak{P}_0 - \mathfrak{P}_{T+1}, \tag{60}$$

which yields the final result by multiplying $2/(T+1)\gamma$ on both sides. Note that $\mathfrak{P}_{T+1} \geq J(\bar{\boldsymbol{\theta}}_{T+1}) \geq J^*$, then we complete the proof.

## C  Proof for Theorem 3 and Theorem 4

In this section, we provide the proofs of Theorem 3-4 due to the the similar steps in their proofs. Under Assumption 2, we can easily obtain that funciton $F_i$ satisifies $L_f$-Lipschitz smoothness. Thus, we have the following modified descending result:

**Lemma 6.** *Under Assumption 1 and set $\eta \leq 1/2L_f$, the following inequality holds for GT-SRVR and GT-SRVRI:*

$$J(\bar{\boldsymbol{\theta}}_{T+1}) - J(\bar{\boldsymbol{\theta}}_0) + \frac{8\gamma L_f^2}{\mu\eta}\big[\|\bar{\boldsymbol{\omega}}_{T+1} - \boldsymbol{\omega}_{T+1}^*\|^2 - \|\boldsymbol{\omega}_0^* - \bar{\boldsymbol{\omega}}_0\|^2\big]$$
$$\leq -\frac{\gamma}{2}\sum_{t=0}^{T}\|\nabla J(\bar{\boldsymbol{\theta}}_t)\|^2 - \gamma L_f^2\sum_{t=0}^{T}\|\boldsymbol{\omega}_t^* - \bar{\boldsymbol{\omega}}_t\|^2 - \Big(\frac{\gamma}{2} - \frac{L_J\gamma^2}{2} - \frac{40\gamma^3 L_\omega^2 L_f^2}{\mu^2\eta^2}\Big)\sum_{t=0}^{T}\|\bar{\mathbf{p}}_t\|^2 - \frac{2\gamma\eta L_f^2}{\mu}\sum_{t=0}^{T}\|\bar{\mathbf{d}}_t\|^2$$
$$+ \Big(\frac{72\gamma L_f^4}{\mu^2 m} + \frac{2\gamma L_f^2}{m}\Big)\sum_{t=0}^{T}[\mathcal{E}(\boldsymbol{\theta}_t) + \mathcal{E}(\boldsymbol{\omega}_t)] + \frac{2\gamma}{m}\sum_{t=0}^{T}\|\nabla_{\boldsymbol{\omega}}F_t - \mathbf{v}_t\|^2 + \frac{72L_f^2\gamma}{m\mu^2}\sum_{t=0}^{T}\|\nabla_{\boldsymbol{\theta}}F_t - \mathbf{u}_t\|^2.$$

Next, we bound the error of the gradient estimators as the follows:

**Lemma 7** (Error of Gradient Estimator). *Under Assumption 2, we have the following error bounds for the estimators:*

$$\sum_{t=0}^{T}\|\mathbf{v}_t - \nabla_{\boldsymbol{\theta}} F_t\|^2 \leq \sum_{t=1}^{T}\mathbb{E}\|\mathbf{v}_{(n_t-1)q} - \nabla_{\boldsymbol{\theta}} F(\boldsymbol{\theta}_{(n_t-1)q}, \boldsymbol{\omega}_{(n_t-1)q})\|^2$$
$$+ L_f^2\big(\|\boldsymbol{\theta}_t - \boldsymbol{\theta}_{t-1}\|^2 + \|\boldsymbol{\omega}_t - \boldsymbol{\omega}_{t-1}\|^2\big), \quad (61)$$

$$\sum_{t=0}^{T}\|\mathbf{u}_t - \nabla_{\boldsymbol{\omega}} F_t\|^2 \leq \sum_{t=1}^{T}\mathbb{E}\|\mathbf{u}_{(n_t-1)q} - \nabla_{\boldsymbol{\omega}} F(\boldsymbol{\theta}_{(n_t-1)q}, \boldsymbol{\omega}_{(n_t-1)q})\|^2$$
$$+ L_f^2\big(\|\boldsymbol{\theta}_t - \boldsymbol{\theta}_{t-1}\|^2 + \|\boldsymbol{\omega}_t - \boldsymbol{\omega}_{t-1}\|^2\big), \quad (62)$$

*where $n_t$ is the largest positive integer that satisfies $(n_t - 1)q \leq t$.*

*Proof.* From the algorithm update, we have:

$$\|\underbrace{\mathbf{v}_{i,t} - \nabla_{\boldsymbol{\theta}} F_{i,t}}_{A_{i,t}}\|^2 = \|\mathbf{v}_{i,t-1} + \frac{1}{|\mathcal{S}_{i,t}|}\sum_{j\in\mathcal{S}_{i,t}}\nabla_{\boldsymbol{\theta}} f_{i,j}(\boldsymbol{\theta}_{i,t}, \boldsymbol{\omega}_{i,t}) - \nabla_{\boldsymbol{\theta}} f_{i,j}(\boldsymbol{\theta}_{i,t-1}, \boldsymbol{\omega}_{i,t-1}) - \nabla_{\boldsymbol{\theta}} F_{i,t}\|^2$$

$$= \|\underbrace{\mathbf{v}_{i,t-1} - \nabla_{\boldsymbol{\theta}} F_{i,t-1}}_{A_{i,t-1}} + \underbrace{\frac{1}{|\mathcal{S}_{i,t}|}\sum_{j\in\mathcal{S}_{i,t}}\nabla_{\boldsymbol{\theta}} f_{i,t}(\boldsymbol{\theta}_{i,t}, \boldsymbol{\omega}_{i,t}) - \nabla_{\boldsymbol{\theta}} f_{i,t}(\boldsymbol{\theta}_{i,t-1}, \boldsymbol{\omega}_{i,t-1}) + \nabla_{\boldsymbol{\theta}} F_{i,t-1} - \nabla_{\boldsymbol{\theta}} F_{i,t}}_{B_{i,t}}\|^2$$

$$= \|A_{i,t-1}\|^2 + \|B_{i,t}\|^2 + 2\langle A_{i,t-1}, B_{i,t}\rangle. \quad (63)$$

Note that $\mathbb{E}_t[B_{i,t}] = 0$, where the expectation is taken over the randomness in $t$th iteration. Thus,

$$\mathbb{E}_t\|A_{i,t}\|^2 = \|A_{i,t-1}\|^2 + \mathbb{E}_t\|B_{i,t}\|^2. \quad (64)$$

Also, with $|\mathcal{S}_{i,t}| = q$, we have

$$\mathbb{E}_t\|B_{i,t}\|^2 = \mathbb{E}_t\|\frac{1}{|\mathcal{S}_{i,t}|}\sum_{j\in\mathcal{S}_{i,t}}\nabla_{\boldsymbol{\theta}} f_{i,j}(\boldsymbol{\theta}_{i,t}, \boldsymbol{\omega}_{i,t}) - \nabla_{\boldsymbol{\theta}} f_{i,j}(\boldsymbol{\theta}_{i,t-1}, \boldsymbol{\omega}_{i,t-1}) - \nabla_{\boldsymbol{\theta}} F_{i,t} + \nabla_{\boldsymbol{\theta}} F_{i,t-1}\|^2$$

$$\leq \frac{1}{|\mathcal{S}_{i,t}|^2}\sum_{j\in\mathcal{S}_{i,t}}\mathbb{E}_t\|\nabla_{\boldsymbol{\theta}} f_{i,j}(\boldsymbol{\theta}_{i,t}, \boldsymbol{\omega}_{i,t}) - \nabla_{\boldsymbol{\theta}} f_{i,j}(\boldsymbol{\theta}_{i,t-1}, \boldsymbol{\omega}_{i,t-1}) - \nabla_{\boldsymbol{\theta}} F_{i,t} + \nabla_{\boldsymbol{\theta}} F_{i,t-1}\|^2$$

$$\leq \frac{L_f^2}{q}\big(\|\boldsymbol{\theta}_{i,t} - \boldsymbol{\theta}_{i,t-1}\|^2 + \|\boldsymbol{\omega}_{i,t} - \boldsymbol{\omega}_{i,t-1}\|^2\big). \quad (65)$$

Taking full expectation and telescoping (65) over $t$ from $(n_t - 1)q + 1$ to $t$, where $t \leq n_t q - 1$, we have

$$\mathbb{E}\|A_t\|^2 \leq \mathbb{E}\|A_{t-1}\|^2 + \frac{L_f^2}{q}\mathbb{E}\big(\|\boldsymbol{\theta}_t - \boldsymbol{\theta}_{t-1}\|^2 + \|\boldsymbol{\omega}_t - \boldsymbol{\omega}_{t-1}\|^2\big)$$

$$\leq \mathbb{E}\|A_{(n_t-1)q}\|^2 + \sum_{r=(n_t-1)q+1}^{t}\frac{L_f^2}{q}\mathbb{E}\big(\|\boldsymbol{\theta}_r - \boldsymbol{\theta}_{r-1}\|^2 + \|\boldsymbol{\omega}_r - \boldsymbol{\omega}_{r-1}\|^2\big). \quad (66)$$

Thus, we have:

$$\sum_{k=0}^{t}\mathbb{E}\|A_k\|^2 = \sum_{k=0}^{q-1}\mathbb{E}\|A_k\|^2 + \cdots + \sum_{k=(n_t-1)q}^{t}\mathbb{E}\|A_k\|^2$$

$$\leq q\|A_0\|^2 + \sum_{k=1}^{q-1}\sum_{r=1}^{k}\frac{L_f^2}{q}\big(\|\boldsymbol{\theta}_r - \boldsymbol{\theta}_{r-1}\|^2 + \|\boldsymbol{\omega}_r - \boldsymbol{\omega}_{r-1}\|^2\big)$$

$$+ \cdots$$

$$+ \big(t - (n_t-1)q\big)\|A_{(n_t-1)q}\|^2 + \sum_{k=(n_t-1)q+1}^{t} \sum_{r=(n_t-1)q+1}^{k} \frac{L_f^2}{q}\big(\|\boldsymbol{\theta}_r - \boldsymbol{\theta}_{r-1}\|^2 + \|\boldsymbol{\omega}_r - \boldsymbol{\omega}_{r-1}\|^2\big)$$

$$\leq q\|A_0\|^2 + \sum_{r=1}^{q-1}\sum_{k=r}^{q-1} \frac{L_f^2}{q}\big(\|\boldsymbol{\theta}_r - \boldsymbol{\theta}_{r-1}\|^2 + \|\boldsymbol{\omega}_r - \boldsymbol{\omega}_{r-1}\|^2\big)$$

$$+\cdots$$

$$+ \big(t - (n_t-1)q\big)\|A_{(n_t-1)q}\|^2 + \sum_{r=(n_t-1)q+1}^{t} \sum_{k=r}^{t} \frac{L_f^2}{q}\big(\|\boldsymbol{\theta}_r - \boldsymbol{\theta}_{r-1}\|^2 + \|\boldsymbol{\omega}_r - \boldsymbol{\omega}_{r-1}\|^2\big)$$

$$\leq q\|A_0\|^2 + \sum_{r=1}^{q-1} L_f^2\big(\|\boldsymbol{\theta}_r - \boldsymbol{\theta}_{r-1}\|^2 + \|\boldsymbol{\omega}_r - \boldsymbol{\omega}_{r-1}\|^2\big)$$

$$+\cdots$$

$$+ \big(t - (n_t-1)q\big)\|A_{(n_t-1)q}\|^2 + \sum_{r=(n_t-1)q+1}^{t} L_f^2\big(\|\boldsymbol{\theta}_r - \boldsymbol{\theta}_{r-1}\|^2 + \|\boldsymbol{\omega}_r - \boldsymbol{\omega}_{r-1}\|^2\big)$$

$$= \sum_{r=0}^{t}\|A_{(n_r-1)q}\|^2 + \sum_{r=1}^{t} L_f^2\big(\|\boldsymbol{\theta}_r - \boldsymbol{\theta}_{r-1}\|^2 + \|\boldsymbol{\omega}_r - \boldsymbol{\omega}_{r-1}\|^2\big). \tag{67}$$

Thus, we have:

$$\sum_{t=0}^{T}\|\mathbf{v}_t - \nabla_{\boldsymbol{\theta}} F_t\|^2 \leq \sum_{t=0}^{T}\mathbb{E}\|\mathbf{v}_{(n_t-1)q} - \nabla_{\boldsymbol{\theta}} F_{(n_t-1)q}\|^2 + \sum_{t=1}^{T} L_f^2\big(\|\boldsymbol{\theta}_t - \boldsymbol{\theta}_{t-1}\|^2 + \|\boldsymbol{\omega}_t - \boldsymbol{\omega}_{t-1}\|^2\big) \tag{68}$$

Similarly, we have:

$$\sum_{t=0}^{T}\|\mathbf{u}_t - \nabla_{\boldsymbol{\omega}} F_t\|^2 \leq \sum_{t=0}^{T}\mathbb{E}\|\mathbf{u}_{(n_t-1)q} - \nabla_{\boldsymbol{\omega}} F_{(n_t-1)q}\|^2 + \sum_{t=1}^{T} L_f^2\big(\|\boldsymbol{\theta}_t - \boldsymbol{\theta}_{t-1}\|^2 + \|\boldsymbol{\omega}_t - \boldsymbol{\omega}_{t-1}\|^2\big). \tag{69}$$

$\square$

**Lemma 8** (Differential Bound on Estimator for GT-SRVR and GT-SRVRI). *Under Assumption 2, the following inequalities hold:*

$$\sum_{t=1}^{T}\mathbb{E}\|\mathbf{v}_t - \mathbf{v}_{t-1}\|^2 \leq \sum_{t=1}^{T} 4L_f^2\mathbb{E}\|\boldsymbol{\theta}_{t-1} - \boldsymbol{\theta}_t\|^2 + 4L_f^2\mathbb{E}\|\boldsymbol{\omega}_{t-1} - \boldsymbol{\omega}_t\|^2 + \sum_{t=0}^{T} 6\mathbb{E}\|\mathbf{v}_{(n_t-1)q} - \nabla_{\boldsymbol{\theta}} F_{(n_t-1)q}\|^2,$$

$$\sum_{t=1}^{T}\mathbb{E}\|\mathbf{u}_t - \mathbf{u}_{t-1}\|^2 \leq \sum_{t=1}^{T} 4L_f^2\mathbb{E}\|\boldsymbol{\theta}_{t-1} - \boldsymbol{\theta}_t\|^2 + 4L_f^2\mathbb{E}\|\boldsymbol{\omega}_{t-1} - \boldsymbol{\omega}_t\|^2 + \sum_{t=0}^{T} 6\mathbb{E}\|\mathbf{u}_{(n_t-1)q} - \nabla_{\boldsymbol{\omega}} F_{(n_t-1)q}\|^2.$$

*Proof.* For $\|\mathbf{v}_t - \mathbf{v}_{t-1}\|^2$, we have i) when $t \in \big((n_t-1)q, n_tq-1\big] \cap \mathbb{Z}$,

$$\mathbb{E}\|\mathbf{v}_t - \mathbf{v}_{t-1}\|^2 = \sum_{i=1}^{m}\mathbb{E}\|\frac{1}{|\mathcal{S}_{i,t}|}\sum_{j\in\mathcal{S}_{i,t}} \nabla_{\boldsymbol{\theta}} f_{i,j}(\boldsymbol{\theta}_{i,t}, \boldsymbol{\omega}_{i,t}) - \nabla_{\boldsymbol{\theta}} f_{i,k}(\boldsymbol{\theta}_{i,t-1}, \boldsymbol{\omega}_{i,t-1})\|^2$$

$$\leq \frac{1}{|\mathcal{S}_{i,t}|^2}\sum_{i=1}^{m}\sum_{j\in\mathcal{S}_{i,t}}\mathbb{E}\|\nabla_{\boldsymbol{\theta}} f_{i,j}(\boldsymbol{\theta}_{i,t}, \boldsymbol{\omega}_{i,t}) - \nabla_{\boldsymbol{\theta}} f_{i,j}(\boldsymbol{\theta}_{i,t-1}, \boldsymbol{\omega}_{i,t-1})\|^2$$

$$\overset{(a)}{\leq} L_f^2\sum_{i=1}^{m}\mathbb{E}\|\boldsymbol{\theta}_{i,t-1} - \boldsymbol{\theta}_{i,t}\|^2 + L_f^2\sum_{i=1}^{m}\mathbb{E}\|\boldsymbol{\omega}_{i,t-1} - \boldsymbol{\omega}_{i,t}\|^2$$

$$= L_f^2\mathbb{E}\|\boldsymbol{\theta}_{t-1} - \boldsymbol{\theta}_t\|^2 + L_f^2\mathbb{E}\|\boldsymbol{\omega}_{t-1} - \boldsymbol{\omega}_t\|^2, \tag{70}$$

where (a) is by $q \geq 1$ and Assumption 2.

ii) when $t = n_t q$ and $t > 0$,

$$\mathbb{E}\|\mathbf{v}_t - \mathbf{v}_{t-1}\|^2 = \mathbb{E}\|\mathbf{v}_t - \nabla_{\boldsymbol{\theta}}F_t + \nabla_{\boldsymbol{\theta}}F_t - \nabla_{\boldsymbol{\theta}}F_{t-1} + \nabla_{\boldsymbol{\theta}}F_{t-1} - \mathbf{v}_{t-1}\|^2$$

$$\leq 3\mathbb{E}\|\mathbf{v}_t - \nabla_{\boldsymbol{\theta}}F_t\|^2 + 3\mathbb{E}\|\nabla_{\boldsymbol{\theta}}F_t - \nabla_{\boldsymbol{\theta}}F_{t-1}\|^2 + 3\mathbb{E}\|\nabla_{\boldsymbol{\theta}}F_{t-1} - \mathbf{v}_{t-1}\|^2$$

$$\overset{(a)}{\leq} 3\mathbb{E}\|\mathbf{v}_t - \nabla_{\boldsymbol{\theta}}F_t\|^2 + 3\mathbb{E}\|\nabla_{\boldsymbol{\theta}}F_{t-1} - \mathbf{v}_{t-1}\|^2 + 3L_f^2\mathbb{E}\|\boldsymbol{\theta}_{t-1} - \boldsymbol{\theta}_t\|^2 + 3L_f^2\mathbb{E}\|\boldsymbol{\omega}_{t-1} - \boldsymbol{\omega}_t\|^2$$

$$\overset{(b)}{\leq} 3\mathbb{E}\|\mathbf{v}_{n_t q} - \nabla_{\boldsymbol{\theta}}F_{n_t q}\|^2 + 3\mathbb{E}\|\mathbf{v}_{(n_t-1)q} - \nabla_{\boldsymbol{\theta}}F_{(n_t-1)q}\|^2$$

$$+ \sum_{r=(n_t-1)q+1}^{n_t q-1} \frac{3L_f^2}{q}\mathbb{E}\big(\|\boldsymbol{\theta}_r - \boldsymbol{\theta}_{r-1}\|^2 + \|\boldsymbol{\omega}_r - \boldsymbol{\omega}_{r-1}\|^2\big)$$

$$+ 3L_f^2\mathbb{E}\|\boldsymbol{\theta}_{n_t q-1} - \boldsymbol{\theta}_{n_t q}\|^2 + 3L_f^2\mathbb{E}\|\boldsymbol{\omega}_{n_t q-1} - \boldsymbol{\omega}_{n_t q}\|^2, \tag{71}$$

where (a) is by Assumption 2, and (b) is by setting $t = n_t q$ and Lemma 7.

Telescoping (71) from $r = (n_t - 1)q + 1$ to $n_t q$,

$$\sum_{r=(n_t-1)q+1}^{n_t q} \mathbb{E}\|\mathbf{v}_r - \mathbf{v}_{r-1}\|^2 \leq \sum_{r=(n_t-1)q+1}^{n_t q-1} L_f^2\mathbb{E}\|\boldsymbol{\theta}_{r-1} - \boldsymbol{\theta}_r\|^2 + L_f^2\mathbb{E}\|\boldsymbol{\omega}_{r-1} - \boldsymbol{\omega}_r\|^2$$

$$+ 3\mathbb{E}\|\mathbf{v}_{n_t q} - \nabla_{\boldsymbol{\theta}}F_{n_t q}\|^2 + 3\mathbb{E}\|\mathbf{v}_{(n_t-1)q} - \nabla_{\boldsymbol{\theta}}F_{(n_t-1)q}\|^2$$

$$+ \sum_{r=(n_t-1)q+1}^{n_t q-1} \frac{3L_f^2}{q}\mathbb{E}\big(\|\boldsymbol{\theta}_r - \boldsymbol{\theta}_{r-1}\|^2 + \|\boldsymbol{\omega}_r - \boldsymbol{\omega}_{r-1}\|^2\big)$$

$$+ 3L_f^2\mathbb{E}\|\boldsymbol{\theta}_{n_t q-1} - \boldsymbol{\theta}_{n_t q}\|^2 + 3L_f^2\mathbb{E}\|\boldsymbol{\omega}_{n_t q-1} - \boldsymbol{\omega}_{n_t q}\|^2$$

$$\leq \sum_{r=(n_t-1)q+1}^{n_t q} 4L_f^2\mathbb{E}\|\boldsymbol{\theta}_{r-1} - \boldsymbol{\theta}_r\|^2 + 4L_f^2\mathbb{E}\|\boldsymbol{\omega}_{r-1} - \boldsymbol{\omega}_r\|^2$$

$$+ 3\mathbb{E}\|\mathbf{v}_{n_t q} - \nabla_{\boldsymbol{\theta}}F_{n_t q}\|^2 + 3\mathbb{E}\|\mathbf{v}_{(n_t-1)q} - \nabla_{\boldsymbol{\theta}}F_{(n_t-1)q}\|^2, \tag{72}$$

which leads to the following:

$$\sum_{t=1}^{T}\mathbb{E}\|\mathbf{v}_t - \mathbf{v}_{t-1}\|^2 \leq \sum_{t=1}^{T} 4L_f^2\mathbb{E}\|\boldsymbol{\theta}_{t-1} - \boldsymbol{\theta}_t\|^2 + 4L_f^2\mathbb{E}\|\boldsymbol{\omega}_{t-1} - \boldsymbol{\omega}_t\|^2 + \sum_{t=0}^{T} 6\mathbb{E}\|\mathbf{v}_{(n_t-1)q} - \nabla_{\boldsymbol{\theta}}F_{(n_t-1)q}\|^2.$$

Similarly, we have

$$\sum_{t=1}^{T}\mathbb{E}\|\mathbf{u}_t - \mathbf{u}_{t-1}\|^2 \leq \sum_{t=1}^{T} 4L_f^2\mathbb{E}\|\boldsymbol{\theta}_{t-1} - \boldsymbol{\theta}_t\|^2 + 4L_f^2\mathbb{E}\|\boldsymbol{\omega}_{t-1} - \boldsymbol{\omega}_t\|^2 + \sum_{t=0}^{T} 6\mathbb{E}\|\mathbf{u}_{(n_t-1)q} - \nabla_{\boldsymbol{\omega}}F_{(n_t-1)q}\|^2.$$

This completes the proof of the lemma. $\qquad\square$

With the defined potential function $\mathfrak{p}$, we have

$$\mathbb{E}\mathfrak{p}_{T+1} - \mathfrak{p}_0 \leq -\frac{\gamma}{2}\sum_{t=0}^{T}\mathbb{E}\|\nabla J(\bar{\boldsymbol{\theta}}_t)\|^2 - \gamma L_f^2 \sum_{t=0}^{T}\mathbb{E}\|\boldsymbol{\omega}_t^* - \bar{\boldsymbol{\omega}}_t\|^2$$

$$- \Big(\frac{\gamma}{2} - \frac{L_J\gamma^2}{2} - \frac{40\gamma^3 L_{\boldsymbol{\omega}}^2 L_f^2}{\mu^2\eta^2}\Big)\sum_{t=0}^{T}\mathbb{E}\|\bar{\mathbf{p}}_t\|^2 - \frac{2\gamma\eta L_f^2}{\mu}\sum_{t=0}^{T}\mathbb{E}\|\bar{\mathbf{d}}_t\|^2$$

$$- \Big(1 - (1+c_1)\lambda^2 - \frac{72\gamma L_f^4}{\mu^2} - 2\gamma L_f^2\Big)\sum_{t=0}^{T}\frac{\mathcal{E}(\boldsymbol{\theta}_t)}{m}$$

$$- \Big(1 - (1+c_2)\lambda^2 - \frac{72\gamma L_f^4}{\mu^2} - 2\gamma L_f^2\Big)\sum_{t=0}^{T}\frac{\mathcal{E}(\boldsymbol{\omega}_t)}{m}$$

$$- \Big(1 - (1+c_1)\lambda^2 - (1+\frac{1}{c_1})\gamma\Big)\gamma\sum_{t=0}^{T}\frac{\mathcal{E}(\mathbf{p}_t)}{m}$$

$$- \left(1 - (1 + c_2)\lambda^2 - (1 + \frac{1}{c_2})\eta\right)\eta \sum_{t=0}^{T} \frac{\mathcal{E}(\mathbf{d}_t)}{m}$$

$$\underbrace{+ \frac{2\gamma}{m} \sum_{t=0}^{T} \mathbb{E}\|\nabla_{\boldsymbol{\omega}} F_t - \mathbf{v}_t\|^2 + \frac{72L_f^2\gamma}{m\mu^2} \sum_{t=0}^{T} \mathbb{E}\|\nabla_{\boldsymbol{\theta}} F_t - \mathbf{u}_t\|^2}_{R_1}$$

$$\underbrace{+ (1 + \frac{1}{c_1})\frac{\gamma}{m} \sum_{t=1}^{T} \mathbb{E}\|\mathbf{v}_t - \mathbf{v}_{t-1}\|^2 + (1 + \frac{1}{c_2})\frac{\eta}{m} \sum_{t=1}^{T} \mathbb{E}\|\mathbf{u}_t - \mathbf{u}_{t-1}\|^2}_{R_2}. \qquad (73)$$

First, for the term $R_1$, we have

$$\frac{2\gamma}{m} \sum_{t=0}^{T} \mathbb{E}\|\nabla_{\boldsymbol{\omega}} F_t - \mathbf{v}_t\|^2 + \frac{72L_f^2\gamma}{m\mu^2} \sum_{t=0}^{T} \mathbb{E}\|\nabla_{\boldsymbol{\theta}} F_t - \mathbf{u}_t\|^2$$

$$\leq \frac{2\gamma}{m} \mathbb{E}\Big( \sum_{t=0}^{T} \|\mathbf{v}_{(n_t-1)q} - \nabla_{\boldsymbol{\theta}} F_{(n_t-1)q}\|^2 + \sum_{t=1}^{T} L_f^2 \big( \|\boldsymbol{\theta}_t - \boldsymbol{\theta}_{t-1}\|^2 + \|\boldsymbol{\omega}_t - \boldsymbol{\omega}_{t-1}\|^2 \big) \Big)$$

$$+ \frac{72L_f^2\gamma}{m\mu^2} \mathbb{E}\Big( \sum_{t=0}^{T} \|\mathbf{u}_{(n_t-1)q} - \nabla_{\boldsymbol{\omega}} F_{(n_t-1)q}\|^2 + \sum_{t=1}^{T} L_f^2 \big( \|\boldsymbol{\theta}_t - \boldsymbol{\theta}_{t-1}\|^2 + \|\boldsymbol{\omega}_t - \boldsymbol{\omega}_{t-1}\|^2 \big) \Big)$$

$$= L_f^2 \Big( \frac{2\gamma}{m} + \frac{72L_f^2\gamma}{m\mu^2} \Big) \sum_{t=1}^{T} \mathbb{E}\big( \|\boldsymbol{\theta}_t - \boldsymbol{\theta}_{t-1}\|^2 + \|\boldsymbol{\omega}_t - \boldsymbol{\omega}_{t-1}\|^2 \big)$$

$$+ \frac{2\gamma}{m} \sum_{t=0}^{T} \mathbb{E}\|\mathbf{v}_{(n_t-1)q} - \nabla_{\boldsymbol{\theta}} F_{(n_t-1)q}\|^2 + \frac{72L_f^2\gamma}{m\mu^2} \sum_{t=0}^{T} \mathbb{E}\|\mathbf{u}_{(n_t-1)q} - \nabla_{\boldsymbol{\omega}} F_{(n_t-1)q}\|^2. \qquad (74)$$

Then, for term $R_2$, we can bound it as follows:

$$(1 + \frac{1}{c_1})\frac{\gamma}{m} \sum_{t=1}^{T} \mathbb{E}\|\mathbf{v}_t - \mathbf{v}_{t-1}\|^2 + (1 + \frac{1}{c_2})\frac{\eta}{m} \sum_{t=1}^{T} \mathbb{E}\|\mathbf{u}_t - \mathbf{u}_{t-1}\|^2$$

$$\leq (1 + \frac{1}{c_1})\frac{\gamma}{m} \Big( \sum_{t=1}^{T} 4L_f^2 \mathbb{E}\|\boldsymbol{\theta}_{t-1} - \boldsymbol{\theta}_t\|^2 + 4L_f^2 \mathbb{E}\|\boldsymbol{\omega}_{t-1} - \boldsymbol{\omega}_t\|^2 + \sum_{t=0}^{T} 6\mathbb{E}\|\mathbf{v}_{(n_t-1)q} - \nabla_{\boldsymbol{\theta}} F_{(n_t-1)q}\|^2 \Big)$$

$$+ (1 + \frac{1}{c_2})\frac{\eta}{m} \Big( \sum_{t=1}^{T} 4L_f^2 \mathbb{E}\|\boldsymbol{\theta}_{t-1} - \boldsymbol{\theta}_t\|^2 + 4L_f^2 \mathbb{E}\|\boldsymbol{\omega}_{t-1} - \boldsymbol{\omega}_t\|^2 + \sum_{t=0}^{T} 6\mathbb{E}\|\mathbf{u}_{(n_t-1)q} - \nabla_{\boldsymbol{\omega}} F_{(n_t-1)q}\|^2 \Big)$$

$$\leq (1 + \frac{1}{c_1})\frac{6\gamma}{m} \sum_{t=0}^{T} 6\mathbb{E}\|\mathbf{v}_{(n_t-1)q} - \nabla_{\boldsymbol{\theta}} F_{(n_t-1)q}\|^2 + (1 + \frac{1}{c_2})\frac{6\eta}{m} \sum_{t=0}^{T} \mathbb{E}\|\mathbf{u}_{(n_t-1)q} - \nabla_{\boldsymbol{\omega}} F_{(n_t-1)q}\|^2$$

$$+ \frac{4L_f^2}{m} \Big( (1 + \frac{1}{c_1})\gamma + (1 + \frac{1}{c_2})\eta \Big) \sum_{t=1}^{T} \mathbb{E}\big( \|\boldsymbol{\theta}_{t-1} - \boldsymbol{\theta}_t\|^2 + \|\boldsymbol{\omega}_{t-1} - \boldsymbol{\omega}_t\|^2 \big). \qquad (75)$$

Thus, we have:

$$R_1 + R_2 \leq \frac{4L_f^2}{m} \Big( (1 + \frac{1}{c_1})\gamma + (1 + \frac{1}{c_2})\eta + \frac{\gamma}{2} + \frac{18L_f^2\gamma}{\mu^2} \Big) \sum_{t=1}^{T} \mathbb{E}\|\boldsymbol{\theta}_{t-1} - \boldsymbol{\theta}_t\|^2$$

$$+ \frac{4L_f^2}{m} \Big( (1 + \frac{1}{c_1})\gamma + (1 + \frac{1}{c_2})\eta + \frac{\gamma}{2} + \frac{18L_f^2\gamma}{\mu^2} \Big) \sum_{t=1}^{T} \mathbb{E}\|\boldsymbol{\omega}_{t-1} - \boldsymbol{\omega}_t\|^2$$

$$+ \Big( (1 + \frac{1}{c_1})\frac{6\gamma}{m} + \frac{72\gamma}{m} \Big) \sum_{t=0}^{T} \mathbb{E}\|\mathbf{v}_{(n_t-1)q} - \nabla_{\boldsymbol{\theta}} F_{(n_t-1)q}\|^2$$

$$+ \left( (1 + \frac{1}{c_2}) \frac{6\eta}{m} + \frac{72 L_f^2 \gamma}{m \mu^2} \right) \sum_{t=0}^{T} \mathbb{E} \| \mathbf{u}_{(n_t - 1)q} - \nabla_{\boldsymbol{\omega}} F_{(n_t-1)q} \|^2$$

$$\leq \frac{4 L_f^2}{m} \left( (1 + \frac{1}{c_1}) \gamma + (1 + \frac{1}{c_2}) \eta + \frac{\gamma}{2} + \frac{18 L_f^2 \gamma}{\mu^2} \right) \sum_{t=1}^{T} \mathbb{E} \left( 8 \mathcal{E}(\boldsymbol{\theta}_{t-1}) + 4 \gamma^2 \mathcal{E}(\mathbf{p}_{t-1}) + 4 \gamma^2 m \| \bar{\mathbf{p}}_{t-1} \|^2 \right)$$

$$+ \frac{4 L_f^2}{m} \left( (1 + \frac{1}{c_1}) \gamma + (1 + \frac{1}{c_2}) \eta + \frac{\gamma}{2} + \frac{18 L_f^2 \gamma}{\mu^2} \right) \sum_{t=1}^{T} \mathbb{E} \left( 8 \mathcal{E}(\boldsymbol{\omega}_{t-1}) + 4 \eta^2 \mathcal{E}(\mathbf{d}_{t-1}) + 4 \eta^2 m \| \bar{\mathbf{d}}_{t-1} \|^2 \right)$$

$$+ \left( (1 + \frac{1}{c_1}) \frac{6\gamma}{m} + \frac{2\gamma}{m} \right) \sum_{t=0}^{T} \mathbb{E} \| \mathbf{v}_{(n_t - 1)q} - \nabla_{\boldsymbol{\theta}} F_{(n_t - 1)q} \|^2$$

$$+ \left( (1 + \frac{1}{c_2}) \frac{6\eta}{m} + \frac{72 L_f^2 \gamma}{m \mu^2} \right) \sum_{t=0}^{T} \mathbb{E} \| \mathbf{u}_{(n_t - 1)q} - \nabla_{\boldsymbol{\omega}} F_{(n_t - 1)q} \|^2. \tag{76}$$

Plugging the above results, we have

$$\mathbb{E} \mathfrak{p}_{T+1} - \mathfrak{p}_0 \leq -\frac{\gamma}{2} \sum_{t=0}^{T} \mathbb{E} \| \nabla J(\bar{\boldsymbol{\theta}}_t) \|^2 - \gamma L_f^2 \sum_{t=0}^{T} \mathbb{E} \| \boldsymbol{\omega}_t^* - \bar{\boldsymbol{\omega}}_t \|^2$$

$$- c_{\bar{\mathbf{p}}} \sum_{t=0}^{T} \gamma \mathbb{E} \| \bar{\mathbf{p}}_t \|^2 - c_{\bar{\mathbf{d}}} \sum_{t=0}^{T} \gamma \eta \mathbb{E} \| \bar{\mathbf{d}}_t \|^2 - c_{\boldsymbol{\theta}} \sum_{t=0}^{T} \frac{\mathcal{E}(\boldsymbol{\theta}_t)}{m}$$

$$- c_{\boldsymbol{\omega}} \sum_{t=0}^{T} \frac{\mathcal{E}(\boldsymbol{\omega}_t)}{m} - c_{\mathbf{p}} \sum_{t=0}^{T} \frac{\gamma \mathcal{E}(\mathbf{p}_t)}{m} - c_{\mathbf{d}} \sum_{t=0}^{T} \frac{\eta \mathcal{E}(\mathbf{d}_t)}{m}$$

$$+ \left( (1 + \frac{1}{c_1}) \frac{6\gamma}{m} + \frac{2\gamma}{m} \right) \sum_{t=0}^{T} \mathbb{E} \| \mathbf{v}_{(n_t - 1)q} - \nabla_{\boldsymbol{\theta}} F_{(n_t - 1)q} \|^2$$

$$+ \left( (1 + \frac{1}{c_2}) \frac{6\eta}{m} + \frac{72 L_f^2 \gamma}{m \mu^2} \right) \sum_{t=0}^{T} \mathbb{E} \| \mathbf{u}_{(n_t - 1)q} - \nabla_{\boldsymbol{\omega}} F_{(n_t - 1)q} \|^2, \tag{77}$$

where

$$c_{\bar{\mathbf{p}}} = \frac{1}{2} - \frac{L_J \gamma}{2} - \frac{40 \gamma^2 L_{\boldsymbol{\omega}}^2 L_f^2}{\mu^2 \eta^2} - 16 L_f^2 \gamma^2 \left( (1 + \frac{1}{c_1}) + (1 + \frac{1}{c_2}) \frac{\eta}{\gamma} + \frac{1}{2} + \frac{18 L_f^2}{\mu^2} \right), \tag{78}$$

$$c_{\bar{\mathbf{d}}} = \frac{2 L_f^2}{\mu} - 16 L_f^2 \eta \left( (1 + \frac{1}{c_1}) + (1 + \frac{1}{c_2}) \frac{\eta}{\gamma} + \frac{1}{2} + \frac{18 L_f^2}{\mu^2} \right), \tag{79}$$

$$c_{\boldsymbol{\theta}} = 1 - (1 + c_1) \lambda^2 - \frac{72 \gamma L_f^4}{\mu^2} - 2 \gamma L_f^2 - 32 L_f^2 \gamma \left( (1 + \frac{1}{c_1}) + (1 + \frac{1}{c_2}) \frac{\eta}{\gamma} + \frac{1}{2} + \frac{18 L_f^2}{\mu^2} \right), \tag{80}$$

$$c_{\boldsymbol{\omega}} = 1 - (1 + c_2) \lambda^2 - \frac{72 \gamma L_f^4}{\mu^2} - 2 \gamma L_f^2 - 32 L_f^2 \gamma \left( (1 + \frac{1}{c_1}) + (1 + \frac{1}{c_2}) \frac{\eta}{\gamma} + \frac{1}{2} + \frac{18 L_f^2}{\mu^2} \right), \tag{81}$$

$$c_{\mathbf{p}} = 1 - (1 + c_1) \lambda^2 - (1 + \frac{1}{c_1}) \gamma - 16 L_f^2 \gamma^2 \left( (1 + \frac{1}{c_1}) + (1 + \frac{1}{c_2}) \frac{\eta}{\gamma} + \frac{1}{2} + \frac{18 L_f^2}{\mu^2} \right), \tag{82}$$

$$c_{\mathbf{d}} = 1 - (1 + c_2) \lambda^2 - (1 + \frac{1}{c_2}) \eta - 16 L_f^2 \eta \gamma \left( (1 + \frac{1}{c_1}) + (1 + \frac{1}{c_2}) \frac{\eta}{\gamma} + \frac{1}{2} + \frac{18 L_f^2}{\mu^2} \right). \tag{83}$$

Choose $c_1 = c_2 = 1/\lambda - 1$, and define $C_0 = \frac{1}{1-\lambda}(1 + \frac{\eta}{\gamma}) + \frac{1}{2} + \frac{18 L_f^2}{\mu^2}$. It follows that

$$c_{\bar{\mathbf{p}}} = \frac{1}{2} - \frac{L_J \gamma}{2} - \frac{40 \gamma^2 L_{\boldsymbol{\omega}}^2 L_f^2}{\mu^2 \eta^2} - 16 C_0 L_f^2 \gamma^2, \tag{84}$$

$$c_{\bar{\mathbf{d}}} = \frac{2 L_f^2}{\mu} - 16 C_0 L_f^2 \eta, \tag{85}$$

$$c_{\boldsymbol{\theta}} = c_{\boldsymbol{\omega}} = 1 - \lambda - \frac{72\gamma L_f^4}{\mu^2} - 2\gamma L_f^2 - 32C_0 L_f^2 \gamma, \tag{86}$$

$$c_{\mathbf{p}} = 1 - \lambda - \frac{\gamma}{1-\lambda} - 16C_0 L_f^2 \gamma^2, \tag{87}$$

$$c_{\mathbf{d}} = 1 - \lambda - \frac{\eta}{1-\lambda} - 16C_0 L_f^2 \eta\gamma. \tag{88}$$

To ensure $c_{\bar{\mathbf{p}}} \geq 0$, we have

$$
\begin{aligned}
c_{\bar{\mathbf{p}}} &\overset{(a)}{\geq} \frac{1}{4} - \frac{L_J \gamma}{2} - 16C_0 L_f^2 \gamma^2 \\
&\overset{(b)}{\geq} \frac{1}{4} - \frac{(L_f + L_f^2/\mu)\gamma}{2} - \frac{(1-\lambda)\gamma}{2} \overset{(c)}{\geq} 0,
\end{aligned}
\tag{89}
$$

where (a) follows from $\kappa := \gamma/\eta \leq \mu^2/13L_f^2$ and Lemma 9, (b) is due to $\gamma \leq (1-\lambda)/32C_0 L_f^2$ and Lemma 11, and (c) is from $\gamma \leq 1/2\big((L_f + L_f^2/\mu) + (1-\lambda)\big)$. By setting $\eta \leq 1/8\mu C_0$, we have $c_{\bar{\mathbf{d}}} \geq 0$. By setting $\gamma \leq (1-\lambda)/(\frac{1}{2} + \frac{72L_f^4}{\mu^2} + 2L_f^2 + 32C_0 L_f^2)$, we have $c_{\boldsymbol{\theta}} = c_{\boldsymbol{\omega}} \geq \gamma/2$. To ensure $c_{\mathbf{p}} \geq 0$,

$$
\begin{aligned}
c_{\mathbf{p}} &= 1 - \lambda - \frac{\gamma}{1-\lambda} - 16C_0 L_f^2 \gamma^2 \\
&\overset{(a)}{\geq} 1 - \lambda - \frac{\gamma}{1-\lambda} - \frac{(1-\lambda)\gamma}{2} \overset{(b)}{\geq} 0,
\end{aligned}
\tag{90}
$$

where (a) follows from $\gamma \leq (1-\lambda)/32C_0 L_f^2$ and (b) is due to $\gamma \leq 1/\big(1/2 + 1/(1-\lambda)^2\big)$. Similarly, with $\eta \leq 1/\big(1/2 + 1/(1-\lambda)^2\big)$, we have $c_{\mathbf{d}} \geq 0$.

To summarize, we need the following conditions to ensure $c_{\bar{\mathbf{p}}} \geq 0$, $C_{\bar{\mathbf{d}}} \geq 0$, $C_{\mathbf{p}} \geq 0$, $C_{\mathbf{d}} \geq 0$, $C_{\boldsymbol{\theta}} \geq \gamma/2$, $c_{\boldsymbol{\omega}} \geq \gamma/2$,

$$\kappa = \gamma/\eta \leq \mu^2/13L_f^2, \tag{91}$$

$$\gamma \leq \min\left\{ \frac{1}{2}\Big(L_f + \frac{L_f^2}{\mu} + (1-\lambda)\Big),\ \frac{1-\lambda}{(\frac{1}{2} + \frac{72L_f^4}{\mu^2} + 2L_f^2 + 32C_0 L_f^2)},\ \big(1/2 + 1/(1-\lambda)^2\big)^{-1} \right\}, \tag{92}$$

$$\eta \leq \min\left\{ \frac{1}{8\mu C_0},\ \big(1/2 + 1/(1-\lambda)^2\big)^{-1} \right\}, \tag{93}$$

which can be satisfied by:

$$\kappa = \gamma/\eta \leq \mu^2/13L_f^2, \tag{94}$$

$$
\begin{aligned}
\eta \leq \min\Big\{ &\frac{13L_f^2}{2\mu^2}\Big(L_f + \frac{L_f^2}{\mu} + (1-\lambda)\Big),\ \frac{26(1-\lambda)L_f^2}{(\mu^2 + 144L_f^4 + 4L_f^2\mu^2 + 64C_0 L_f^2\mu^2)}, \\
&\frac{1}{8\mu C_0},\ \frac{13L_f^2}{\mu^2\big(1/2 + 1/(1-\lambda)^2\big)} \Big\},
\end{aligned}
\tag{95}
$$

where the constant $C_0$ is defined as $C_0 = \frac{1}{1-\lambda}(1 + \frac{1}{\kappa}) + \frac{1}{2} + \frac{18L_f^2}{\mu^2}$. Also, it can be easily verified that $\frac{1}{8\mu C_0} \leq \frac{1}{2L_f}$.

With the above conditions, we have:

$$
\begin{aligned}
\frac{\gamma}{2} &\sum_{t=0}^{T} \mathbb{E}\|\nabla J(\bar{\boldsymbol{\theta}}_t)\|^2 + 2L_f^2 \mathbb{E}\|\boldsymbol{\omega}_t^* - \bar{\boldsymbol{\omega}}_t\|^2 + \frac{\mathcal{E}(\boldsymbol{\theta}_t)}{m} + \frac{\mathcal{E}(\boldsymbol{\omega}_t)}{m} \\
&\leq \mathbb{E}[\mathfrak{p}_0 - \mathfrak{p}_{T+1}] + \Big(\frac{6\gamma}{m\lambda} + \frac{2\gamma}{m}\Big) \sum_{t=0}^{T} \mathbb{E}\|\mathbf{v}_{(n_t-1)q} - \nabla_{\boldsymbol{\theta}} F_{(n_t-1)q}\|^2 \\
&\quad + \Big(\frac{6\eta}{m\lambda} + \frac{72L_f^2\gamma}{m\mu^2}\Big) \sum_{t=0}^{T} \mathbb{E}\|\mathbf{u}_{(n_t-1)q} - \nabla_{\boldsymbol{\omega}} F_{(n_t-1)q}\|^2.
\end{aligned}
\tag{96}
$$

For GT-SRVR, the outer loop calculates the full gradients. Thus, we have $\mathbb{E}\|\mathbf{v}_{(n_t-1)q} - \nabla_{\boldsymbol{\theta}}F_{(n_t-1)q}\|^2 = \mathbb{E}\|\mathbf{u}_{(n_t-1)q} - \nabla_{\boldsymbol{\omega}}F_{(n_t-1)q}\|^2 = 0$. Then, we have the stated result in Theorem 3:

$$\frac{1}{T+1}\sum_{t=0}^{T}\mathbb{E}\|\nabla J(\bar{\boldsymbol{\theta}}_t)\|^2 + 2L_f^2\mathbb{E}\|\boldsymbol{\omega}_t^* - \bar{\boldsymbol{\omega}}_t\|^2 + \frac{\mathcal{E}(\boldsymbol{\theta}_t)}{m} + \frac{\mathcal{E}(\boldsymbol{\omega}_t)}{m} \leq \frac{2\mathbb{E}[\mathfrak{p}_0 - \mathfrak{p}_{T+1}]}{(T+1)\gamma}. \tag{97}$$

For GT-SRVRI, we have that

$$\sum_{t=0}^{T}\mathbb{E}\|\mathbf{v}_{(n_t-1)q} - \nabla_{\boldsymbol{\theta}}F_{(n_t-1)q}\|^2 = \sum_{t=0}^{T}\mathbb{E}\|\mathbf{u}_{(n_t-1)q} - \nabla_{\boldsymbol{\omega}}F_{(n_t-1)q}\|^2$$

$$= \sum_{t=0}^{T}\frac{m\sigma^2\mathbb{1}_{(|\mathcal{R}_{i,(n_t-1)q}|)}}{|\mathcal{R}_{i,(n_t-1)q}|} \leq \sum_{t=0}^{T}\frac{m\sigma^2\mathbb{1}_{(|\mathcal{R}_{i,\lfloor t/q\rfloor q}|<n)}}{\min\{(\lfloor t/q\rfloor+1)^{\alpha}q, \lceil c_{\epsilon}\epsilon^{-2}\rceil\}}$$

$$\leq \sum_{t=0}^{T}m\sigma^2\max\{\frac{\mathbb{1}_{(\lfloor t/q\rfloor+1)^{\alpha}q<n)}}{(\lfloor t/q\rfloor+1)^{\alpha}q}, \frac{\epsilon^2}{c_{\epsilon}}\} \leq m\sigma^2(\sum_{r=1}^{\infty}\frac{\mathbb{1}_{(\tau^{\alpha}q<n)}}{\tau^{\alpha}} + \frac{\epsilon^2(T+1)}{c_{\epsilon}})$$

$$\leq m\sigma^2(1 + \int_{1}^{\frac{n}{q}^{1/\alpha}}\frac{1}{\tau^{\alpha}}d\tau + \frac{\epsilon^2(T+1)}{c_{\epsilon}})$$

$$\leq \begin{cases} m\sigma^2(\frac{1}{1-\alpha}(\frac{n}{q})^{(\frac{1}{\alpha}-1)} - \frac{\alpha}{1-\alpha} + \frac{\epsilon^2(T+1)}{c_{\epsilon}}), & \text{if } \alpha > 0 \text{ and } \alpha \neq 0, \\ m\sigma^2(1 + \log(\frac{n}{q}) + \frac{\epsilon^2(T+1)}{c_{\epsilon}}), & \text{if } \alpha = 1. \end{cases} \tag{98}$$

Thus, for GT-SRVRI, we have the following convergence results:

$$\frac{1}{T+1}\sum_{t=0}^{T}\mathbb{E}\|\nabla J(\bar{\boldsymbol{\theta}}_t)\|^2 + 2L_f^2\mathbb{E}\|\boldsymbol{\omega}_t^* - \bar{\boldsymbol{\omega}}_t\|^2 + \frac{\mathcal{E}(\boldsymbol{\theta}_t)}{m} + \frac{\mathcal{E}(\boldsymbol{\omega}_t)}{m}$$

$$\leq \frac{2\mathbb{E}[\mathfrak{p}_0 - \mathfrak{p}_{T+1}]}{(T+1)\gamma} + (\frac{12}{\lambda}(1+\frac{1}{r}) + 4 + \frac{144L_f^2}{\mu^2})(\frac{\epsilon^2}{c_{\epsilon}} + \frac{C(n,q,\alpha)}{T+1})\sigma^2, \tag{99}$$

where the constant $C(n,q,\alpha)$ is defined as

$$C(n,q,\alpha) \triangleq \begin{cases} \frac{1}{1-\alpha}(\frac{n}{q})^{(\frac{1}{\alpha}-1)} - \frac{\alpha}{1-\alpha}, & \text{if } \alpha > 0 \text{ and } \alpha \neq 1, \\ \log(\frac{n}{q}) + 1, & \text{if } \alpha = 1. \end{cases} \tag{100}$$

With $\mathfrak{p}_{T+1} \geq J^*$, we reach the conclusion.

# D  Supporting lemmas

**Lemma 9.** *Under Assumption 1, $\boldsymbol{\omega}^*(\boldsymbol{\theta}) = \arg\max_{\boldsymbol{\omega}} F(\boldsymbol{\theta}, \boldsymbol{\omega})$ is Lipschitz continuous, i.e., there exists a positive constant $L_{\boldsymbol{\omega}}$, such that*

$$\|\boldsymbol{\omega}^*(\boldsymbol{\theta}) - \boldsymbol{\omega}^*(\boldsymbol{\theta}')\| \leq L_{\boldsymbol{\omega}}\|\boldsymbol{\theta} - \boldsymbol{\theta}'\|, \ \forall \boldsymbol{\theta}, \boldsymbol{\theta}' \in \mathbb{R}^d, \tag{101}$$

*where the Lipschitz constant is $L_{\boldsymbol{\omega}} = L_F/\mu$ for Algorithm 1 and $L_{\boldsymbol{\omega}} = L_f/\mu$ for Algorithm 2.*

*Proof.* See Lemma 4.3 in [28]. $\qquad\square$

**Lemma 10.** *Under Assumption 1, the function $J(\boldsymbol{\theta}) = F(\boldsymbol{\theta}, \boldsymbol{\omega}^*(\boldsymbol{\theta}))$ satisfies that $\nabla J(\boldsymbol{\theta}) = \nabla_{\boldsymbol{\theta}}F(\boldsymbol{\theta}, \boldsymbol{\omega}^*(\boldsymbol{\theta}))$.*

*Proof.* Since $J(\boldsymbol{\theta}) = F(\boldsymbol{\theta}, \boldsymbol{\omega}^*(\boldsymbol{\theta}))$, by chain rule, we have

$$dJ(\boldsymbol{\theta}) = \frac{\partial F(\boldsymbol{\theta}, \boldsymbol{\omega})}{\partial \boldsymbol{\theta}}\Big|_{\boldsymbol{\omega}=\boldsymbol{\omega}^*(\boldsymbol{\theta})} \cdot d\boldsymbol{\theta} + \frac{\partial F(\boldsymbol{\theta}, \boldsymbol{\omega})}{\partial \boldsymbol{\omega}}\Big|_{\boldsymbol{\omega}=\boldsymbol{\omega}^*(\boldsymbol{\theta})} \cdot \frac{\partial \boldsymbol{\omega}^*(\boldsymbol{\theta})}{\partial \boldsymbol{\theta}} \cdot d\boldsymbol{\theta}, \tag{102}$$

where $\partial F(\boldsymbol{\theta}, \boldsymbol{\omega})/\partial \boldsymbol{\theta}$ and $\partial F(\boldsymbol{\theta}, \boldsymbol{\omega})/\partial \boldsymbol{\omega}$ are respectively the partial differential of $F$ w.r.t the first variate $\boldsymbol{\theta}$ and the second variate $\boldsymbol{\omega}$. Note that $\boldsymbol{\omega}^*(\boldsymbol{\theta})$ is the unique optimal point such that $F(\boldsymbol{\theta}, \boldsymbol{\omega})$

reaches the maximums. So, it follows that $\frac{\partial F(\boldsymbol{\theta},\boldsymbol{\omega})}{\partial \boldsymbol{\omega}}|_{\boldsymbol{\omega}=\boldsymbol{\omega}^*(\boldsymbol{\theta})} = 0$ for all $\boldsymbol{\theta}$. Also, from Lemma 9, we have $\partial\omega^*(\boldsymbol{\theta})/\partial\boldsymbol{\theta}$ is bounded. Thus, it follows that

$$dJ(\boldsymbol{\theta}) = \left.\frac{\partial F(\boldsymbol{\theta},\boldsymbol{\omega})}{\partial\boldsymbol{\theta}}\right|_{\boldsymbol{\omega}=\boldsymbol{\omega}^*(\boldsymbol{\theta})} \cdot d\boldsymbol{\theta}, \tag{103}$$

which is $\nabla J(\boldsymbol{\theta}) = \nabla_{\boldsymbol{\theta}} F(\boldsymbol{\theta},\boldsymbol{\omega}^*(\boldsymbol{\theta}))$.

Additionally, we can follow the detailed derivation of Theorem D2 in [2] to give a more rigorous proof for Eq. (103). The difference between our Lemma 10 and Theorem D2 in [2] is that Ref. [2] adopted the uniformly bounded gradient assumption (Hypotheses D2.2 in [2]). In the followings, we show that under our assumptions, Eq. (103) can be still obtained with similar derivation in [2]. Here we will adopt the notations in [2]. Because of the strong concavity, there is only one set $\{\boldsymbol{\nu}_n\}$ in $\mathcal{W}(\mathbf{u})$ for all $\mathbf{u}$ and $\boldsymbol{\nu}^* = \lim_{n\to} \boldsymbol{\nu}_n$ exists. From Assumption 1 (e), $D_1 J(\mathbf{u},\boldsymbol{\nu};h)$ is bounded. Thus, due to Assumption 1 (b), there exists a $N_1$, with $n \geq N_1$, $D_1 J(\mathbf{u},\boldsymbol{\nu}_n;h)$ is also bounded. So, Proposition 2 in [2] holds. Meanwhile, because of the continuity of $D_1 J(\mathbf{u},\boldsymbol{\nu})$ and $D_1 J(\mathbf{u},\boldsymbol{\nu}^*) < \infty$, there exists $N_2$ with $n \geq \max\{N_1, N_2\}$, the function $t \mapsto J(\mathbf{u} + th, \boldsymbol{\nu}_n)$ has a bounded directional derivative for all $t \in [0, t_n]$. Thus, Proposition 3 in [2] holds. Because Proposition 2 and Proposition 3 in [2] still holds under our assumptions, we can then reach Eq. (103), and so the result in Lemma 10 immediately follows. $\qquad\square$

**Lemma 11.** *Under Assumption 1, the funciton $J(\boldsymbol{\theta}) = F(\boldsymbol{\theta},\omega^*(\boldsymbol{\theta}))$ w.r.t $\boldsymbol{\theta}$ is Lipschitz smooth, i.e., there exists a positive constant $L_J$, such that*

$$\|\nabla J(\boldsymbol{\theta}) - \nabla J(\boldsymbol{\theta}')\| \leq L_J \|\boldsymbol{\theta} - \boldsymbol{\theta}'\|, \quad \forall \boldsymbol{\theta}, \boldsymbol{\theta}' \in \mathbb{R}^d, \tag{104}$$

*where the Lipschitz constant is $L_J = L_F + L_F^2/\mu$ for Algorithm 1 and $L_J = L_f + L_f^2/\mu$ for Algorithm 2.*

*Proof.* The lemma follows immediately from Lemma 4.3 in [28] and Lemma 10. $\qquad\square$