# OpenReview forum: " Taming Communication and Sample Complexities in Decentralized Policy Evaluation for Cooperative Multi-Agent Reinforcement Learning"
_NeurIPS.cc/2021/Conference — NeurIPS 2021 Poster_

### Official Review · Reviewer_ZsKo · 2021-07-09

**Rating:** 7
**Confidence:** 5

**Summary:**

This work analyzes decentralized policy evaluation problem for MARL under non-linear approximation for the first time, and the sample complexity is further reduced via variance reduction technique. The prevalent compact feasible set assumption is relaxed to mild assumptions on the objective function.

**Ethical Concerns:**

I think this work has no ethical concerns.

**Ethics Review Area:**

["I don’t know"]

**Limitations And Societal Impact:**

Limitations are not mentioned.
Both the authors and I do not think this theoretical work could have any immediate societal impact.

**Main Review:**

Originality: This work is claimed as the first analysis on decentralized policy evaluation problem for MARL under non-linear approximation. However, advice (1) below probably invalidates this claim, so this work is not that original.

Quality: The submission is technically sound for minimax optimization but not for RL. See advice (1) below for detail.

Clarity: The submission is clear and well organized.

Significance: The results are not significant. See advice (1) below for detail.

Advice and questions:
(1) Your evaluation metric $\frac{1}{T+1} \sum_{t=0}^{T} \mathbb{E}\left[\mathfrak{M}_{t}\right]$ is only about the empirical objective function $F$ and its related $J$, which depends on the fixed data. In contrast, all the policy evaluation works I read use the objective function $\mathcal{L}$ (eq. (1)) to ensure generalization ability. In addition, your Assumptions 1-3 are also data-dependent. In this case, I can envision that the proof simply repeats or slightly adjusts that of the finite-sum minimax optimization, without using the generating distribution of the Markovian samples, yes? If yes, this analysis is simply about finite-sum minimax optimization not RL. $\textbf{This is the main reason why I reject this paper. If you can solve this issue, I will accept this paper.}$

To ensure convergence on $\mathcal{L}$, each iteration needs to use a different sample mini-batch to avoid high correlation among stochastic gradients and high computation complexity, as done by almost all the works I have seen. The data should be treated as randomly generated from MDP, not as fixed. Then, bias and variance analysis of the stochastic gradient that estimates $\nabla \mathcal{L}$ are perhaps also needed. Finally, the evaluation metric should rely on $\mathcal{L}$ not dataset, and the complexity should not contain data size $n$.

Also, it is more desirable to replace the data-dependent assumptions into some more basic assumptions on the MDP. For example, bounded reward mapping $R$ can imply and thus replace Assumption 1a. Assumptions on $V_{\theta}$ are perhaps also desirable such as Lipschitz or L-smooth properties, which might imply some of Assumptions 1-3.

(2) In the following sentence in the abstract, you could add “under nonlinear function approximation”, since [14], [37], [42] achieved lower complexity than yours under linear function approximation.

“To our knowledge, this paper is the first work that achieves O(ε−2) in both sample  and communication complexities in decentralized policy evaluation for cooperative MARL. ”

(3) Typo: In the V function definition in the first paragraph of Section 2.1, change $s_{t+1}$ into $a_t$.

(4) Typo: In eq. (2), change $\frac{1}{m}\sum_{t=1}^m$ into $\frac{1}{n}\sum_{t=1}^n$.

(5) The author sequence of [14] should be
Gang Wang, Songtao Lu, Georgios B. Giannakis, Gerald Tesauro, Jian Sun3

(6) You might also cite the following papers on policy evaluation using linear function approximation.

- Sun, J., Wang, G., Giannakis, G. B., Yang, Q., & Yang, Z. (2020, June). Finite-time analysis of decentralized temporal-difference learning with linear function approximation. In International Conference on Artificial Intelligence and Statistics (pp. 4485-4495). PMLR.

- Chen, Z., Zhou, Y., & Chen, R. (2021). Multi-Agent Off-Policy TD Learning: Finite-Time Analysis with Near-Optimal Sample Complexity and Communication Complexity. ArXiv:2103.13147.

- Liu, R., & Olshevsky, A. (2021). Distributed TD (0) with Almost No Communication. ArXiv:2104.07855.

**Time Spent Reviewing:**

About 5 hours

---

> ### Author Response · Authors · 2021-08-10
> **Response to Reviewer ZsKo's Comments**
>
> > **Your Comment:** Advice and questions: (1) Your evaluation metric $\frac{1}{T+1} \sum_{t=0}^{T} \mathbb{E} [\mathfrak{M_t} ]$ is only about the empirical objective function $F$ and its related $J$, which depends on the fixed data. In contrast, all the policy evaluation works I read use the objective function $\mathcal{L}$ (eq. (1)) to ensure generalization ability. In addition, your Assumptions 1-3 are also data-dependent. In this case, I can envision that the proof simply repeats or slightly adjusts that of the finite-sum minimax optimization, without using the generating distribution of the Markovian samples, yes? If yes, this analysis is simply about finite-sum minimax optimization not RL.
> To ensure convergence on $\mathcal{L}$, each iteration needs to use a different sample mini-batch to avoid high correlation among stochastic gradients and high computation complexity, as done by almost all the works I have seen. The data should be treated as randomly generated from MDP, not as fixed. Then, bias and variance analysis of the stochastic gradient that estimates $\nabla \mathcal{L}$ are perhaps also needed. Finally, the evaluation metric should rely on $\mathcal{L}$ not dataset, and the complexity should not contain data size $n$.
> Also, it is more desirable to replace the data-dependent assumptions into some more basic assumptions on the MDP. For example, bounded reward mapping $R$ can imply and thus replace Assumption 1a. Assumptions on $V_{\theta}$ are perhaps also desirable such as Lipschitz or $L$-smooth properties, which might imply some of Assumptions 1-3.
>
> **Our Response:** Thanks for your comment on the difference between our finite sample policy evaluation (PE) problem with online PE problem with Markovian sampling. In our work, we are solving the class of offline PE problem in RL tasks over a network of collaborative multi-agents. Such offline PE problems are of significant interest in the RL community and have been studied in many existing works on RL (e.g., [R1-3]). The main goal of our work is to develop both sample and communication-efficient algorithms for multi-agent non-linear value function approximation. As with most machine learning problems, the problem of policy evaluation in MARL contains two types of errors, namely optimization error and generalization error. In our work, we focus on the ***optimization error*** by studying the convergence performance of our algorithms for min-max problem under the decentralized multi-agent framework, as the decentralized min-max problem itself is highly non-trivial and is still being actively researched.
>
> For the generalization error resulted from neglecting the Markovian nature of the samples, we note that as the trajectory sample size increases and under some bounded high-order moment assumptions, the empirical objectives $F$ and $J $ will converge to their expected one $L$ and theoretical MSPBE, so that the generalization error could be mitigated. Motivated by the large amount of sample size, we developed GT-SRVR and GT-SRVRI to lower the sample complexity.
>
> Nonetheless, we would like to thank the reviewer for providing the references and offering advice on including Markovian sampling to improve our analysis. In fact, we are aware that there have been a series of recent works on RL optimization with Markovian data (e.g., [R4-6]). We will add a discussion and related work in our manuscript. Also, as non-linear approximation with Markovian online sampling is an important open problem in its own right, which deserves an independent paper. This topic will be left as the next step in our future study.
>
> > **Your Comment:** Advice (2) In the following sentence in the abstract, you could add “under nonlinear function approximation”, since [14], [37], [42] achieved lower complexity than yours under linear function approximation.
>
> **Our Response:** Thanks for your suggestion. We have updated the abstract to make the statement more precise, as you suggested.
>
> > **Your Comment:** (3) Typo: In the V function definition in the first paragraph of Section 2.1, change $s_{t+1}$ into $a_t$.
> (4) Typo: In eq. (2), change $\frac{1}{m}\sum_{t=1}^{m}$ into $\frac{1}{n}\sum_{t=1}^{n}$ .
> (5) The author sequence of [14] should be Gang Wang, Songtao Lu, Georgios B. Giannakis, Gerald Tesauro, Jian Sun3
>
> **Our Response:** Thanks for catching this typo. We will fix the typo in our revised version of this paper.
>
>
> > **Your Comment:** (6) You might also cite the following papers on policy evaluation using linear function approximation.
>
> **Our Response:** Thank you for the pointers to these related references! In the revised version of this paper, we have included them into our manuscript and compared them with our work in the related work section.
>
> [R1] Du, Simon S., et al. "Stochastic variance reduction methods for policy evaluation." International Conference on Machine Learning. PMLR, 2017.
>
> [R2] Qiu, Shuang, et al. "Single-timescale stochastic nonconvex-concave optimization for smooth nonlinear TD learning." arXiv preprint arXiv:2008.10103 (2020).
>
> [R3] Wai, Hoi To, et al. "Multi-agent reinforcement learning via double averaging primal-dual optimization." Advances in Neural Information Processing Systems 2018 (2018): 9649-9660.
>
> [R4] Srikant, Rayadurgam, and Lei Ying. "Finite-time error bounds for linear stochastic approximation and TD learning." Conference on Learning Theory. PMLR, 2019.
>
> [R5] Wang, Gang, and Georgios B. Giannakis. "Finite-time error bounds for biased stochastic approximation with applications to Q-learning." International Conference on Artificial Intelligence and Statistics. PMLR, 2020.
>
> [R6] Sun, J., Wang, G., Giannakis, G. B., Yang, Q., & Yang, Z. (2020, June). Finite-time analysis of decentralized temporal-difference learning with linear function approximation. In International Conference on Artificial Intelligence and Statistics (pp. 4485-4495). PMLR.

---

> > ### Comment · Reviewer_ZsKo · 2021-08-14
> > **Reviewer ZsKo's 2nd comment**
> >
> > Thank you for your response.
> >
> > (1) I'm glad to see that there are some ICML and Neurips papers working on optimization error of fixed data without considering data distribution. Thank you for letting me know that.
> >
> > To my knowledge, this proposed algorithm with variance reduction+gradient tracking is also novel in decentralized minimax optimization. **Even after removing RL setting, the algorithm, assumption and theorem on decentralized minimax optimization can still remain unchanged, right? If so, I think this paper would better be adapted to the area of decentralized minimax optimization**, which has applications to not only RL. **About this suggestion, I'd like to hear from the other reviewers.**
> >
> > I will raise my rating to 6.
> >
> > (2) The variance reduction technique you used is called SPIDER, which has been used for centralized minimax optimization. [1] You might mention "SPIDER" and cite [1].
> >
> > Reference:
> >
> > [1] Xu, T., Wang, Z., Liang, Y., and Poor, H. V. Enhanced first and zeroth order variance reduced algorithms for min-max optimization. arXiv:2006.09361, 2020a.

---

> > > ### Author Response · Authors · 2021-08-16
> > > **Response to Reviewer ZsKo's 2nd Comments**
> > >
> > > Thank you for your further comments and your willingness to raise your rating! The followings are our responses to your updated comments.
> > >
> > > > **Your Comment:** *Even after removing RL setting, the algorithm, assumption and theorem on decentralized minimax optimization can still remain unchanged, right?*
> > >
> > > **Our Responses:** Correct. As you mentioned, beyond the multi-agent RL decentralized policy evaluation problem (MARL-DPE), our algorithms and theoretical results are still applicable for solving the other decentralized nonconvex-strongly-concave problems containing consensus constraints and satisfying the assumptions in this paper. For example, it is possible to adapt our work to solve other mini-max applications under decentralized network framework, such as primal-dual ERM problem [R4], fairness machine learning problem [R5], etc.
> > >
> > > > **Your Comment:** *If so, I think this paper would better be adapted to the area of decentralized minimax optimization, which has applications to not only RL.*
> > >
> > > **Our Responses:** Thanks for your constructive suggestions, which will increase the impacts of our results! In the revised version of this paper, we will follow your advice and add remarks in the introduction and technical sections to highlight other potential applications of our algorithmic results.
> > >
> > > Although our work has other potential applications, in terms of how to position our paper, we would like to emphasize that the MARL-DPE problem is still one of the most important motivating examples and one of the most appropriate applications to showcase the significance of lowing sample and communication complexities in decentralized min-max problems. This is also evidenced by the growing literature, e.g., Refs. [11, 12, 14, 24, 28, 37, 42, 46, 47] in the paper, although none of them considered both decentralized min-max and nonlinear function approximation at the same time. Indeed, our main goal of this work is to improve the state-of-the-art of RL DPE to consider both decentralized min-max optimization for MARL-DPE and nonlinear approximation (e.g., MARL with deep neural networks). Also, having a direct *structural comparison* with these existing works (see Table 1 in our paper) is the key reason that our paper is rooted in the RL context.
> > >
> > > To see why MARL-DPE is the most relevant context, we note that in many MARL applications, particularly in wireless edge networking environments (e.g., multi-agent robotic systems), each agent is equipped with very limited computation and communication resources, (e.g., the limited battery power and/or memory in a multi-robotic system [R1-2]). Thus, our sample-and-communication-efficient algorithms are critical to reduce the computation and communication workloads, which entail lower power consumption, low latency, lower memory storage requirements, etc. Also, as discussed in [R3], DPE is a major subroutine of general MARL for long-term team-averaged reward maximization. Therefore, reducing the sample complexity of DPE is highly desirable in accelerating the learning process of  MARL tasks.
> > >
> > > Due to the rationale stated above and our paper’s connections to the literature of MARL-DPE in terms of low sample and communication complexities, we still prefer to position our results in paper collectively as a building block that contributes to the field of MARL-DPE. But in the revised version, we will dedicate *a standalone section* pointing out that our algorithms, theoretical results, and proof techniques also contribute to the broader area of decentralized min-max optimization and could be of independent interest. Thanks!
> > >
> > >
> > > > **Your Comment:** *The variance reduction technique you used is called SPIDER, which has been used for centralized minimax optimization. [1] You might mention "SPIDER" and cite [1].*
> > >
> > > **Our Responses:** Your are correct. We agree that our proposed GT-SRVR and GT-SRVRI adopted similar techniques as in SPIDER [R6] and SPIDER-Boost [R7]. Such techniques were first proposed for centralized nonconvex minimization problem and then extended to SREDA [R8] and SREDA-Boost (Ref. [1] in your comment) for centralized nonconvex-strongly min-max problem. We thank you for the pointers to these related works. We will cite these papers and add further discussions in Section 2.
> > >
> > > However, here we would also like to point out that there are significant differences between our methods and SREDA/SREDA-Boost, which are summarized as follows:
> > >
> > > * *(Initialization):* SREDA/SREDA-Boost requires to use the iSARAH subroutine to initialize the dual variable (Line 4 in Algorithm 1 of [1]), which results in higher computational cost. However, there is no such requirement in our algorithms.
> > >
> > > * *(Hyperparameter Tuning):* Regarding the implementation complexity, SREDA/SREDA-Boost involve **seven** hyper-parameters to tune or estimate: initial accuracy $\zeta$, learning rates $\alpha_t$ $\beta$, batch-sizes $S_1$ $S_2$, period $q$ $m$ (Line 1 in Algorithm 1 of [1]). Additionally, in order to achieve a good convergence performance, SREDA/SREDA-Boost requires the prior knowledge of parameter $\kappa$ to properly select these hyper-parameters (cf. Theorem 1 in [1]). In contrast, in our proposed algorithm, GT-SRVR only requires **three** hyper-parameters to select: learning rates $\gamma$ and $\eta$, and period $q$, while GT-SRVRI requires **five** hyper-parameters:  those three parameters in GT-SRVR plus pre-set accuracy $\epsilon$ and batch-size incremental factor $\alpha$. In our Theorems 3&4, we show that the conditions on these hyper-parameters to guarantees the convergence of our algorithms are quite mild and easy to tune in practice, e.g., sufficiently small learning rates $\gamma$ and $\eta$. Thus, the implementation of our proposed GT-SRVR/GT-SRVRI algorithms is much simpler.
> > >
> > > * *(Simplicity):* SREDA/SREDA-Boost update primal and dual variables in an alternating fashion. For the dual/maximization part, SREDA/SREDA-Boost requires a concave maximization subroutine, which executes $m$ iterations for updating dual variable $y$. In contrast, our algorithms update primal and dual variables simultaneously, which yields a *single-loop* algorithm. In such a way, the implementation complexity of our algorithms is much lower than SREDA/SREDA-Boost.
> > >
> > > * *(Optimality):* With respect to  the convergence metric, SREDA/SREDA-Boost only focuses on the first-order stationarity of minimization part, i.e., $\|\nabla \Psi(x)\|$ (ref. to Theorem 1 in [1]). But our Theorem 3&4 adopted the metric containing the first-order stationarity of minimization part, optimality of maximization part, as well as network consensus. It can be shown that our metric is a stronger metric for decentralized nonconvex-strongly-concave mini-max problem in the sense that an $O(1/K)$ convergence rate under our metric implies an $O(1/K)$ convergence rate under metric in [1], while the converse is not necessarily true.
> > >
> > > * *(Applicability/Generality):* With regard to the applicability of the algorithms,  our proposed algorithms work for decentralized network systems (e.g., multi-agent RL). But SREDA/SREDA-Boost is only applicable for single-machine computation systems, and their extension to decentralized settings is still open and might be non-trivial.
> > >
> > > Due to the above unique features of GT-SRVR/GT-SRVRI compared with the existing works and SREDA/SREDA-Boost in particular,  we believe that this work significantly advances the state-of-the-art decentralized mini-max algorithms.
> > >
> > > [R1] Berahas, Albert S., et al. "Balancing communication and computation in distributed optimization." IEEE Transactions on Automatic Control 64.8 (2018): 3141-3155.
> > >
> > > [R2] Lavendelis, Egons, et al. "Multi-agent robotic system architecture for effective task allocation and management." Recent Researches in Communications, Electronics, Signal Processing & Automatic (2012): 22-24.
> > >
> > > [R3] Zhang, Kaiqing, et al. "Fully decentralized multi-agent reinforcement learning with networked agents." International Conference on Machine Learning. PMLR, 2018
> > >
> > > [R4] Zhang, Yuchen, and Xiao Lin. "Stochastic primal-dual coordinate method for regularized empirical risk minimization." International Conference on Machine Learning. PMLR, 2015.
> > >
> > > [R5] Baharlouei, Sina, et al. "Rényi Fair Inference." International Conference on Learning Representations. 2019.
> > >
> > > [R6] Fang, Cong, et al. "SPIDER: near-optimal non-convex optimization via stochastic path integrated differential estimator." Proceedings of the 32nd International Conference on Neural Information Processing Systems. 2018.
> > >
> > > [R7] Wang, Zhe, et al. "Spiderboost and momentum: Faster variance reduction algorithms." Advances in Neural Information Processing Systems 32 (2019): 2406-2416.
> > >
> > > [R8] Luo, Luo, et al. "Stochastic Recursive Gradient Descent Ascent for Stochastic Nonconvex-Strongly-Concave Minimax Problems." Advances in Neural Information Processing Systems 33 (2020).

---

> > > > ### Comment · Reviewer_ZsKo · 2021-08-16
> > > > **Reviewer ZsKo's 3rd Comments**
> > > >
> > > > Thanks for your response.
> > > >
> > > > **Now I see your more contributions to the decentralized minimax area and would like to further raise my rating to 7.**
> > > >
> > > > Standing in either MARL-DPE or decentralized minimax area looks fine for me. In either case, I think it better to add bullet point(s) in the introduction to include your above-mentioned contributions to the decentralized minimax area, at least briefly summarize.

---

> > > > > ### Author Response · Authors · 2021-08-16
> > > > > **Response to ZsKo's 3rd Comments**
> > > > >
> > > > > Thanks so much for your further comments! Yes, we agree that it will be a great idea to add the bullets in the introduction to include our above-mentioned contributions to the field of decentralized minimax optimization. Will do!
> > > > >
> > > > > Thanks so much again for all your constructive comments, which significantly improve the quality of our work!

---

### Official Review · Reviewer_6xn4 · 2021-07-14

**Rating:** 7
**Confidence:** 3

**Summary:**

This paper considers decentralized MARL policy evaluation with nonlinear function approximation. The authors first reformulate the policy evaluation problem as a decentralized nonconvex-strongly-concave minimax saddle point problem and then develop a decentralized gradient-based descent ascent algorithm called GT-GDA that enjoys a convergence rate of O(1/T ).


**Limitations And Societal Impact:**

Yes

**Main Review:**

Pros:
This work is the first to investigate the decentralized PE (DPE) problem for MARL with nonlinear function approximation.

Cons:
GT-GDA has the same sample complex and communication complexity as DHPD [1] in which linear function approximations are considered. It is not clear to me, from both practical and theoretical perspectives, the advantage of considering nonlinear functional approximations.

[1] Dongsheng Ding, Xiaohan Wei, Zhuoran Yang, Zhaoran Wang, and Mihailo R Jovanovic. Fast multi-agent temporal-difference learning via homotopy stochastic primal-dual method. In Optimization Foundations for Reinforcement Learning Workshop, 33rd Conference on Neural Information Processing Systems, 2019.

It will be nice to provide some numerical experiments to highlight and demonstrate the advantage of the proposed framework as there are several closely related works such as [2] and [3].

[2] Hoi-To Wai, Mingyi Hong, Zhuoran Yang, Zhaoran Wang, and Kexin Tang. Variance reduced policy evaluation with smooth function approximation. Advances in Neural Information Processing Systems, 32:5784–5795, 2019.

[3] Hoi-To Wai, Zhuoran Yang, Zhaoran Wang, and Mingyi Hong. Multi-agent reinforcement learning via double averaging primal-dual optimization. In Advances in Neural Information Processing Systems, pages 9649–9660, 2018.



#############after authors' response#############

I am happy with the responses from the authors (especially on the further elaboration of the contributions and explanation on why the direct comparisons with existing algorithms are not applicable). I will raise my score to 7.


**Time Spent Reviewing:**

4

---

> ### Author Response · Authors · 2021-08-10
> **Response to Reviewer 6xn4's Comments**
>
> > **Your Comment:** 1. GT-GDA has the same sample complex and communication complexity as DHPD [1] in which linear function approximations are considered. It is not clear to me, from both practical and theoretical perspectives, the advantage of considering nonlinear functional approximations.
>
> **Our Response:** We studied the non-linear approximation setting because the deep neural network models, which are highly non-linear, have been shown to be highly effective in function approximation in many reinforcement learning algorithms and applications (e.g., [R1-3]). In addition, non-linear neural network models significantly outperform the traditional linear approximation schemes in the following aspects: 1) Linear approximation schemes are based on their pre-defined basis space, which may not be able to approximate the non-linear value function with high accuracy; 2) Non-linear neural network approximation can handle the cases where the states space that is mixed with continuous and (infinite) discrete state values; 3) Non-linear neural network approximation usually have a better generalization performance than linear approximation [R4-6].
>
> > **Your Comment:** 2. It will be nice to provide some numerical experiments to highlight and demonstrate the advantage of the proposed framework as there are several closely related works such as [2] and [3].
>
> **Our Response:** We thank the reviewer for the pointers to these related works (they have been cited and discussed in our related work section). We didn’t compare with these works due to the following reasons:
>
> *Ref. [2] only focused on single-agent policy evaluation, while our algorithm is designed for multi-agent reinforcement learning policy evaluation, which requires parameter consensus over a network. Thus, Ref. [2] is not directly comparable to our work.
>
> *Ref. [3] considers linear approximation for its value function, while we consider non-linear approximation (e.g., neural network models). Due to this key difference, the convergence performance of Ref. [3] is not directly comparable to that of our work.
>
> [R1] Van Hasselt, Hado, Arthur Guez, and David Silver. "Deep reinforcement learning with double q-learning." Proceedings of the AAAI conference on artificial intelligence. Vol. 30. No. 1. 2016.
>
> [R2] Li, Li, Yisheng Lv, and Fei-Yue Wang. "Traffic signal timing via deep reinforcement learning." IEEE/CAA Journal of Automatica Sinica 3.3 (2016): 247-254.
>
> [R3] Li, Yuxi. "Deep reinforcement learning: An overview." arXiv preprint arXiv:1701.07274 (2017).
>
> [R4] Zhang, Chiyuan, et al. "A study on overfitting in deep reinforcement learning." arXiv preprint arXiv:1804.06893 (2018).
>
> [R5] Zhang, Amy, Nicolas Ballas, and Joelle Pineau. "A dissection of overfitting and generalization in continuous reinforcement learning." arXiv preprint arXiv:1806.07937 (2018).
>
> [R6] Kawaguchi, Kenji, Leslie Pack Kaelbling, and Yoshua Bengio. "Generalization in deep learning." arXiv preprint arXiv:1710.05468 (2017).

---

> ### Author Response · Authors · 2021-08-24
> **Response to Reviewer 6xn4's Updated review**
>
> We thank Reviewer 6xn4 again for the updated review and the increased score to 7! All your reviews, comments, questions, and suggestions have significantly improved the quality of our work! We will definitely incorporate all your suggestions and comments in the revised version of this paper!

---

### Official Review · Reviewer_d6sP · 2021-07-17

**Rating:** 7
**Confidence:** 3

**Summary:**

This paper studies the problem of policy evaluation in a decentralized multi-agent reinforcement learning setting with non-linear function approximation. The authors present a gradient tracking based descent-ascent optimization algorithm GT-GDA that aggregates global gradients by a network-weighted aggregation, and consequently updates local parameters by descent/ascent. The authors establish convergence of the proposed algorithm and additionally analyse a reduced-variant extension. Experimental results establish convergence in several numerical benchmarks.

**Limitations And Societal Impact:**

The authors have not discussed the limitations of their work.

**Main Review:**

Originality and quality: The paper makes a complete analysis of the presented algorithm for decentralized policy evaluation in the nonlinear setting, and additionally analyses a variant with reduced variance as well. Few points:
- Given that the authors have discussed a new problem setting, it would be helpful to pin-point the exact technical contributions (from both algorithmic and theoretical points of view), as it is unclear from the current manuscript what aspects of the paper are derived from existing research.
- The motivation for the problem is still lacking: I would imagine the non-linear setting to be more difficult than the linear function approximation setting, however, it is unclear to me what technical challenges one would face in extending linear algorithms to the non-linear setting.
- The experimental results are a bit lacking: only one problem is examined, and there are no baselines to compete with. While it is clear that the authors are tackling _decentralized nonlinear_ function approximation, it would be interesting nonetheless to compare their algorithm with linear baselines in a linear (or nearly-linear) environment. Additionally, the ablations in the supplementary material can be extended to larger systems to examine the effect of sparsity: the 6-node system considered does not seem challenging based on Figure 6.

Clarity: The paper is generally well-written but requires proof-reading to remove spelling and grammar issues.

Significance: The contributions are focused and tackle a somewhat niche problem (decentralized nonlinear policy evaluation).

Specific Questions:
- How do the potential functions described in Theorems 1 and 2 compare with existing single-agent and linear function approximation potentials?


**Time Spent Reviewing:**

6

---

> ### Author Response · Authors · 2021-08-10
> **Response to Reviewer d6sP's Comments**
>
> > **Your Comment:** 1. Given that the authors have discussed a new problem setting,..., as it is unclear from the current manuscript what aspects of the paper are derived from existing research.
>
> **Our Response:** Thanks for your comment. We have summarized our contributions in the bullets in Section 1 and Section 2.3. Here, we would like to further clarify our technical contributions from both theoretical and algorithmic points of view:
>
> * Most of the existing works studied the policy evaluation problem under either linear approximation or single-agent framework. In our work, we take the first attempt to solve the ***decentralized*** policy evaluation (DPE) problem with ***non-linear*** approximation for cooperative MARL. Note that the prior work in [R1] has shown that general TD algorithms might not converge for non-linear value function approximation. In our work, we proposed three DPE algorithms and established the first theoretical results for their convergence performance.
>
> * We reformulation the DPE problem as a decentralized non-convex-strongly-concave optimization problem, which requires consensus on both primal and dual variables, which is much more difficult than conventional decentralized minimization problems. Toward this end, we proposed a `hybrid’ scheme that non-trivially integrates gradient tracking and variance reduction techniques for both primal and dual variables to achieve significantly improved convergence performance. We note that this hybrid scheme also introduced more sophisticated algorithmic structures that necessitate new proof techniques.
>
> * To reduce the sample complexity and balance the communication complexity, we proposed a “double variance reduction” scheme in GT-SRVR for both primal and dual variable updates. We showed that the sample complexity is reduced to $\mathcal{O}(m\sqrt{n}\epsilon^2)$ while the communication complexity is kept at $\mathcal{O}(\epsilon^2)$. To further accelerate the algorithm and reduce sample complexity, we developed GT-SRVRI, which has a simple algorithmic structure and is easy to be implemented. All the above results are new in the literature.
>
> * In our theoretical study, we relax the commonly assumed compactness condition on the feasible domains with Assumption (d-e) some mild assumption on the objective function. This result may be of independent interest for many general decentralized mini-max problem.
>
> > **Your Comment:** 2. The motivation for the problem is still lacking...what technical challenges one would face in extending linear algorithms to the non-linear setting.
>
> **Our Response:** Thanks for your comment. Here, we would like to explain the difference between linear and nonlinear approximations for policy evaluation. Under linear approximation for policy evaluation, the problem boils down to finding a solution for a linear equation system, which is in essence similar to solving a relatively easy strongly-convex optimization problem. In stark contrast, under non-linear approximation for policy evaluation, the problem possesses a non-convex-strongly-concave structure, and it is far more challenging to find a saddle point solution. In our work, we developed algorithms to efficiently find the first-order stationary point for the primal variable and optimal point for dual variable. Furthermore, our problem becomes harder as we require to find the ***consensus*** solution under decentralized setting. To our best knowledge, this problem has not been considered in the literature.
>
> > **Your Comment:** 3. The experimental results are a bit lacking: only one problem is examined... the 6-node system considered does not seem challenging based on Figure 6.
>
> **Our Response:** Due to the significant differences in models and MSPBE objectives between non-linear and linear approximation settings, it would be unfair to compare the convergence of our non-linear approximation algorithm with the other algorithms with linear approximation as baselines. But we agree with the reviewer that our paper will benefit from having further experimental results. In this rebuttal period, we have been able to run additional experiments and add new experimental results on the ***prediction performance*** comparison between the non-linear approximation from our algorithms and the linear approximation from existing works.
>
>
> For additional experiments, we first compare Mean Squared Error with the ground truth value function and the estimated value function over 3 independent runs under original experiment setting of our paper with nonlinear approximation. We note that the ground truth can be calculated using tabular policy evaluation and the estimated value functions is learned by our stated algorithms SRVR and SRVR$\mathcal{I}$. The mean square error (MSE) of SRVR is $0.084 \pm 0.003$, MSE of SRVR$\mathcal{I}$ is $0.092 \pm 0.005$. Furthermore, with linear approximation apply on our stated algorithms, we have MSE result as $0.1747 \pm 0.012$ on SRVR and $0.1895 \pm 0.014$ on SRVR$\mathcal{I}$.
>
> Moreover, in this rebuttal period, we have been able to add experimental results for larger network systems (20 nodes) in the revised version of this paper.
>
>  Specifically, we adopt an additional experiment on the environment of *Cooperative Navigation* task in [27], which consists of **20 agents**. The learning rate is fixed at $0.1$ and the discount factor is $0.95$.
> We can also observe that both GT-SRVR and GT-SRVR$\mathcal{I}$ have better sample efficiency and converges fast from Table 1-4.
>
> Table1. MSPBE value vs. communication round.
>
> |Algorithms$\backslash$Iterations|0|5|20|90|
> |:---:|:------:|:------:|:------:|:------:|
> |DSGDA|$1.022\times10^5$|$1.198\times10^{-2}$|$2.195\times10^{-1}$|$1.117\times10^{-1}$|
> |GT-SGDA|$1.022\times10^5$|$1.626\times10^{-2}$|$2.183\times10^{-4}$|$1.192\times10^{-5}$|
> |GT-GDA|$1.022\times10^5$|$4.299\times10^{-3}$|$7.415\times10^{-5}$|$1.921\times10^{-6}$|
> |GT-SRVR|$1.022\times10^5$|$4.232\times10^{-3}$|$7.416\times10^{-5}$|$1.926\times10^{-6}$|
> |GT-SRVR$\mathcal{I}$|$1.022\times10^5$|$4.369\times10^{-3}$|$7.391\times10^{-5}$|$2.002\times10^{-6}$|
>
>
> Table2. Convergence metric vs. communication round.
>
> |Algorithms$\backslash$Iterations|0|5|20|90|
> |:---:|:------:|:------:|:------:|:------:|
> |DSGDA|$1.676\times10^2$|$3.820\times10^{-3}$|$3.178\times10^{-3}$|$3.177\times10^{-3}$|
> |GT-SGDA|$1.676\times10^2$|$1.569\times10^{-3}$|$2.777\times10^{-4}$|$1.820\times10^{-5}$|
> |GT-GDA|$1.676\times10^2$|$8.817\times10^{-4}$|$1.709\times10^{-4}$|$9.154\times10^{-7}$|
> |GT-SRVR|$1.676\times10^2$|$8.846\times10^{-4}$|$1.705\times10^{-4}$|$9.177\times10^{-7}$|
> |GT-SRVR$\mathcal{I}$|$1.676\times10^2$|$8.343\times10^{-4}$|$1.700\times10^{-4}$|$9.595\times10^{-7}$|
>
> Table3. MSPBE value vs. sample complexity.
>
> |Algorithms$\backslash$Samples|200|500|1000|1500|
> |:---:|:------:|:------:|:------:|:------:|
> |DSGDA|$5.167\times10$|$1.702$|$0.102$|$0.101$|
> |GT-SGDA|$7.635\times10^{-4}$|$6.992\times10^{-5}$|$2.037\times10^{-5}$|$2.036\times10^{-5}$|
> |GT-GDA|$1.022\times10^5$|$3.747\times10^{3}$|$1.013\times10^{-2}$|$1.010\times10^{-2}$|
> |GT-SRVR|$1.022\times10^5$|$8.208\times10^{-4}$|$4.107\times10^{-5}$|$1.777\times10^{-5}$|
> |GT-SRVR$\mathcal{I}$|$1.858\times10^{-4}$|$2.919\times10^{-5}$|$5.114\times10^{-6}$|$5.113\times10^{-6}$|
>
> Table4. Convergence metric vs. sample complexity.
>
> |Algorithms$\backslash$Samples|200|500|1000|1500|
> |:---:|:------:|:------:|:------:|:------:|
> |DSGDA|$3.820\times10^{-3}$|$3.177\times10^{-3}$|$3.177\times10^{-3}$|$3.177\times10^{-3}$|
> |GT-SGDA|$4.534\times10^{-4}$|$1.602\times10^{-4}$|$3.997\times10^{-5}$|$1.290\times10^{-5}$|
> |GT-GDA|$1.676\times10^2$|$3.398\times10$|$1.257\times10^{-3}$|$6.712\times10^{-4}$|
> |GT-SRVR|$1.676\times10^2$|$4.634\times10^{-4}$|$9.792\times10^{-5}$|$3.245\times10^{-5}$|
> |GT-SRVR$\mathcal{I}$|$2.730\times10^{-4}$|$6.363\times10^{-5}$|$4.591\times10^{-6}$|$4.907\times10^{-7}$|
>
> > **Your Comment:** 4. Clarity: The paper is generally well-written but requires proof-reading to remove spelling and grammar issues.
>
> **Our Response:** Thanks for your suggestions. We will carefully proofread and fix the spelling and grammar issues in the revised version of this paper.
>
> > **Your Comment:** 5. How do the potential functions described in Theorems 1 and 2 compare with existing single-agent and linear function approximation potentials?
> **Our Response:** Thanks for your question and we would further clarify the potential function in here. There are six terms in our potential function:
>
> * $J(\bar{\theta}_t)$ measures the empirical MSBPE on the averaged primal variable, which would be close to zero if the value function is best fit;
>
> * $||\bar{\omega}_t-\omega^*_t||$ measures the optimality of the dual variable,  which is zero when it finds the unique solution for the maximization part;
>
> * The third and forth term, $\frac{1}{m}\sum||\theta_{i,t}-\bar{\theta}_t||^2+||\omega_{i,t}-bar{\omega}_t||^2$, measure the consensus error on the primal and dual variables; and the last two terms are the consensus error on the tracked gradient. In comparison, the single-agent PE problem [R2] does not have the last four consensus error terms over multi-agents, and so it is dramatically different from our DPE problem.
>
> *Also, for the linear approximation PE problem in [R3], the first term would be replaced with $||\bar{\theta_{t}} - \theta^*||^2$ as it can be viewed as a strongly convex optimization, while the other terms are the same.
>
> [R1] Tsitsiklis, John N., and Benjamin Van Roy. "Analysis of temporal-diffference learning with function approximation." Advances in neural information processing systems. 1997.
>
> [R2] Qiu, Shuang, et al. "Single-timescale stochastic nonconvex-concave optimization for smooth nonlinear TD learning." arXiv preprint arXiv:2008.10103 (2020)
>
> [R3] Wai, Hoi To, et al. "Multi-agent reinforcement learning via double averaging primal-dual optimization." Advances in Neural

---

> > ### Comment · Reviewer_d6sP · 2021-08-23
> > **Thanks for the response**
> >
> > Thanks to the authors for answering my queries. After going through the discussions and responses with other reviewers I have decided to update my score.

---

> > > ### Author Response · Authors · 2021-08-23
> > > **Response to Reviewer d6sP's 2nd Comments**
> > >
> > > Thanks for your positive feedback and constructive comments! We will incorporate your suggestions and comments in the revised version of this paper!

---

### Official Review · Reviewer_qMVk · 2021-07-20

**Rating:** 5
**Confidence:** 3

**Summary:**

This paper studies decentralized cooperative policy evaluation. The main idea is to reformulate it as some standard decentralized optimization problem and then apply the gradient tracking and variance reduction technique.


**Limitations And Societal Impact:**

This work is purely theoretical. I am not aware of any potential negative societal impact.


**Main Review:**

I feel the problem setting is kind of artificial and is not well motivated by real-world applications. In this setting, the network topology is static and the value function of each agent has the same parameterization (e.g., if the value function is a neural network, then each agent shall use the same NN architecture). I cannot think of a practical application where all these conditions are met while we still cannot do centralized PE. Also in this setting, why do we want to do PE?

The technical contributions look marginal. Formulating decentralized PE as a decentralized optimization problem is straightforward, and the techniques for solving the reformulated problem have been widely studied in the decentralized optimization literature.


**Time Spent Reviewing:**

3

---

> ### Author Response · Authors · 2021-08-10
> **Response to Reviewer qMVk's Comments**
>
> > **Your Comment:** 1. I feel the problem setting is kind of artificial and is not well motivated by real-world applications. In this setting, the network topology is static and the value function of each agent has the same parameterization (e.g., if the value function is a neural network, then each agent shall use the same NN architecture). I cannot think of a practical application where all these conditions are met while we still cannot do centralized PE. Also in this setting, why do we want to do PE?
>
> **Our Response:** In this work, we focus on the optimization problem for the multi-agent policy evaluation. The adopted multi-agent system is a well-formulated framework and has been widely studied in the literature of both theoretical and applied machine learning [R1-4]. One of our major contributions is that we leveraged the neural network to approximate the joint value function, which is a complex non-linear function and make our objective become non-convex-strongly-concave function. As neural network is a powerful tool for function approximation in RL, we believe our algorithm can be adopted in many real-world applications. The following are point-to-point responses to your questions:
>
> * *Static Network:* Our algorithms and theoretical results can be directly generalized to the stochastic network setting. First, our algorithms can be directly applied to time-varying network topologies, as long as the network is connected in each time-slot. Further, our algorithm can work with the cases where the network could be disconnected at any particular time instant, as long as it satisfies the $B$-connectedness as in [R8-9] (i.e., the network graph is connected averaged over $B$ communication rounds).
>
> * *Common Value Function with Same Parameterization:* The reason that the agents share a common value function (i.e., with same parameterization) is that we study the ***cooperative*** multi-agent reinforcement learning – a foundational MARL paradigm (see the comprehensive survey on MARL in [R1] below). In cooperative MARL, the value function is based on the ***joint states***. As a result, the parameterization of the value function is the same across the agents. Note that cooperative MARL has found many useful applications (e.g., Traffic light control [R5], autonomous driving [R6], finantical trading [R7].
>
> * *Decentralized Setting for PE:* As we discussed in section 2, we consider the decentralized learning setting due to i) the locally observed rewards are private and cannot be shared with the other agents/central server; ii) it would be difficult to set up a central sever in many MARL applications while decentralized setting is more flexible (e.g.,wireless network [R15], UAV network [R16]); iii) the central server is more vulnerable to cyber-attacks and would be a significant communication bottleneck. [R13-14].
>
> > **Your Comment:** 2. The technical contributions look marginal. Formulating decentralized PE as a decentralized optimization problem is straightforward, and the techniques for solving the reformulated problem have been widely studied in the decentralized optimization literature.
>
> **Our Response:** Thanks for your comment. Here, we would like to further clarify our contributions as follows:
> * For decentralized policy evaluation in MARL, we proposed new algorithms to solve the ***non-convex-strongly-concave min-max*** problem under ***decentralized*** setting, which is significantly harder than the widely studied minimization problem in decentralized settings.
>
> * To efficiently solve the decentralized min-max problem, our algorithms are primal-dual algorithms where we update the two variables simultaneously rather than alternatively. Thus, our proposed algorithms are much simpler and significantly different from existing related works that require to solve maximization subproblem for dual variable at each iteration.
>
> * We proposed a strong convergence metric (see Eq. (7)), including the first-order stationary for primal variable, optimality for dual variable and consensus for both primal and dual variables. Based on this new convergence metric, we established a convergence rate of $O(1/K)$, which is also new in the literature.
>
> * We further proposed a “hybrid” scheme that non-trivially integrates variance reduction and gradient tracking techniques for both primal and dual variables. Compared with algorithms [R11-12] with simple stochastic gradient updates and variable mixing, our proposed scheme enjoys much improved theoretical and numerical performances. We note that the theoretical analysis of the more sophisticated hybrid scheme is more involved compared to existing works and necessitates new proof techniques.
>
> [R1] Zhang, Kaiqing, Zhuoran Yang, and Tamer Başar. "Multi-agent reinforcement learning: A selective overview of theories and algorithms." Handbook of Reinforcement Learning and Control(2021): 321-384
>
> [R2] Tan, Ming. "Multi-agent reinforcement learning: Independent vs. cooperative agents." Proceedings of the tenth international conference on machine learning. 1993.
>
> [R3] Busoniu, Lucian, Robert Babuska, and Bart De Schutter. "Multi-agent reinforcement learning: A survey." 2006 9th International Conference on Control, Automation, Robotics and Vision. IEEE, 2006.
>
> [R4] Zhang, Kaiqing, et al. "Fully decentralized multi-agent reinforcement learning with networked agents." International Conference on Machine Learning. PMLR, 2018.
>
> [R5] Wiering, Marco A. "Multi-agent reinforcement learning for traffic light control." Machine Learning: Proceedings of the Seventeenth International Conference (ICML'2000). 2000.
>
> [R6] Kiran, B. R., Sobh, I., Talpaert, V., Mannion, P., Al Sallab, A. A., Yogamani, S., & Pérez, P. (2021). Deep reinforcement learning for autonomous driving: A survey. IEEE Transactions on Intelligent Transportation Systems.
>
> [R7] Lee, Jae Won, and Byoung-Tak Zhang. "Stock trading system using reinforcement learning with cooperative agents." Proceedings of the Nineteenth International Conference on Machine Learning. 2002.
>
> [R8] Nedic, Angelia, Alex Olshevsky, and Wei Shi. "Achieving geometric convergence for distributed optimization over time-varying graphs." SIAM Journal on Optimization 27.4 (2017): 2597-2633.
>
> [R9] Nedić, Angelia, and Alex Olshevsky. "Distributed optimization over time-varying directed graphs." IEEE Transactions on Automatic Control 60.3 (2014): 601-615.
>
> [R10] Rogozin, Alexander, and Alexander Gasnikov. "Projected gradient method for decentralized optimization over time-varying networks." arXiv preprint arXiv:1911.08527 (2019).
>
> [R11] Liu, Mingrui, et al. "A Decentralized Parallel Algorithm for Training Generative Adversarial Nets." arXiv preprint arXiv:1910.12999 (2019).
> [R12] Liu, Weijie, et al. "A decentralized proximal point-type method for saddle point problems." arXiv preprint arXiv:1910.14380(2019).
>
> [R13] Wu, Zhaoxian, et al. "Byzantine-resilient decentralized TD learning with linear function approximation." IEEE Transactions on Signal Processing (2021).
>
> [R14] Lian, Xiangru, et al. "Can decentralized algorithms outperform centralized algorithms? a case study for decentralized parallel stochastic gradient descent." Proceedings of the 31st International Conference on Neural Information Processing Systems. 2017.
>
> [R15] Yao, Fuqiang, and Luliang Jia. "A collaborative multi-agent reinforcement learning anti-jamming algorithm in wireless networks." IEEE Wireless Communications Letters 8.4 (2019): 1024-1027.
>
> [R16] Cui, Jingjing, Yuanwei Liu, and Arumugam Nallanathan. "Multi-agent reinforcement learning-based resource allocation for UAV networks." IEEE Transactions on Wireless Communications 19.2 (2019): 729-743.

---

### Decision · Program_Chairs · 2021-09-27

**Decision:**

Accept (Poster)

**Comment:**

This paper studies decentralized policy evaluation for cooperative multi-agent RL. The main approach of this paper is (1) reduce the problem to a minimax optimization problem, and (2) develop new algorithms for solving this minimax problem using gradient tracking and variance reduction technique. As pointed out by reviewers, the main concerns of this paper are that (a) the main technique contribution is mostly in optimization, while the paper currently mainly sells as a RL paper, without detailed comparison in terms of rate with existing optimization algorithms. (b) the strongly concave assumption lacks justification in the RL setting. Except these, most reviewers are convinced that the paper is well-written, the results are solid, and this paper makes interesting contributions. We hope authors can address above concerns in the final version.